# naRNA-LL37 composite DAMPs define sterile NETs as self-propagating drivers of inflammation

Francesca Bork [iD][1], Carsten L Greve[1], Christine Youn[2], Sirui Chen[1], Vinicius N C Leal[1,14], Yu Wang[2], Berenice Fischer [iD][3], Masoud Nasri[4], Jule Focken[5], Jasmin Scheurer[5], Pujan Engels [iD][1], Marissa Dubbelaar [iD][6,7], Katharina Hipp [iD][8], Baher Zalat [iD][1], Andras Szolek[1], Meng-Jen Wu[2], Birgit Schittek [iD][5,9,10], Stefanie Bugl[1], Thomas A Kufer [iD][11], Markus W Löffler [iD][6,9,12], Mathias Chamaillard [iD][13], Julia Skokowa [iD][4,9], Daniela Kramer[3], Nathan K Archer [iD][2] & Alexander N R Weber [iD][1,9,10][✉]

## Abstract

**Neutrophil extracellular traps (NETs) are a key antimicrobial feature of cellular innate immunity mediated by polymorphonuclear neutrophils (PMNs). NETs counteract microbes but are also linked to inflammation in atherosclerosis, arthritis, or psoriasis by unknown mechanisms. Here, we report that NET-associated RNA (naRNA) stimulates further NET formation in naive PMNs via a unique TLR8-NLRP3 inflammasome-dependent pathway. Keratinocytes respond to naRNA with expression of psoriasis-related genes (e.g., IL17, IL36) via atypical NOD2-RIPK signaling. In vivo, naRNA drives temporary skin inflammation, which is drastically ameliorated by genetic ablation of RNA sensing. Unexpectedly, the naRNA-LL37 'composite damage-associated molecular pattern (DAMP)' is pre-stored in resting neutrophil granules, defining sterile NETs as inflammatory webs that amplify neutrophil activation. However, the activity of the naRNA-LL37 DAMP is transient and hence supposedly self-limiting under physiological conditions. Collectively, upon dysregulated NET release like in psoriasis, naRNA sensing may represent both a potential cause of disease and a new intervention target.**

**Keywords** Neutrophil Extracellular Trap; RNA; Toll-like Receptors; NLRP3 Inflammasome; DAMP
**Subject Categories** Immunology; RNA Biology; Signal Transduction

## Introduction

The formation of neutrophil extracellular traps (NETs), since its discovery in 2004 (Brinkmann et al, 2004), has emerged as a fascinating phenomenon of host defense. Hereby neutrophils, the primary leukocyte population, extrude their genomic DNA to form web-like structures that were proposed to be able to trap and kill microbial invaders such as bacteria or fungi (reviewed in Kruger et al, 2015). DNA is thus a defining structural and functional feature of NETs. In addition, DNA-associated proteins, histones and High-Mobility-Group-Protein B1 (HMGB-1), antimicrobial peptides like LL37, and enzymes such as myeloperoxidase (MPO) are released during the formation of NETs and contribute to their antimicrobial function (Kruger et al, 2015). While their primary role has been firmly established to be antimicrobial, the reported involvement of NETs in sterile inflammatory conditions such as atherosclerosis (Warnatsch et al, 2015), rheumatoid arthritis (RA) (Song et al, 2020) and COVID-19 (Middleton et al, 2020) has remained somewhat enigmatic. More than 1000 publications have sought to detail the multi-faceted phenomenon of NET formation and effector function execution (reviewed in Boeltz et al, 2019; Melbouci et al, 2023). Specifically, investigations of the inflammation-promoting effects of sterile NETs focused on DNA and protein components (e.g., histones or enzymes like neutrophil elastase or myeloperoxidase) (Lai et al, 2020; Tsourouktsoglou et al, 2020). Instead, another primary cellular biomolecule, RNA, has so far received very little attention in the context of vertebrate NETs. Interestingly, a study in insects showed that hemocytes (macrophage-like immune cells) can release both DNA and RNA in NET-

[1]Institute of Immunology, Department of Innate Immunity, University of Tübingen, Auf der Morgenstelle 15, 72076 Tübingen, Germany. [2]Department of Dermatology, Johns Hopkins University School of Medicine, Baltimore, MD 21231, USA. [3]Department of Dermatology, University Medical Center of the Johannes Gutenberg-University Mainz, Mainz, Germany. [4]Division of Translational Oncology, Department of Oncology, Hematology, Clinical Immunology and Rheumatology, University Hospital Tübingen, Otfried-Müller Str. 10, 72076 Tübingen, Germany. [5]Department of Dermatology, University Hospital Tübingen, Liebermeisterstr. 25, 72076 Tübingen, Germany. [6]Institute of Immunology, Department of Peptide-based Immunotherapy, University of Tübingen, Auf der Morgenstelle 15, 72076 Tübingen, Germany. [7]Quantitative Biology Center (QBiC), University of Tübingen, Auf der Morgenstelle 10, 72076 Tübingen, Germany. [8]Electron Microscopy Facility, Max Planck Institute for Biology Tübingen, Max-Planck-Ring 5, 72076 Tübingen, Germany. [9]iFIT – Cluster of Excellence (EXC 2180) "Image-Guided and Functionally Instructed Tumor Therapies", University of Tübingen, Tübingen, Germany. [10]CMFI – Cluster of Excellence (EXC 2124) "Controlling microbes to fight infection", University of Tübingen, Tübingen, Germany. [11]Institute of Nutritional Medicine, Department of Immunology, University of Hohenheim, Fruwirthstr. 12, 70593 Stuttgart, Germany. [12]Institute for Clinical and Experimental Transfusion Medicine, Medical Faculty, University of Tübingen, Otfried-Müller-Str. 4/1, 72076 Tübingen, Germany. [13]University of Lille, CNRS, Inserm, CHU Lille, Institut Pasteur de Lille, U1019 - UMR 8204 - CIIL - Centre d'Infection et d'Immunité de Lille, F-59000 Lille, France. [14]Present address: Laboratory of Immunogenetics, Department of Immunology, Institute of Biomedical Science, University of São Paulo (USP), São Paulo, SP, Brazil. [✉]E-mail: alexander.weber@uni-tuebingen.de

like structures during microbe-triggered clotting reactions, and in response to either extracellular RNA or DNA (Altincicek et al, 2008). Our lab recently showed that human polymorphonuclear neutrophils (PMNs) are responsive to RNA, but not DNA, in combination with the antimicrobial peptide LL37 (Herster et al, 2020). In addition, earlier work described a role for DNA- and RNA-LL37 complexes in the activation of plasmacytoid dendritic cells (Ganguly et al, 2009; Lande et al, 2007). This has given rise to the notion that under certain conditions, e.g., in psoriasis patients, tolerance to self-DNA and -RNA can be 'accidentally' broken by LL37 (Lande et al, 2015) involving the nucleic acid-sensing Toll-like receptor (TLR) 7, TLR8 (ssRNA) and TLR9 (DNA) in humans, and Tlr7, Tlr13 and Tlr9 in mice (Eigenbrod and Dalpke, 2015; Lind et al, 2022). Synthetic or viral double-stranded RNA was also shown to act in concert with LL37 (Adase et al, 2016; Kato et al, 2023) and, in general, many roles have been ascribed to so-called extracellular RNA (exRNA): for example, macrophage polarization, recruitment of leukocytes to the site of inflammation, leukocyte rolling on the vascular endothelium, as well as integrin-mediated firm adhesion of immune cells and promotion of thrombosis (Preissner et al, 2020). In addition, exRNA is considered a reliable biomarker for various diseases such as cancer or cardiovascular pathologies (Schmidt et al, 2005; Zernecke and Preissner, 2016). Under sterile conditions, vascular injury, tissue damage, or ischemia have been suggested to trigger release of exRNA along with other cellular material (Preissner et al, 2020). Nevertheless, it remains unclear whether the physiologically detectable amount of exRNA in the serum of patients result from these relatively slow processes, or rather, an as-yet unidentified, more rapid process of RNA extrusion has to be postulated.

We show here that RNA contained in NETs, so-called NET-associated RNA or naRNA, drives such a self-amplifying inflammatory loop not only engaging PMNs but also macrophages and keratinocytes. naRNA responsiveness was dependent on TLR8 and Tlr13 in human and murine myeloid immune cells, respectively. In addition, we identified a unique, priming-independent NLRP3-ASC-caspase-1-gasdermin D (GSDMD) inflammasome pathway in neutrophils, connecting TLR8 and NET formation in response to naRNA. Notably, in mice naRNA caused considerable skin inflammation in a Tlr13-dependent manner. Furthermore, in a well-established model of psoriatic skin inflammation, genetic ablation of RNA sensing strongly ameliorated skin inflammation. Importantly, an observed pre-association of naRNA with LL37 in resting neutrophils from healthy donors indicates that the NET process inevitably entails extrusion of this novel 'composite' damage-associated molecular pattern (DAMP). Not only does this have ramifications for concepts of tolerance to self-RNA; it also identifies naRNA as a sterile pro-inflammatory agent of early immune responses. As most pathological conditions involving NETs are sterile, our data argue that a key function of sterile NET formation is DAMP release, with naRNA-LL37 acting as a molecular beacon to report on PMN activation and alert tissue and other immune cells.

# Results

## naRNA is a common component of NETs

To first investigate whether naRNA is a common component of NETs, we compared NETs released by human PMNs in response to

well-defined molecular agonists namely, phorbol myristate acetate (PMA), different complexes of LL37 with purified RNA (synthetic, as well as total fungal and bacterial RNA from *Staphylococcus aureus* and *Candida albicans*, respectively), nigericin and the live pathogen *C. albicans*. Confocal microscopy analysis using the well-characterized ribosomal RNA (rRNA)-specific antibody, Y10b (Lerner et al, 1981), revealed the presence of RNA in all corresponding NETs, independent of the stimulus (Fig. 1A, quantified in Fig. 1B; control of antibody staining in Fig. EV1A,B). The RNA signal was readily detectable along the web-like DNA threads in human NETs. Similar results were obtained in NETs released by murine bone marrow-derived PMNs (BM-PMNs, Fig. EV1C). To avoid the use of staining reagents, we also metabolically labeled primary human hematopoietic stem cells (HSCs) with 5-ethynyluridine (5-EU), a nucleotide that can be incorporated in cellular RNA but not DNA, and that is amenable to subsequent labeling by click chemistry (Presolski et al, 2011). These HSCs were then differentiated to neutrophils (Sioud, 2015) and the differentiation validated by microscopy and flow cytometric analysis (Fig. EV1E,F). NETs released from HSC-derived PMNs could be fluorescently labeled with click reagents only when grown in 5-EU-containing medium (Fig. EV1D). Moreover, the overlap between click-label and rRNA confirmed the specificity of the staining with anti-rRNA antibodies and indicated that naRNA contained rRNA but also other types of cellular RNAs. This experiment also unequivocally confirmed that upon PMN stimulation, cellular RNA is turned into a component of the NET, wherein it decorates the fine, web-like DNA structures. The localization of naRNA on DNA strands was further confirmed by high-resolution fluorescence microscopy and 3D analysis (AiryScan, Fig. EV1G and Movie EV1) as well as scanning electron microscopy (SEM, Fig. 1C; controls Fig. EV1H). Extraction of naRNA from PMA-induced NETs (Fig. EV1I) and subsequent RNA-sequencing analysis confirmed that purified naRNA contained multiple RNA types, with noticeably more non-ribosomal RNA than the corresponding total cellular RNA of PMNs (Fig. 1D). Collectively, these results indicate that naRNA is a canonical component of NETs released upon multiple stimuli from both human and murine neutrophils.

## naRNA is a potent 'composite' DAMP propagating NET formation in primary PMNs

The first proposed function of NETs in host defense was trapping and killing bacteria (Brinkmann et al, 2004). Therefore, we first explored if naRNA participated in this process. However, the antibacterial effect of NETs on live *Staphylococcus aureus* (evidenced by lower CFUs compared to resting PMNs) was similar to that of RNase-treated NETs, whereas DNase digestion reduced the drop in CFU discernibly but non-significantly (Fig. EV2A). Consequently, we turned our attention to a possible role for naRNA as a DAMP, since primary PMNs can respond to synthetic RNA-LL37 complexes with NET formation, and our earlier analysis indicated that NETs also contain LL37 (Herster et al, 2020). To prepare naRNA-containing stimulants, PMA-induced NETs were prepared as described in Methods and the original stimulus, PMA, removed by extensive washing before harvesting (Fig. EV2B; Appendix Fig. S1). Interestingly, these 'PMA NETs', when applied to naive PMNs from another donor, potently induced new NETs (Fig. EV2C, quantification Fig. 1E). Similar results were obtained with NETs generated using a lower PMA concentration or the ionophore A23187 (Appendix Fig. S1). Treatment with an RNase

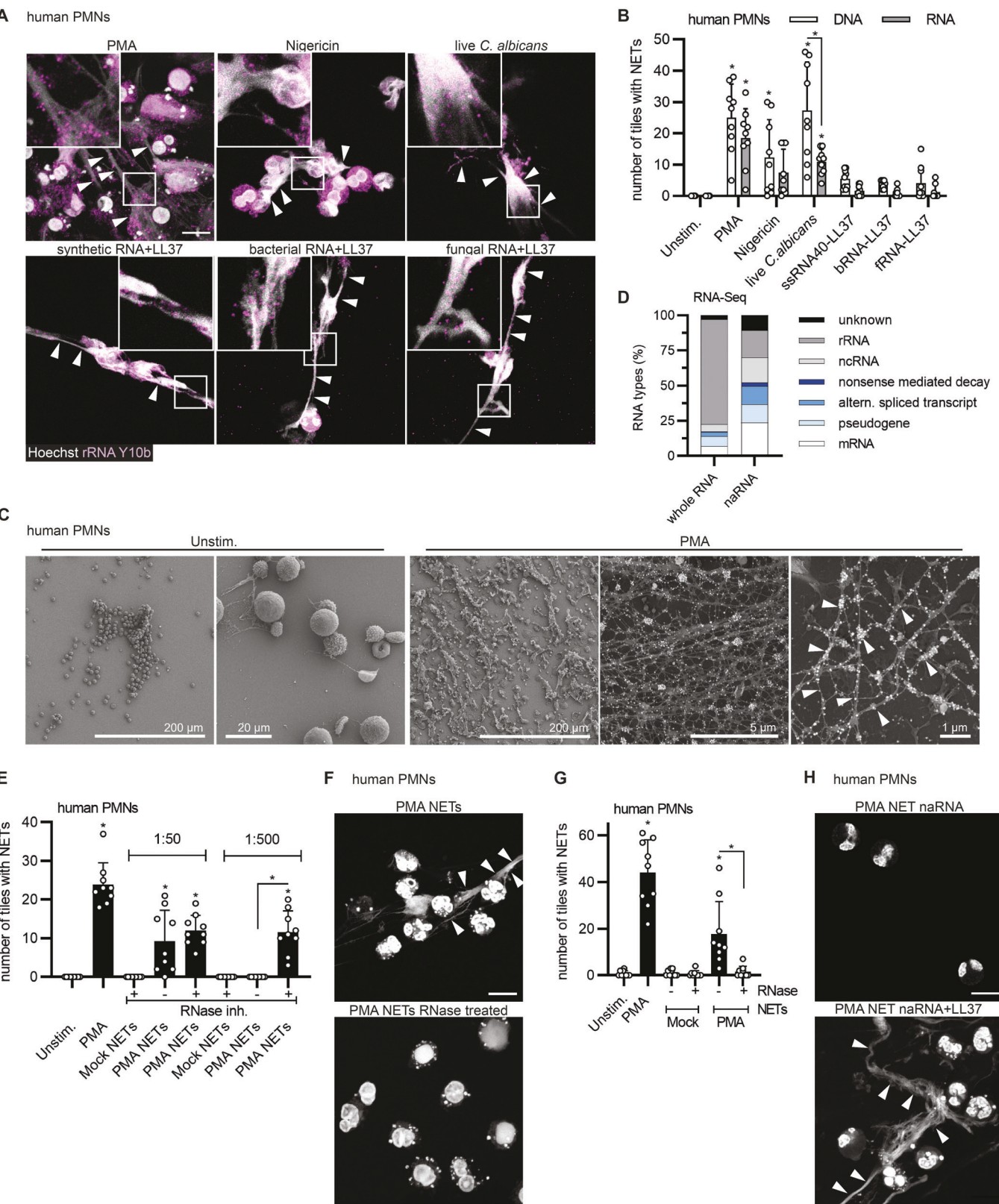

◄ **Figure 1.  naRNA is a canonical component of NETs.**

(A, B) Confocal microscopy of primary human PMNs stimulated as indicated for 3 h and stained for naRNA (anti-rRNA Y10b, magenta) and DNA (Hoechst 33342, white, $n = 3$ biological replicates, scale bar: 10 μm, arrowheads point to selected NET strands; representative images in (A) were quantified in (B) (each dot represents the number of NET-positive tiles in one image quantified from three images/condition). (C) Scanning electron microscopy of human primary PMNs treated as indicated and using anti-rRNA primary and immunogold (white arrow)-labeled secondary antibodies and silver enhancement ($n = 3$ biological replicates, representative images, scale bars as indicated; the two rightmost images show composite images with signals from secondary electron and backscattered electron detectors for topography and additional material information, respectively). (D) RNAseq of PMA NET naRNA ($n = 4$ biological replicates) and whole PMN RNA ($n = 1$ biological replicate, combined data). (E) Quantification of confocal microscopy of primary human PMNs, which were stimulated with NET content (harvested with/without RNase inhibitor and diluted 1:50 or 1:500), and then stained for NETs/DNA using DNA (Hoechst 33342) signal to quantify NET formation ($n = 3$ biological replicates, combined data, each dot represents the number of NET-positive tiles in one image quantified from three images/condition). (F) as in (E) but with/without pre-digestion of NET content with RNase A ($n = 3$ biological replicates, representative images, scale bar: 10 μm). (G) Quantification of (F) ($n = 3$ biological replicates, combined data, each dot represents the number of NET-positive tiles in one image quantified from three images/condition). (H) As in (E) but using purified naRNA (cf. Fig. EV1I) alone or in complex with exogenously added LL37 ($n = 3$ biological replicates, representative images, scale bar: 10 μm). Data information: In (B), (E), and (G), data are presented as mean + SD. *$p < 0.05$ according to one-way ANOVA. Data shown in (D) have been deposited in the NCBI Gene Expression Omnibus under accession number GSE253440. Source data are available online for this figure.

---

inhibitor enhanced the stimulatory effect of PMA NETs rendering a 1:500 dilution of PMA-NETs more effective than a 1:50 dilution of non-treated PMA-NETs (Figs. 1E and EV2C). Interestingly, similar NET preparations from mock-treated PMNs (here referred to as 'mock NETs') did not stimulate NET formation under the same experimental conditions. To further confirm the pivotal role of naRNA in the self-amplificatory mechanism of NETs, PMA NETs were pre-treated with RNase A to create naRNA-free NETs. As shown by immunofluorescence (IF) microscopy, RNase treatment strongly reduced NET propagation (Fig. 1F, controls Fig. EV2D, quantified in Fig. 1G). The degradation of naRNA in NETs was further confirmed by rapid loss of the RNA signal (using the RNA-selective dye, SYTO RNAselect (Herster et al, 2020)) in time-lapse digestion analysis (Movie EV2). Bearing in mind that PMNs do not respond to DNA or DNA-LL37 complexes (Herster et al, 2020), the opposing effects of RNase inhibitor and RNase treatments thus clearly indicated naRNA (and not DNA) to be the relevant immunostimulatory component for NETs to drive the activation of naive PMNs. In line with our previous work showing that synthetic RNA or LL37 alone cannot trigger PMN activation (Herster et al, 2020), purified 'naked' (i.e., stripped of DNA or proteins) naRNA isolated from PMA NETs (cf. Fig. EV1I) was unable to activate NET formation; however, re-complexing with exogenous LL37 was sufficient to restore NET formation (Fig. 1H, controls in Fig. EV2E). Therefore, on their own naRNA (cf. Fig. 1H) and LL37 (Herster et al, 2020) are not DAMPs but, in the context of NETs, act in concert as a novel type of 'composite' DAMP.

## naRNA DAMP activity in neutrophils is dependent on RNA sensors TLR8/Tlr13 and NLRP3 inflammasome activation

To validate naRNA as an immunostimulatory RNA component from the receptor side, we turned to human embryonic kidney 293T (HEK293T) cells which do not respond to RNA, unless transfected with plasmids encoding the human single-strand RNA (ssRNA) sensors TLR7 or TLR8 (Colak et al, 2014). NF-κB reporter assays revealed that TLR7 or TLR8- (both ssRNA sensors) but not TLR9 (DNA sensor)-transfected HEK293T cells stimulated with PMA NETs or mock NETs showed robust NF-κB activity only in response to PMA NETs (Fig. 2A). R848 (TLR7 and TLR8 agonist) and TL8 or ssRNA+DOTAP (both TLR8 agonists) served as controls. Of note, compared to the cognate agonist CpG, PMA NETs poorly activated

the DNA sensor TLR9, indicating that naRNA is a superior immune stimulant compared to NET DNA in this system. Accordingly, the observed NET response of primary PMNs (Fig. 2B, quantified in Fig. 2C) could be completely blocked by the TLR8-specific inhibitor, CU-CPT9a (Zhang et al, 2018), similar to the effects of Cl-amidine, a widely used inhibitor of multiple peptidyl arginine deiminases (PADs) (Knuckley et al, 2010) and some NET pathways (Kenny et al, 2017; Warnatsch et al, 2015), which also blocked the response to PMA (Figs. 2C and EV3A). This revealed the RNA sensor (Heil et al, 2004) to be the specific naRNA receptor in primary human PMNs. We further genetically validated the involvement of RNA sensing in naRNA-mediated NET propagation using bone marrow (BM)-PMNs from Unc93b1- or Tlr13-defective mice. Whereas in Unc93b1-defective mice signaling of all endosomal TLRs is abrogated (Tabeta et al, 2006), mice lacking Tlr13 (Eigenbrod and Dalpke, 2015), specifically lack ssRNA sensing (Li and Chen, 2012). Confocal microscopy analysis showed that wild type (WT) BM-PMNs responded readily to both PMA (used as control) and PMA NETs, whereas Unc93b1 (Fig. EV3B) or Tlr13 knock-out (KO) PMNs (Fig. 2D, quantified in E) only responded to the TLR-independent PMA control but not PMA NETs. In addition, naive BM-PMNs from Tlr9 KO mice, which lack DNA sensing, responded to both PMA and PMA NETs with NET formation just like WT BM-PMNs (Fig. EV3C), emphasizing the redundancy of DNA and the key role of naRNA in the self-propagating properties of NETs. Having unequivocally confirmed naRNA to be a primary immune stimulant in NETs, we sought to establish the downstream signaling pathway, and we speculated naRNA might trigger a novel neutrophil TLR8-NLRP3 pathway, based on similarities to monocytes (Weber et al, 2020; Munzer et al, 2021; Vierbuchen et al, 2017). Indeed, the selective NLRP3 inhibitor, MCC950/CRID3 (Coll et al, 2015), significantly blocked NET release by naive human primary PMN in response to PMA NETs and nigericin, but not PMA (Fig. EV3D, quantified in Fig. 2F). Furthermore, similar to synthetic RNA+LL37, PMA NETs induced low but priming-independent release of interleukin (IL)-1β in naive PMN, which could be antagonized by treatment with the TLR8-inhibitor CU-CPT9a, the NLRP3 inhibitor MCC950 or the gasdermin D (GSDMD)-inhibitor disulfiram (Fig. 2G). Newly released human NETs in response to PMA NETs additionally showed condensed 'specks' of the NLRP3 adaptor ASC (Fig. 2H) (Gaidt et al, 2016). Our study therefore establishes a novel TLR8-NLRP3-ASC-caspase-1-GSDMD signaling pathway to NET formation in response to extracellular RNA, including naRNA, in neutrophils.

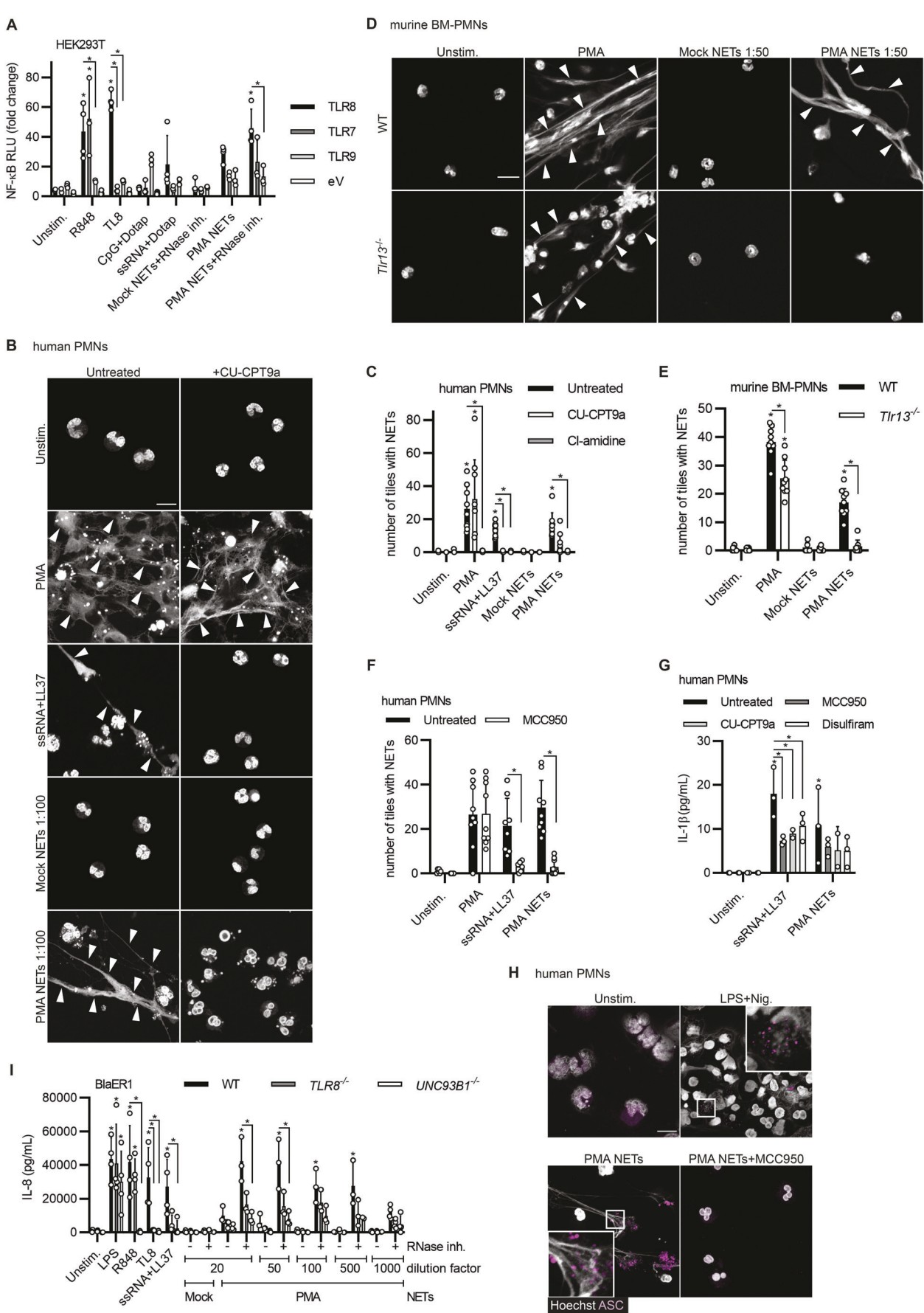

**Figure 2. naRNA drives TLR8 and NLRP3-dependent NET propagation in human PMN and TLR8-dependent macrophage activation.**

(A) NF-κB dual luciferase assay in HEK293T cell transiently transfected and stimulated as indicated (eV = empty vector, $n$ = 3–5 biological replicates, combined data). (B) Confocal microscopy of human primary PMNs, which were stimulated as indicated in the presence or absence of the TLR8-inhibitor CU-CPT9a (100 nM) and stained for NETs/DNA (Hoechst 33342, white, $n$ = 3 biological replicates, representative images, scale bar: 10 μm, white arrows indicate NETs). (C) Quantification of (B) using DNA (Hoechst 33342) signal to quantify NET formation ($n$ = 3 biological replicates, combined data, each dot represents the number of NET-positive tiles in one image, three images/condition). (D) Confocal microscopy analysis of primary C57BL/6 WT or $Tlr13^{-/-}$ murine BM-PMNs stimulated as indicated and stained for NETs/DNA (Hoechst 33342, white, $n$ = 3 biological replicates, representative images, scale bar: 10 μm, white arrows indicate NETs). (E) Quantification of (D) as in (C) ($n$ = 3 biological replicates, combined data). (F) Quantification of confocal microscopy as in (B, C) but using the NLRP3-inhibitor MCC950 (10 μM; $n$ = 3 biological replicates, combined data, each dot represents the number of NET-positive tiles in one image, three images/condition,). (G) Levels of IL-1β, as measured by triplicate ELISA from human PMN stimulated as in (F) in the presence or absence of MCC950 (10 μM), CU-CPT9a (100 nM) or disulfiram (25 μM; $n$ = 3 biological replicates, combined data). (H) Confocal microscopy of primary human PMNs stimulated for 3 h as indicated (PMA NETs, 1:50 dilution) or primed for 3 h with LPS (10 ng/mL), subsequently stimulated for 2 h (nigericin, 5 μM) in the presence or absence of MCC950 (10 μM) and stained for ASC (magenta) and DNA (Hoechst 33342, white, $n$ = 3 biological replicates, scale bar: 10 μm, representative images). (I) Levels of IL-8, as measured by ELISA, from WT, $TLR8^{-/-}$ and $UNC93B1^{-/-}$ BlaER1 macrophage-like cells stimulated as indicated for 18 h ($n$ = 3–4 biological replicates, combined data, each dot represents one biological replicate). Data information: In (A), (C), (E–G), and (I), data are presented as mean + SD. In (A), (E–G), and (I), *$p$ < 0.05 according to one-way ANOVA. In (C), *$p$ < 0.05 according to non-parametric one-way ANOVA. Please note that selected panels in (B), (D), and (H) also appear in Fig. EV3A, B, and D, respectively, as these two experiments were carried out simultaneously or were part of the same experiment, and hence control conditions (e.g., unstimulated) are identical. Source data are available online for this figure.

## PMN-derived naRNA triggers broader immune cell activation

Given the potent effect of naRNA on naive PMNs (*cf.* Fig. 2B–H) and the ability of human macrophages to sense RNA via TLR8 (Ishii et al, 2014), we hypothesized that naRNA might also directly activate macrophages, which are rapidly recruited at the sites of PMN activation in peripheral tissues (Mahdavian Delavary et al, 2011), contributing to an inflammatory process in vivo. We thus assessed the effects of naRNA on genetically modified macrophage-like cell lines: BlaER1 macrophages (Vierbuchen et al, 2017) responded to PMA NETs but not mock NETs with IL-8 release, and this was TLR8-dependent as evidenced by comparisons of WT and *TLR8*-edited BlaER1 cells (Fig. 2I). Use of a naRNA-stabilizing RNase inhibitor during NET preparation (*cf.* Fig. EV2B) drastically increased the potency of PMA NETs on these macrophages. In addition, TLR7 and TLR8-edited THP-1 macrophage-like cells (Coch et al, 2019) responded to PMA NETs with relatively lower release of IL-8 than WT THP-1 cells (Fig. EV4A), confirming RNA as the active agent in monocyte/macrophage activation. Human peripheral blood mononuclear cells (PBMCs) were also stimulated with different NET preparations. RNA-stabilized PMA NETs consistently elicited higher levels of TNF, IL-6 and IL-8 than non-stabilized PMA- or mock NETs (Fig. EV4B–D). However, this effect was not sensitive to the TLR8 inhibitor Cu-CPT9a, which would be consistent with the ability of TLR7 to also sense naRNA (*cf.* Fig. 2A) and possibly due to a mixture of cells being present. Further analysis revealed that the natural killer (NK) cell line NK-92 MI showed low release of interferon γ (IFN γ) in response to RNA-stabilized PMA NETs (Fig. EV4E). Collectively, our results show that the RNA component of NETs, naRNA, can engage not only bystander neutrophils in a feed-forward inflammatory response but also other myeloid (monocytes/macrophages) and lymphoid (NK cells) innate immune cells.

## naRNA activates psoriasis-associated gene transcription in primary keratinocytes

To investigate whether non-hematopoietic cells with immune functions, e.g., tissue cells like keratinocytes, could also respond to RNA-stabilized PMA NETs, N/TERT-1 keratinocytes (Dickson et al, 2000) monolayer cultures were exposed to PMA-NETs (Fig. EV4F). Similar to exposure to synthetic RNA+LL37, this triggered a dose-dependent release of IL-8, which was increased by RNA stabilization using RNase inhibitors, indicating a dependence on RNA. Similar results were obtained for primary normal human epidermal keratinocytes (NHEK) from healthy donors (Fig. 3A). qPCR analysis showed that the psoriasis-associated genes *IL17C*, *LL37*, and *IL36G* (Fig. 3B,C,D, respectively) were induced upon stimulation with PMA NETs. However, pre-digestion of NETs with RNase A abolished the stimulatory effects of NETs on keratinocytes, whereas pre-digestion with DNase I did not. Analyses in primary murine keratinocytes confirmed the findings made for human cells, namely a NET-dependent upregulation of the genes *Il17C, Cxcl10* (Fig. 3E,F, respectively), *Cxcl1, Cxcl2*, and *Cxcl5* (Fig. EV4G,H,I, respectively). Of note, in an 'in vivo'-like human skin equivalent 3D model in which natural keratinocyte differentiation was recapitulated by NHEK cells (Bitschar et al, 2019), PMA but not mock NETs significantly induced *IL8* mRNA and protein (Figs. 3G and EV4J). To investigate which RNA receptor-mediated keratinocyte responses to naRNA, we first screened and ruled out the main RNA pattern recognition receptors (PRRs), e.g., TLRs (adaptor MYD88 and TRIF siRNA knockdown Fig. EV4K,L, respectively), RIG-I/Mda (adaptor MAVS knockdown Fig. EV4M) and NLRP1 (siRNA knockdown Fig. EV4N,O). Since the PRR NOD2 was reported as an RNA sensor in non-immune, lung epithelial cells (Sabbah et al, 2009), we interrogated its involvement in keratinocyte naRNA responsiveness using NOD2 siRNA knockdown (Fig. EV4P). Under these conditions, we observed a significant decrease in the release of IL-8 upon treatment of NHEK with MDP, a typical NOD2 agonist, synthetic RNA and PMA NETs, but not with the TLR2 stimulus Pam$_3$CSK$_4$ (Fig. 3H). Similar results were obtained in NHEKs treated with GSK583, an inhibitor of the NOD2 pathway kinase RIPK2 (Haile et al, 2016) (Fig. 3I). To confirm the involvement of NOD2 in keratinocyte RNA sensing, tail keratinocytes from WT and $Nod2^{-/-}$ mice were isolated, cultured and compared for their response to RNA. As expected, *Nod2*-deficient keratinocytes showed a strongly blunted RNA response as indicated by significantly reduced induction of *IL17C*, *IL36G*, *Cxcl2*, and *Cxcl10* mRNA (Fig. 3J–M, respectively). In addition, pre-treatment of keratinocytes with synthetic RNA significantly reduced *S. aureus* colonization (Fig. EV4Q), indicating a protective effect of ribonucleic acids on those immune-like skin cells. Collectively, naRNA can not only activate primary

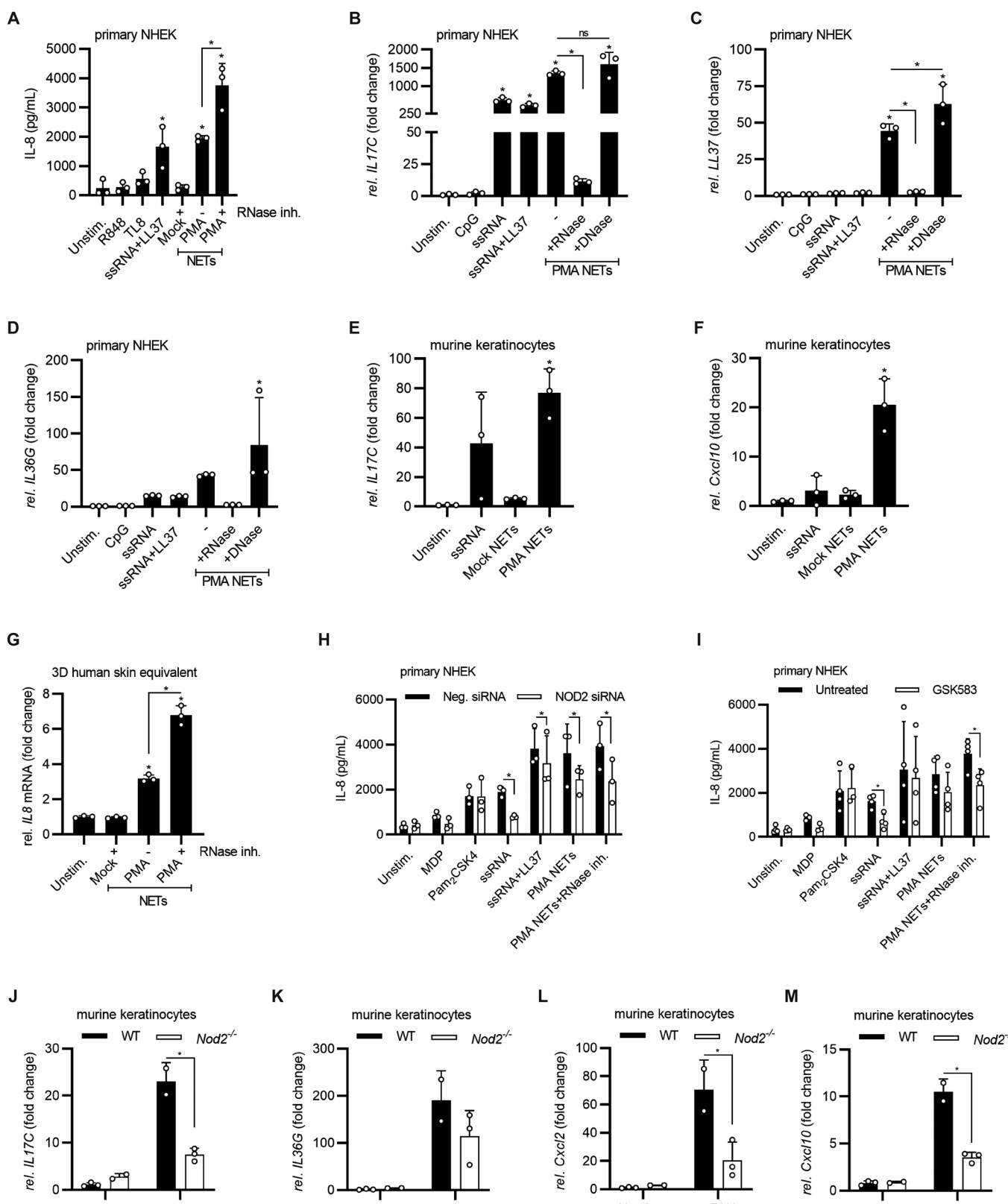

**Figure 3.  NETs induce naRNA and NOD2-dependent activation of keratinocytes.**

(A) Levels of IL-8, as measured by triplicate ELISA upon release by primary normal human epidermal keratinocytes (NHEK) stimulated as indicated for 24 h ($n = 3$ biological replicates, combined data, each dot represents one biological replicate). (B–D) Fold changes in the expression of the (B) *IL17C*, (C) *LL37*, or (D) *IL36G* in NHEK stimulated as indicated for 24 h. qPCR was performed in triplicate and fold changes calculated relative to unstimulated condition ($n = 3$ biological replicates, representative of one biological replicate is shown, each dot represents one technical replicate). (E, F) qPCR as in (B) but for *IL17C* (E) and *Cxcl10* (F) in murine C57BL/6 WT keratinocytes stimulated for 1 h ($n = 3$ biological replicates, combined data, each dot represents one biological replicate). (G) Triplicate qPCR as in B but for *IL8* in NHEK 3D human skin equivalent constructs stimulated as indicated for 24 h ($n = 3$ biological replicates, representative of one biological replicate is shown, each dot represents one technical replicate). (H) As in A but with or without NOD2 siRNA (3.5 nM; $n = 3$ biological replicates, combined data, each dot represents one biological replicate). (I) As in (H) but in the presence or absence of the RIPK2 inhibitor GSK583 (1 µM; $n = 3$–4 biological replicates, combined data, each dot represents one biological replicate). (J–M) Triplicate qPCR as in (B) but for *IL17C* (J), *IL36G* (K), *Cxcl2* (L), or *Cxcl10* (M) in murine C57BL/6 WT or *Nod2* KO keratinocytes stimulated as indicated for 1 h ($n = 3$ biological replicates, combined data, each dot represents one biological replicate). Data information: In (A–M), data are presented as mean + SD. In (A–G) and (J–M), $*p < 0.05$ according to one-way ANOVA. In (H) and (I), $*p < 0.05$ according to two-way ANOVA. Source data are available online for this figure.

neutrophils in vitro but also trigger broad immune activation in other immune and tissue cells/equivalents, especially keratinocytes, in an RNA sensor-dependent manner.

## naRNA displays DAMP activity in vivo in dependence on RNA sensing

To gain an insight into whether naRNA could trigger inflammation in vivo, we intradermally injected RNase inhibitor-stabilized mock or PMA NETs in the ears of C57BL6 mice. RNA-stabilized PMA NETs were almost as potent to induce ear swelling as synthetic RNA+LL37 (Fig. 4A), but even non-stabilized PMA NETs induced a response significantly higher than mock NETs. RNA-stabilized PMA-NETs were also injected into the ears of *LysM*^EGFP/+ mice, in which myeloid cells are GFP-positive, thus enabling in vivo monitoring of cellular influx during skin inflammation. Here, RNA-stabilized PMA NETs elicited even greater cellular influx than synthetic RNA+LL37 and mock NETs (Fig. 4B). Importantly, a comparison of WT and *Tlr13*-deficient animals showed that the ear swelling reaction was naRNA-dependent because RNA-stabilized PMA-NETs were significantly less effective in *Tlr13*-deficient animals (Fig. 4C). Although non-stabilized RNA was not assessed in *LysM*^EGFP/+ and *Tlr13* KO mice, we concluded that RNA contained in PMA NETs can elicit skin inflammation (*cf.* Fig. 4A), especially when stabilized during the NET preparation step (*cf.* Fig. 4B,C).

## RNA recognition contributes to progressive skin inflammation in experimental psoriasis

Finally, we sought to explore if sensing of naRNA via Tlr13 was relevant in an animal model of human disease. The most common murine model of psoriasis uses the TLR7 ligand imiquimod (IMQ) to trigger increasing skin inflammation, which is characterized by epidermal thickening and immune cell infiltration (Gilliet et al, 2004). We previously showed that IMQ treatment also leads to the occurrence of NETs in the tissue (Herster et al, 2020). We therefore speculated that naRNA might be involved in disease progression, a hypothesis that we tested by comparing naRNA sensing in *Tlr13* deficient and WT animals. Although in the early induction phase, WT and *Tlr13* KO animals showed a similar increase in ear thickness, the groups diverged from day 3 onward, after which *Tlr13*-deficient animals were significantly protected from skin inflammation compared to WT mice (Fig. 4D). In addition, the characteristic epidermal thickening upon IMQ treatment was significantly lower in *Tlr13*-deficient mice compared to WT mice

(Fig. 4E, representative image Fig. 4F). Immunofluorescence analysis at the end of the experiment showed lower PMN infiltration and NET formation in WT vs *Tlr13* KO animals as indicated by lower citrullinated histone H3 (citH3) staining (Fig. 4G, quantified in Fig. 4H), a marker for NETs in tissue (Suzuki et al, 2022). A plausible explanation is that IMQ-initiated NET formation (Herster et al, 2020) amplifies skin inflammation in WT animals via NET (and hence naRNA) release, whereas this is reduced in *Tlr13* KO animals. These data indicate that naRNA might significantly contribute to inflammation in this well-established psoriasis disease model by acting as an inflammation-amplifying endogenous DAMP.

## naRNA and LL37 await externalization in an already pre-associated state in healthy donor neutrophils

Thus far, our data align well with the notion that LL37 enables the tolerance to self-RNA to be broken by association with RNA in the assembly of the NET, thereby forming a composite DAMP. Indeed, in confocal microscopy (Fig. 5A), Pearson's co-localization analysis (Fig. 5B) and additional line plot quantification (Figs. 5C and EV5B; antibody controls Fig. EV5A) of PMA-stimulated PMN NETs, RNA and LL37 did in fact co-localize in NETs, i.e., after extrusion. However, most surprisingly, in unstimulated PMNs naRNA and LL37 showed even greater Pearson's co-localization (Fig. 5B) and, unlike DNA, shared co-localization maxima (Figs. 5C and EV5B), suggesting the intriguing notion of pre-association of RNA and LL37 in the same cellular compartments even before extrusion. Even clearer results were observed when the rRNA Y10b antibody was substituted by the antibody-independent RNA dye, SYTO RNAselect. Counterstaining with LL37 antibodies and respective line plot analysis (Figs. 5D and EV5C,D) as well as orthogonal projection (Fig. 5E) and 3D reconstruction of extensive z-stacks (Fig. EV5E) confirmed the co-localization of neutrophil RNA with LL37 in granular structures of resting PMN. To verify this further, rRNA and LL37 double staining of resting PMN samples was performed on ultrathin (50–60 nm) sections (Fig. 5F), in which signals are collected from far smaller volumes than conventional IF. This confirmed the reported localization of LL37 to neutrophil granules as a granule protein (Sorensen et al, 1997), but also the unexpected concomitant presence of rRNA in the same structures. Transmission electron microscopy combined with double staining of rRNA and LL37 using two different immunogold sizes corroborated this further at even higher magnification (Fig. 5G). Although physical pre-association of individual rRNA and LL37 is

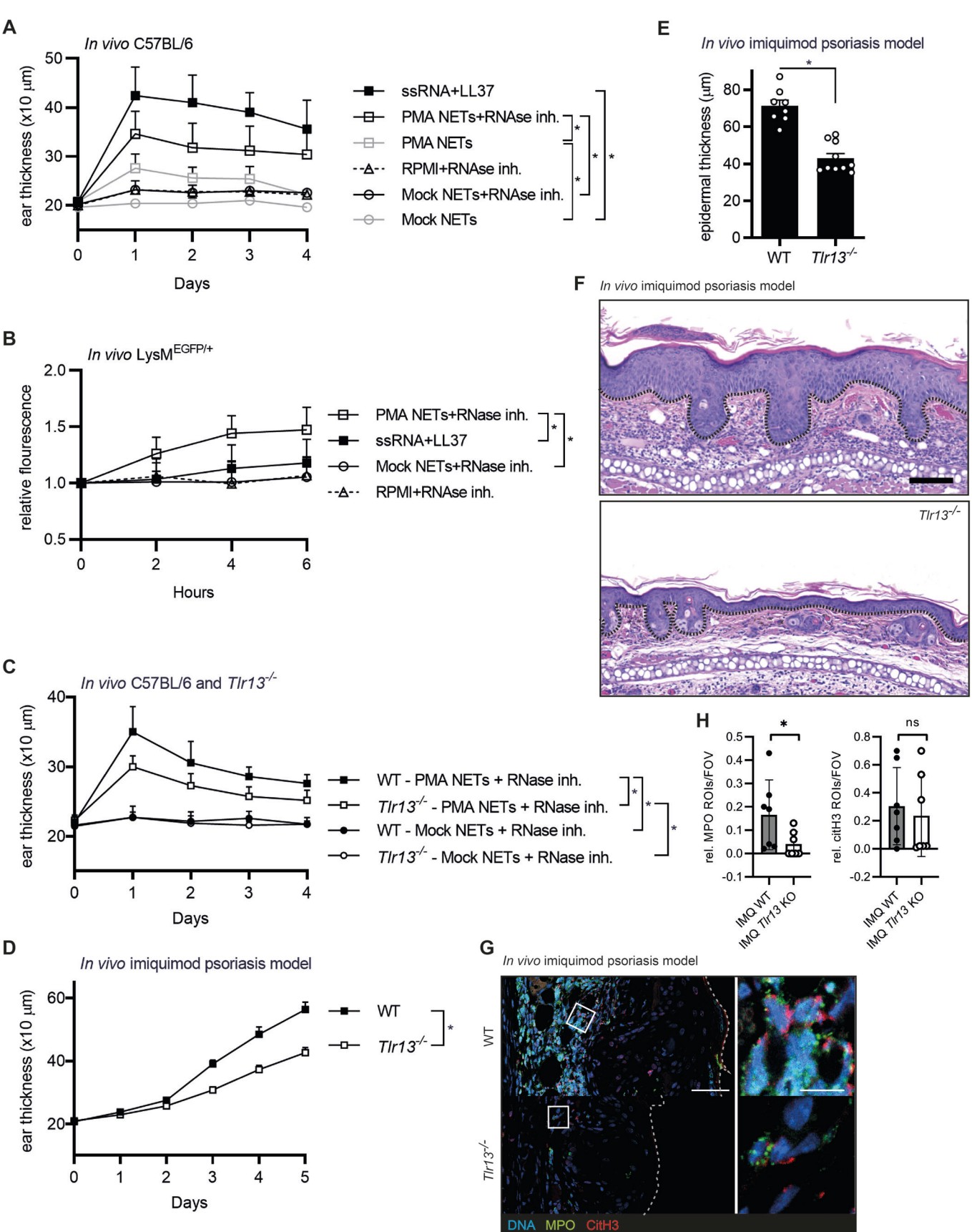

**Figure 4. naRNA is a driver of NET-associated in vivo inflammation.**

(A) Ear thickness quantified daily in WT C57BL6 mice injected intradermally on day 0 as indicated (*n* = 5 biological replicates per group, combined data from 3 experiments). (B) Fluorescence imaging monitored hourly in *LysM*$^{EGFP/+}$ mice injected intradermally at *t* = 0 as indicated (*n* = 10 biological replicates per group, combined data from 2 experiments). (C) as in (A) but also using *Tlr13*$^{-/-}$ mice (*n* = 7 biological replicates each, combined data from 2 experiments). (D) as in (C) but instead of intradermal injection, topical imiquimod application was performed on day 0–4 (C57BL/6 *n* = 7, *Tlr13*$^{-/-}$ *n* = 7 biological replicates, combined data from 2 experiments). (E) Measurement of epidermal thickness of Hematoxylin and Eosin (H&E)-stained ear skin specimens of (D) (C57BL/6 *n* = 8 biological replicates, *Tlr13*$^{-/-}$ *n* = 10 biological replicates, combined data from 2 experiments). (F) Histological analysis of H&E-stained ear skin specimens of (D) (d5 samples, C57BL/6 *n* = 8 biological replicates, *Tlr13*$^{-/-}$ *n* = 10 biological replicates, representative of one biological replicate is shown; dashed line delineates epidermal-dermal border; scale bar: 100 μm). (G) Immunofluorescence analysis of (E/F) using Hoechst (DNA, blue), anti-MPO (PMN, green) and anti-citH3 (NETs, red) staining (day 5 samples, C57BL/6 *n* = 7 biological replicates, *Tlr13*$^{-/-}$ *n* = 7 biological replicates, one biological replicate is shown; dashed line delineates epidermal edge; scale bar: 10 μm in close-up or 50 μm). (H) Quantification of (G) (C57BL/6 *n* = 7 biological replicates, *Tlr13*$^{-/-}$ *n* = 7 biological replicates, combined data from 2 experiments). Data information: In (A–E) and (H), data are presented as mean + SD. In (A–D), *$p < 0.05$ according to two-way ANOVA. In (E) and (H), *$p < 0.05$ according to Student's t-test. Source data are available online for this figure.

difficult to demonstrate in primary neutrophils, the concept of pre-packaging of RNA and LL37 in the same specific compartments suggests a likely interaction and fits with the RNAseq data showing that certain RNA types are lost upon release (*cf.* Fig. 1D), probably because they are not bound to the NET by LL37. Of note, the effect was seen in normal peripheral blood PMN across multiple healthy donors, indicating that LL37 association with RNA is a normal physiological property of PMNs. In turn, and in line with our initial analysis (*cf.* Figs. 1A,B and EV1C), this would mean that every NET naturally contains and externalizes composite naRNA-LL37 DAMPs. This DAMP activity may be strictly required in addition to the aforementioned antimicrobial effects of NETs to mount a sufficient antimicrobial host response. We speculated the short-lived nature of RNA as a biomolecule might be an inherent and self-restricted limiter of DAMP activity in healthy subjects to avoid excessive sterile inflammation. To test whether freshly extruded NETs would have higher DAMP activity than older NETs, we generated PMA NETs, harvested, and transferred them to naive neutrophils or NHEKs immediately (as 'fresh NETs') or after additional 4 h incubation with either freshly added PMN medium or pooled human serum, respectively, to generate 'old NETs' (NET degradation shown in Fig. EV5F). Interestingly, fresh NETs readily induced further NET release from naive PMN (Fig. 5H) and IL-8 release from NHEKs (Fig. 5I), whereas old NETs did so to a lesser extent. However, the latter response was partially rescued when an RNase inhibitor was used to protect naRNA from degradation (Fig. 5I), although some loss of naRNA due to disintegration of the DNA scaffold cannot be excluded. Thus, the release of naRNA could be considered as a DAMP release to elicit a pro-inflammatory response, with potential self-limiting properties to avoid overactivation.

## Discussion

At first sight, it may not seem surprising that the release of NETs, a process that churns up the most critical compartment of a cell, e.g., the nucleus, also inevitably leads to the release of another primary cellular biomolecule, namely RNA. However, independently of whether the process is regulated or not, the release of RNA by NET-forming PMNs appears to have physiological importance as we have demonstrated here. Rather than acting directly in antimicrobial defense (*cf.* Fig. EV2A), naRNA appears to be an LL37-associated DAMP of PMN origin that can be released in the NETting process, and then activates both PMN and other immune

and tissue cells in an RNA sensor-dependent manner. Our conclusions largely focus on in vitro observations using primary human cells. As in vivo responses to NETs prepared without RNase inhibitor are relatively small, further work will be required to delineate the complex interplay between naRNA, endogenous RNases and endogenous RNase inhibitors more fully (Abtin et al, 2009; Probst et al, 2006). Nevertheless, several findings warrant further discussion:

Firstly, our data indirectly challenge previous concepts of self-RNA and LL37 in the 'breaking' of innate immune tolerance in psoriasis: according to previous work, LL37 may 'break' immune tolerance to self-RNA by acting as a physiological 'transfection reagent' of RNA for uptake into immune cells (Ganguly et al, 2009), and by shielding it from RNase degradation (Neumann et al, 2014). However, our data show that RNA and LL37 do not only aggregate and co-localize 'accidentally' in the mesh of the NET. Quite unexpectedly, they were 'pre-associated' in the same intracellular storage compartments during NET formation. The process for pre-association and the precise nature of the RNAs assembled with LL37 await elucidation, but the identification of such a pre-association argues that LL37-RNA complexes are not only 'accidentally assembled' tolerance breakers in certain diseases (Ganguly et al, 2009). Rather, it suggests naRNA-LL37 complexes normally act as naturally pre-associated DAMPs that arm neutrophils at steady state and are thus inherently configured to 'break tolerance' and cause immune-activation in healthy donors—with additional patho-mechanism potentially contributing to establish chronic disease in e.g., psoriasis patients. Moreover, since the here-described phenomena were found in neutrophils and other cells from healthy donors, pre-association (rather than accidental extracellular complexation) points to a very specific but broadly relevant novel role of naRNA-LL37 DAMPs in NET biology in general: In vivo, naRNA-LL37 DAMP activity may be essential to augment the antimicrobial properties of NETs or cells like keratinocytes (*cf.* Fig. EV4Q) and entail the concomitant risk of sterile inflammation as an unavoidable trade-off. On the other hand, inflammation is essential for sterile insult removal, too, and the fact that NETs have been described for a multitude of sterile conditions indicates that antimicrobial activity is only one of several functions of NETs. Our findings support the notion that the NET response in the absence of infection is inevitably a DAMP response, an important new facet in the concepts in NET biology. We speculate that the physiological relevance of deploying this composite DAMP upon NET formation could be to tag fresh NETs with a timed molecular beacon: In this scenario, freshly extruded

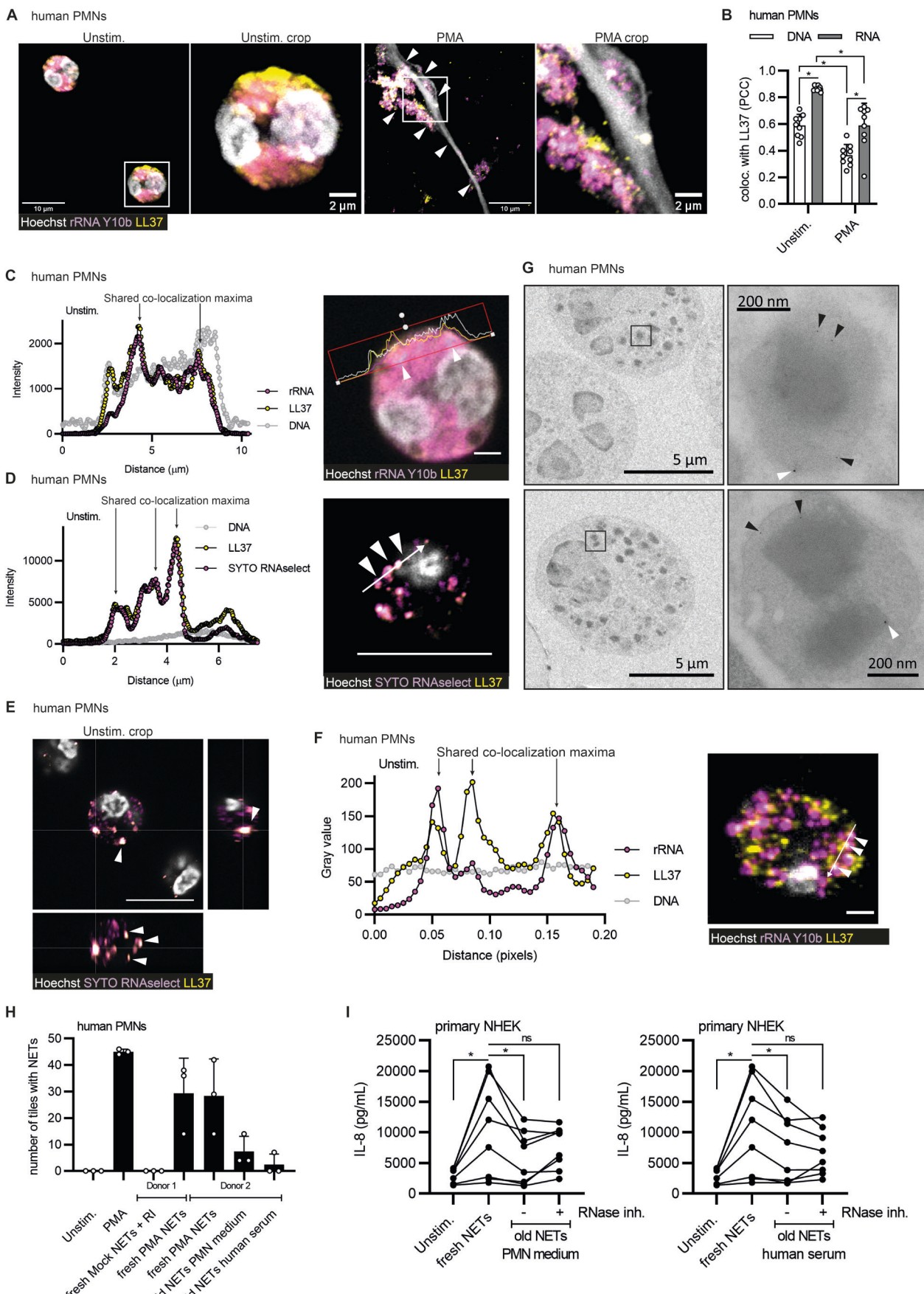

**Figure 5.  naRNA and LL37 are pre-associated in resting neutrophils.**

(A) Confocal microscopy of primary human PMNs stimulated as indicated for 3 h and stained for naRNA (anti-rRNA Y10b, magenta), LL37 (anti-hLL37-DyLight550, yellow), and DNA (Hoechst 33342, white, n = 3 biological replicates, representative images, scale bar 2 or 10 µm as indicated). (B) Pearson's correlation coefficient (co-localization) analysis of (A) (n = 3 biological replicates, combined data, each dot represents one image, three images/condition). (C) Line plot analysis of LL37, RNA, and DNA staining of primary human PMNs stimulated as indicated in (A) was performed using ZenBlue3 software (n = 3 biological replicates, representative graph, scale bar 2 µm). White arrows indicate co-localization of RNA and LL37. (D) As C but staining with SYTO RNAselect instead of anti-rRNA (n = 3 biological replicates, representative graph, scale bar 10 µm). White arrows indicate co-localization of RNA and LL37. (E) As in (D) but showing x,z and y,z projections from multiple z-stacks. White arrows indicate co-localization of RNA and LL37. (F) As in (A/C) but on 50–60 nm ultrathin sections of unstimulated PMNs (n = 3 biological replicates, representative image, scale bar 2 µm). Line plot analysis was performed using ImageJ-Win64 software (n = 3 biological replicates, representative graph). White arrows indicate co-localization of RNA and LL37. (G) Transmission electron microscopy of unstimulated human primary PMNs using anti-rRNA and anti-hLL-37 primary and immunogold (6 nm (black arrow) and 12 nm (white arrow), respectively)-labeled secondary antibodies (n = 3 biological replicates, representative images, scale bars as indicated). (H) Quantification of confocal microscopy of primary human PMNs stimulated as indicated with fresh or old NETs generated by incubation in PMN culture medium or human serum treatment, respectively, stained for DNA (Hoechst 33342) and quantified as before (n = 2 biological replicates, each dot represents the number of NET-positive tiles in one image quantified from three images/condition). (I) Levels of IL-8, as measured by triplicate ELISA upon release from primary normal human epidermal keratinocytes (NHEK) stimulated for 24 h as indicated in (F) (n = 8 biological replicates, combined data, each dot represents one biological replicate). Data information: In (B), (H), and (I), data are presented as mean + SD. In (B) and (I), *p < 0.05 according to one-way ANOVA. In (H), *p < 0.05 according to Kruskal–Wallis test with Dunn's correction. Please note that selected panels in (C) and (E) also appear in Fig. EV5D as these two experiments were carried out simultaneously or were part of the same experiment, and hence control conditions (e.g., unstimulated) are identical. Source data are available online for this figure.

NETs decorated with the naRNA-LL37 DAMP would highlight an acute tissue insult (or the lingering presence of a pathogenic microorganism during non-sterile conditions) to other cells not directly or initially engaged by the threat. The naRNA-LL37 DAMP would thus label 'fresh NETs' as 'requiring attention' and trigger further immune activation, clearance and eventually inflammation resolving activities. Over time, whilst the DNA-related structural and antimicrobial properties of NETs may remain longer, inevitable RNA degradation would lead to deactivation of the composite DAMP, rendering 'old NETs' less immunostimulatory (cf. Fig. 5H,I). In analogy to the concept of bacterial RNA as a unique "vita-pattern-associated molecular pattern (PAMP)" (Sander et al, 2011), reporting specifically on a live (and hence dangerous) microbe, naRNA could thus be considered a "vita-DAMP" that reports on an ongoing NET process and inflammation. We provide preliminary evidence for this putative scenario, but further work will be required to explore more fully the intriguing notion of naRNA as a time-wise, self-restricting molecular vita-DAMP label for NETs in a sterile setting.

Our data also pertain to the origin and effects of 'extracellular RNA' (exRNA). exRNA is a generic term to indicate a hetero-geneous group of RNA molecules which are actively or passively released during sterile inflammation or infectious processes. exRNA can be released in a 'free' state, bound to proteins or phospholipids, in association with extracellular vesicles (EVs) or apoptotic bodies (Preissner et al, 2020). In all these forms, exRNA may function as a DAMP but also as an e.g., procoagulant or regenerative factor (Preissner et al, 2020). Our data identify naRNA as the first type of exRNA for which origin (namely NETs) and process of extra-cellular release in physiologically relevant amounts are clearly defined. We speculate that our findings will help trace multiple descriptions of exRNA on the one side, and to neutrophil traps on the other. For example, exRNA has emerged as disease-relevant molecule in atherosclerosis, where it was described to act as a pro-inflammatory mediator enhancing the recruitment of leukocytes to the site of atherosclerotic lesions as shown in a mouse model of accelerated atherosclerosis (Simsekyilmaz et al, 2014). At the same time, by priming macrophages NETs have been ascribed a role in amplifying sterile inflammatory responses in an independent mouse model of atherosclerosis (Warnatsch et al, 2015). However, never have these two independent strains of research been

connected. By showing NETs to release naRNA, a DAMP type of exRNA, our work connects both lines of enquiry. Likewise, for rheumatoid arthritis (RA), our work makes a plausible link between exRNA in synovial fluid contributing to joint inflammation (Neumann et al, 2018), the emerging role of NETs in RA (Song et al, 2020) and even a hitherto enigmatic but therapeutically relevant role of TLR8 (Sacre et al, 2016; Sacre et al, 2008).

In addition, NETs have been in the spotlight of research on COVID-19-associated thrombo-inflammatory states, like sepsis, thrombosis, and respiratory failure, have been found in the plasma of hospitalized COVID-19 patients, and linked to thrombus formation by several research groups (Zuo et al, 2020; Yaqinuddin and Kashir, 2020; Cicco et al, 2020; Barnes et al, 2020; Golonka et al, 2020). Furthermore, exRNAs have been in focus as procoagulant cofactors in blood coagulation (Kannemeier et al, 2007; Sahu et al, 2017). A link between NET-related and RNA-induced thrombus formation in COVID-19 has never been made but highlights naRNA as a potential contributing factor of NET-induced thrombo-inflammatory states in COVID-19 disease pathogenesis. Collectively, we suspect similar links between NETs and exRNA via naRNA to emerge for cardiovascular diseases and cancer, if the role of naRNA were to be thoroughly assessed. The broad sensitivity of immune and tissue cells to naRNA observed by us makes sense of how exRNA may act pathophysiologically.

Although we believe PMNs to be the primary trap forming population of human leukocytes and hence, naRNA the most significant "trap-associated RNA", it will be interesting to explore whether mast cell (MCETs) (Mollerherm et al, 2016) or macro-phage extracellular traps (METs) (Doster et al, 2018) contain RNA. However, unlike PMN, the latter immune cells are not primary sources of LL37 (Sorensen et al, 1997), so that RNA associated with MCETs or METs would be of lesser physiological relevance than naRNA as a DAMP due to a lack of LL37. Therefore, translational approaches, e.g., to restrict trap-RNA mediated amplification of inflammation, should probably center on PMN-derived naRNA. The use of PAD inhibitors has already been investigated in animal models to treat cancer or atherosclerotic lesions (Knight et al, 2014; Li et al, 2020) and would represent one potential way of eliminating NETs and hence naRNA-mediated inflammation. However, this would also prevent the effects of NETs that are beneficial for host

defense (e.g., physical trapping via DNA and antimicrobial enzymatic activities) and may render treated patients vulnerable to infections. From a translational perspective, our in vivo data indicate that blockade of RNA sensing might be more advantageous, restricting only naRNA-mediated responses. Although probably not applicable to patients, it is underscored by evidence that neuroinflammation upon subarachnoid hemorrhage, which is characterized by a NET pathology, is strongly ameliorated by RNase treatment in vivo (Fruh et al, 2021). More applicable to patients may be the targeting of TLR8 or NLRP3 by small molecular antagonists (Coll et al, 2015; Vlach et al, 2021) or inhibitory oligonucleotides, which are able to block PMN responsiveness to naRNA. Exciting is thus the observed efficacy of TLR inhibitory oligoribonucleotides in a psoriasis mouse model (Jiang et al, 2013) and even a first clinical trial in psoriasis patients (Balak et al, 2017). We could imagine the combined blockade of naRNA effects via TLRs, NOD2, and/or NLRP3, especially when applied topically, to emerge as an effective novel strategy to target multiple exRNA or NET-related inflammatory responses.

# Methods

## Reagents

PMA (tlrl-pma), nigericin (tlrl-nig), LL37 (tlrl-l37), as well as the PRR ligands LPS (tlrl-peklps), R848 (tlrl-r848), TL8 (tlrl-tl8506), MDP (tlrl-mdp), Pam2CSK4 (tlrl-pm2s-1) and the TLR8-inhibitor CU-CPT9a (inh-cc9a) were from Invivogen, Ionomycin was acquired from Sigma (I0634-1MG). FCS (Biowest) was heat-inactivated for 30 min at 56 °C. RNase inhibitor (N2615) was from Promega, RNase A (EN0531), Dnase I (EN0521) and Dnase inhibitor (EN0521) were from Thermo Fisher. The PAD4-inhibitor Cl-amidine (506282) was from Merck Millipore. The NLRP3-inhibitor MCC950 was from Cayman chemicals (Cay17510-1) and the GSDMD-inhibitor disulfiram from Medchem Express (HYB0240). DOTAP (L787.2) was from Roth (see Reagent Table), ssRNA40 was from Eurogentec and CpG PTO 2006 from TIB Molbiol (see Reagent Table). Bacterial RNA isolated from *S. aureus*

## Reagents and tools table

| Reagent/resource | Reference or source | Identifier or catalog number |
|---|---|---|
| **Experimental models** | | |
| *Unc93b1*[3d/3d] | Tabeta et al, 2006, Tatjana Eigenbrod, Heidelberg | |
| *Tlr13*[−/−] | Li and Chen, 2012, Tatjana Eigenbrod, Heidelberg | |
| *Tlr9*[−/−] | Birgit Schittek, Tübingen | |
| LysM[EGFP/+] | Faust et al, 2000, James Chen, Houston | |
| *Tlr13*[−/−] | Li and Chen, 2012, David Nemazee, La Jolla | |
| *NOD2*[−/−] | Institut Pasteur de Lille, U1019 – UMR 8204 – CIIL – Centre d'Infection et d'Immunité de Lille, F-59000, Lille, France | |
| *S. aureus* | Herster et al, 2020 | |
| *C. albicans* SC5314 | Chang et al, 2022 | |
| Neutrophils of healthy human donors | Herster et al, 2020 | |
| PBMCs were isolated from whole blood or buffy coats | Herster et al, 2020 | |
| Stem cells derived from human healthy donors | Sioud, 2020 | |
| BlaER1 cells (WT, *Unc93b*[−/−] and *Tlr8*[−/−]) | Holger Heine, Borstel, Germany (Vierbuchen et al, 2017) | |
| THP-1 cells | Thomas Zillinger, Bonn, Germany (Coch et al, 2019) | |
| N/TERT-1 cells | Prof. James Rheinwald (Dickson et al, 2000) | |
| Normal Human Epidermal Keratinocytes (NHEK) single juvenile donor | PromoCell | C-12002 |
| 3D human skin equivalent | Bitschar et al, 2020 | |
| NK-92 MI cells | Melanie Märklin, University Hospital Tübingen | |
| HEK293T | Colak et al, 2014 | |
| Mouse keratinocytes from tails | Lorscheid et al, 2019 | |
| **Recombinant DNA** | | |
| NF-κB reporter | pGL3 | 6× NF-κB response element |
| Renilla | pRL-TK | *Renilla* |
| hTLR7 | pcDNA3.1 (+) | hTLR7 |
| hTLR8 | pcDNA3.1 (+) | hTLR8 |
| hTLR9 | pSEM-hTLR9 | hTLR9 |

| Reagent/resource | Reference or source | Identifier or catalog number |
|---|---|---|
| **Antibodies** | | |
| Anti-hLL37 | LSBio | LS-B6696-500 |
| Anti-rRNA (Y10b) | Santa Cruz Biotechnology | sc-33678 |
| Anti-rRNA (Y10b) Alexa Fluor® 647 | Santa Cruz Biotechnology | sc-33678 AF647 |
| Anti-ASC MS F-9 | Santa Cruz Biotechnology | sc-271054 |
| Anti-mouse IgG | Thermo Fisher | A-21463 |
| Hoechst 33342 | Thermo Fisher | H21492 |
| SYTO RNAselect Green fluorescent dye | Thermo Fisher | S32703 |
| Anti-mouse Cy3 | Jackson ImmunoResearch | 115-165-146 |
| Anti-rabbit AF488 | Molecular Probes | A-11008 |
| Anti-mouse 6 nm gold | Jackson ImmunoResearch | 115-195-166 |
| Anti-rabbit 12 nm gold | Jackson ImmunoResearch | 111-205-144 |
| Human/Mouse Myeloperoxidase/MPO Antibody | R&D systems | AF3667 |
| Histone H3 Antibody - BSA Free | Novusbio | NB500-171 |
| Chicken anti-Rabbit IgG (H+L) Cross-Adsorbed Secondary Antibody, Alexa Fluor™ 594 | Thermo Fisher | A-21442 |
| Chicken anti-Goat IgG (H+L) Cross-Adsorbed Secondary Antibody, Alexa Fluor™ 488 | Thermo Fisher | A-21467 |
| **Oligonucleotides and sequence-based reagents** | | |
| RNA40 | 5′GsCsCsCsGsUsCsUsGsUsUsGsUsGsUsGsAsCsUsC3′ | Eurogentec |
| CpG PTO 2006 | 5′TsCsGsTsCsGsTsTsTsTsGsTsCsGsTsTsTsTsGsTsCsGsTsT3′ | TIB |
| IL-8 | Hs00174103_m1 | |
| IL17C | Hs00171163_m1 | |
| IL36G | Hs00219742_m1 | |
| LL37 (CAMP) | Hs00189038_m1 | |
| NOD2 | Hs01550753_m1 | |
| TBP (housekeeper for human keratinocytes) | HS00427620_m1 | |
| IL17C F | GGAGACAGCATGAAGGACCTC | |
| IL17C R | GCTTCTGTGGGTAGCGGTTC | |
| Cxcl10 F | CCCACGTGTTGAGATCATTGC | |
| Cxcl10 R | CTCTGCTGTCCATCCATCGC | |
| Cxcl1 F | ACGTGTTGACGCTTCCCTTG | |
| Cxcl1 R | TCCTTTGAACGTCTCTGTCCC | |
| Cxcl2 F | CGCCCAGACAGAAGTCATAGC | |
| Cxcl2 R | CTTTGGTTCTTCCGTTGAGGG | |
| Cxcl5 F | CCCTACGGTGGAAGTCATAGC | |
| Cxcl5 R | GAACACTGGCCGTTCTTTCC | |
| Actin F (housekeeper for murine keratinocytes) | AGGAGTACGATGAGTCCGGC | |
| Actin R (housekeeper for murine keratinocytes) | GGTGTAAAACGCAGCTCAGTA | |
| *NOD2* | Hs_CARD15_3 FlexiTube siRNA 5 nmol | Qiagen |
| – | AllStars Neg. Control siRNA (20 nmol) | Qiagen |
| *MAVS* | Hs_VISA_1 FlexiTube siRNA 5 nmol | Qiagen |
| *MyD88* | Hs_MYD88_5 FlexiTube siRNA | Qiagen |
| *TRIF* | Hs_TICAM1_2 FlexiTube siRNA 5 nmol | Qiagen |
| *NLRP1* | Hs_NALP1_5 FlexiTube siRNA 5 nmol | Qiagen |

| Reagent/resource | Reference or source | Identifier or catalog number |
|---|---|---|
| Chemicals, enzymes, and other reagents | | |
| Cl-amidine | Merck | 506282 |
| CU-CPT9a | Invivogen | inh-cc9a |
| Disulfiram | Medchem Express | HYB0240 |
| DNase I | Thermo Fisher | EN0521 |
| DNase inhibitor 50 mM EDTA | Thermo Fisher | EN0521 |
| DOTAP | Roth | L787.2 |
| Ionomycin | Sigma | I0634-1MG |
| LL37 | Invivogen | tlrl-l37 |
| LPS-EK (ultrapure) | Invivogen | tlrl-peklps |
| MCC950 | Cayman Chemicals | Cay17510-1 |
| MDP | Invivogen | tlrl-mdp |
| Nigericin | Invivogen | tlrl-nig |
| Pam2CSK4 | Invivogen | tlrl-pm2s-1 |
| PMA | Invivogen | tlrl-pma |
| R848 (Resiquimod) | Invivogen | tlrl-r848 |
| RNase A | Thermo Fisher | EN0531 |
| RNase inhibitor | Promega | N2615 |
| TL8-506 | Invivogen | tlrl-tl8506 |
| AF546-Azide | Jena Bioscience | CLK-1283-1 |
| Aminoguanidine-hydrochloride | Merck | 396494-25G |
| $CuSO_4$-click chemistry grade | Jena Bioscience | CLK-MI004-50.1 |
| 5-Ethynyluridine | Jena Bioscience | CLK-N002-10 |
| Na-Ascorbate-click chemistry grade | Jena Bioscience | CLK-MI005-1G |
| THPTA (Tris((1-hydroxy-propyl-1H-1,2,3-triazol-4-yl) methyl)amine) | Jena Bioscience | CLK-1010-25 |
| Software | | |
| Excel 2019 | Microsoft | |
| Prism 8 | GraphPad | |
| ImageJ | NIH | Win64 |
| Zen Blue | Zeiss | Version 3 |
| FlowJo V10 | | |
| FACSDiva | BD Bioscience | Version 6 |
| FlowJo | FlowJo LLC | Version 10 |
| GENECODE | EMBL-EBI | – |
| Living Image software | Caliper | – |
| Prism | GraphPad | Version 8 |
| QuantStudio Real-Time-PCR software | Thermo Fisher | Version 1.3 |
| Salmon | – | Version 1.5.0 |
| Living Image software | Caliper | |
| tximport | Bioconductor | – |
| SIS cell software | Olympus | – |
| Other | | |
| DyLight 550 Conjugation Kit (Fast) | Abcam | ab201800 |
| 1.54 M $NH_4Cl$ | Roth | 5470.1 |

| Reagent/resource | Reference or source | Identifier or catalog number |
|---|---|---|
| 100 mM KHCO$_3$ | Fluka | 60220 |
| 1 mM EDTA; pH 8 | Thermo Fisher | 15575020 |
| Dissolved in Ampuwa water | Fresenius Kabi | 1833 |
| ELISA MAX™ Deluxe Set Human IL-8 | Biolegend | 431504 |
| ELISA MAX™ Deluxe Set Human IL-6 | Biolegend | 430504 |
| ELISA MAX™ Deluxe Set Human IFN-γ | Biolegend | 430104 |
| ELISA MAX™ Deluxe Set Human IL-1β | Biolegend | 437015 |
| Axioplan microscope | Zeiss | – |
| Compact coating unit | Safematic | CCU-010 |
| Confocal microscope | Zeiss | LSM800 |
| Critical point dryer | Leica Microsystems | CPD300 |
| Cytofunnel™ | Thermo Fisher, Shandon | – |
| Cytospin centrifuge | Shandon | – |
| Cytoclip™ slide clips | Thermo Fisher, Shandon | – |
| EDTA blood collection tubes (S-Monovette) | Sarstedt | 41,931,010 |
| FACS Canto II | BD Bioscience | – |
| IVIS Lumina II imaging system | Caliper | – |
| Manual Caliper | Peacock, Japan | 0.01-10 mm |
| Microtube homogenizer | Merck | BeadBug™ |
| Nanozoomer microscope | Hamamatsu | – |
| NEBNext® Ultra™ II Directional RNA Library Prep Kit | Illumina | protocol for use with rRNA Depleted FFPE RNA |
| Plate reader for dual luciferase reporter assay | BMG Labtech | FLUOstar OPTIMA |
| Poly-L-lysine coated glass coverslips | Electron Microscopy Sciences | 72292-04 |
| Scanning electron microscope (field emission) | Hitachi High Technologies | Regulus 8230 |
| Tapestation | Agilent | 4200 |
| tximport | Bioconductor | – |

was prepared as described (Herster et al, 2020). Fungal RNA from *C. albicans* strain SC5314 was isolated as described below, as well as naRNA isolated from PMA NETs. For complex formation to stimulate cells in a volume of 500 μL, 5.8 μM ssRNA40 (~34.4 μg/mL), fungal RNA (fRNA; 125 ng/mL), bacterial RNA (bRNA; 10 μg/mL) or PMA NET derived naRNA (~600 ng/mL) was mixed with 10 μg LL37 and left for 1 h at room temperature (RT). For a smaller volume of cells, complexes were used in the according fractional amount. For RNA-only or LL37-only controls, the same amounts and volumes were used replacing one of the components with sterile, endotoxin-free H$_2$O. For complex formation with DOTAP, the according RNA or CpG was incubated with the transfection reagent for 10 min at RT prior to stimulation of the cells. For complex formation of MDP and Ionomycin, 20 μg/mL MDP and 0.1875 μg/mL ionomycin were premixed before usage as a stimulus. NET content for stimulation was prepared as described below. Additional details and information on antibodies used for fluorescence microscopy, click chemistry reagents, constructs used for transfection of HEK293T, qPCR primers and siRNAs used for knockdown of *NOD2*, *MyD88*, *TRIF*, *MAVS*, and *NLRP1* are listed in the accompanying Reagent Table.

## Preparation of fungal RNA from *C. albicans*

*C. albicans* SC5314 (kindly cultured and prepared by Tzu-Hsuan Chang, Tübingen) was plated in a slant tube containing YPD agar and grown overnight at 30 °C as described in (Chang et al, 2022). One colony was picked from the slant tube and resuspended in 500 μL YPD medium, centrifuged at 10,000 rpm for 1 min and washed with phosphate-buffered saline (PBS). Afterward, the pellet was resuspended in 200 μL RLT buffer (derived from Rneasy Mini Kit, Qiagen, #74106) and transferred into a 2 mL tube containing 0.5 mm diameter ceramic beads. The tube was filled up to 1 mL with RLT buffer and the fungi were subsequently homogenized by using a microtube homogenizer (BeadBug™, Merck) with an interval of seven times 2 min shaking at 2800 rpm and 1 min cooling break on ice. The supernatant was transferred into a new tube containing 1 mL 75% ethanol (VWR, 20821.330). The further RNA isolation was performed according to the manufacturer's instructions using the Qiagen Rneasy Mini Kit for purification of Total RNA from Animal Tissues (Rneasy Mini Kit, Qiagen, 74106). The RNA was eluted in 30 μL RNase Dnase-free H$_2$O and the concentration was determined with a Nanodrop Spectrophotometer.

## Mice

*Unc93b1$^{3d/3d}$* (Tabeta et al, 2006), *Tlr13$^{-/-}$* (Li and Chen, 2012) (both kindly provided by Tatjana Eigenbrod, Heidelberg and on C57BL/6 background), and WT C57BL/6 mice between 8 and 20 weeks of age were used in accordance with local institutional guidelines on animal experiments, regular hygiene monitoring, and specific locally approved protocols compliant with the German regulations of the Gesellschaft für Versuchstierkunde/Society for Laboratory Animal Science (GV-SOLAS) and the European Health Law of the Federation of Laboratory Animal Science Associations (FELASA) for sacrificing and in vivo work. *Unc93b1$^{3d/3d}$*, *Tlr13$^{-/-}$*, *Nlrp3$^{-/-}$*, and WT C57BL/6 control mice were housed in controlled specific-pathogen-free animal facilities at the Interfaculty Institute of Cell Biology, Tübingen. Local federal authority for the approval of experimental protocols was the Regierungspräsidium Tübingen. *Tlr9$^{-/-}$* mice and matched WT C57BL/6 control mice were a kind gift from Birgit Schittek, Tübingen. LysM$^{EGFP/+}$ (Faust et al, 2000), *Tlr13$^{-/-}$* mice (Li and Chen, 2012) (kindly provided by James Chen, Houston and David Nemazee, La Jolla) and WT (all on a C57BL/6 background) mouse strains were bred and maintained under the specific pathogen-free conditions, with air isolated cages at an American Association for the Accreditation of Laboratory Animal Care (AAALAC)-accredited animal facility at Johns Hopkins University and handled according to procedures described in the Guide for the Care and Use of Laboratory Animals as well as Johns Hopkins University's policies and procedures as set forth in the Johns Hopkins University Animal Care and Use Training Manual, and all animal experiments were approved by the Johns Hopkins University Animal Care and Use Committee (MO21M378). *NOD2$^{-/-}$* and corresponding WT C57BL/6 mice were housed in appropriate facilities at the Institut Pasteur de Lille, U1019 – UMR 8204 – CIIL – Centre d'Infection et d'Immunité de Lille, F-59000, Lille, France, and sacrificed according to locally approved protocols held by M. Chamaillard. Additional WT C57BL/6 control mice were also housed in appropriate facilities at the Department of Dermatology, University Medical Center of the Johannes Gutenberg-University Mainz, Mainz, Germany, and sacrificed according to locally approved protocols held by D. Kramer. In the case of keratinocyte analysis sex-and age-matched mice were used at an age of 6–8 weeks.

## Isolation and stimulation of primary bone-marrow-derived polymorphonuclear neutrophils (BM-PMNs)

Bone-marrow (BM)-PMNs were isolated from the bone marrow as described (Herster et al, 2020). In brief, bones were isolated from the respective mice and the bone marrow was flushed out. Afterward, neutrophils were isolated using magnetic separation (mouse Neutrophil isolation kit, Miltenyi Biotec, 130-097-658) following the manufacturer's instructions. In total, $1 \times 10^5$ cells/well PMNs were seeded in a 24-well plate, and stimulation was carried out with PMA (600 nM), ssRNA+LL37 complex (as previously described), nigericin (50 µM), live *C. albicans* (MOI1) or human NET content (mock control and PMA NETs, 1:50 dilution) for 16 h at 37 °C and 5% CO$_2$. Subsequently, cells were stained for immunofluorescence.

## Study participants and human blood or tissue sample acquisition

All healthy donors included in this study provided their written informed consent before participation. Approval for use of biomaterials was obtained for this project by the local ethics committee of the Medical Faculty Tübingen in accordance with the principles laid down in the Declaration of Helsinki as well as applicable laws and regulations.

## Primary human neutrophil isolation and stimulation

Neutrophils of healthy human donors were isolated as described (Herster et al, 2020). In brief, EDTA-anticoagulated whole blood was diluted in PBS (Thermo Fisher, 14190-169), loaded on Ficoll (1.077 g/mL, Sigma, 10771) and centrifuged for 25 min at $509 \times g$ at RT without brake. Afterward, all layers except the erythrocyte-granulocyte layer were discarded and erythrocyte lysis was performed twice (for 20 and for 10 min) using 1x ammonium chloride erythrocyte lysis buffer (see Reagent Table) at 4 °C on roller shaker. The remaining cells were resuspended in culture medium (RPMI culture medium (Sigma-Aldrich, R8758) + 10% fetal bovine serum (FBS; heat-inactivated at 56 °C for 30 min, sterile filtered, Biowest) to a concentration of $1.6 \times 10^6$ cells/mL. 500 µL of cells were seeded in a 24-well plate for immunofluorescence microscopy or 8 mL of $5 \times 10^6$ cells/mL in 10 cm uncoated dishes for NET preparation and naRNA isolation/isolation of whole PMN RNA. After seeding, the cells were rested for 30 min at 37 °C and 5% CO$_2$, followed by 3 h stimulation with PMA (600 nM), nigericin (50 µM), live *C. albicans* (MOI2), RNA+LL37 complexes (as previously described), NET content at indicated dilutions or 3 h priming with lipopolysaccharide (LPS) (10 ng/mL) and 2 h nigericin stimulation (5 µM) for IF and IL-1β ELISA or 4 h stimulation with PMA (600 nM) for NET preparation. Where indicated, the cells were incubated with 100 nM TLR8-inhibitor CU-CPT9a, 200 µM PAD4-inhibitor Cl-amidine, 10 µM NLRP3-inhibitor MCC950, or 25 µM GSDMD-inhibitor disulfiram 30 min before stimulation and medium was not replaced during incubation with the respective stimuli. Although neutrophil apoptosis was not formally ruled out, there was no evidence for the distinct morphological features associated with neutrophil apoptosis (e.g., apoptotic blebs (Douda et al, 2014) and uniform, unilobar nuclei (Gray et al, 2018)) in inhibitor-only treated neutrophils.

## Primary peripheral blood mononuclear cell (PBMC) isolation and stimulation

PBMCs were isolated from whole blood or buffy coats as described (Herster et al, 2020). In brief, EDTA-anticoagulated blood was diluted in PBS and density gradient separation was performed as described above. The PBMC layer was then carefully transferred into another reaction tube and diluted in PBS (1:1). The cell suspension was spun down at $645 \times g$ for 8 min and the cells were washed two more times in PBS and resuspended in culture medium (RPMI + 10% FBS (heat inactivated at 60 °C) +1% L-glutamine) at a density of $1 \times 10^6$ cells/mL. Afterward, 200 µL of PBMCs were seeded in a 96-well plate and stimulated with LPS (100 ng/mL), R848 (5 µg/mL), TL8 (100 ng/mL), ssRNA (1.6 µg/mL) + DOTAP

(50 μg/mL), ssRNA+LL37 complex (as described above), and respective NET content (1:20 dilution) for 24 h at 37 °C and 5% CO₂. Where indicated, the TLR8 inhibitor CU-CPT9a was added to the cells at a concentration of 1 μM 2 h before stimulation and was not removed for the incubation of the cells with the respective stimuli. After the stimulation, the plate was centrifuged for 5 min at 1500 rpm and the supernatant was collected and stored at −80 °C until the ELISA was performed.

## Preparation of NETs and isolation of naRNA/whole PMN RNA

NETs were prepared by 600 nM PMA treatment for 4 h at 37 °C and 5% CO₂, or cells were left untreated during the incubation as the mock control. After incubation, the neutrophils were gently washed three times with PBS to get rid of PMA as the stimulus for NET formation, any cytokines released by the cells, unbound free RNA and unstimulated PMNs, as those do not adhere to the uncoated petri dish used here. Carryover of PMA was ruled out by transient incubation and washing of Mock NETs with PMA as shown in Appendix Fig. S1. Similar results were obtained when NETs were generated using 60 nM PMA or the ionophore A23187, i.e., NETs generated with lower or alternative stimuli also functioned as DAMPs (also Appendix Fig. S1). In some conditions (as indicated) during the NET preparation process and for storage, naRNA was protected by addition of 10 U/μL RNase inhibitor (= mock or PMA NETs + RNase Inhibitor). For digest of NETs, the NET content was incubated for 20 min at 37 °C with RNase A (Thermo Fisher, EN0531; 100 μg/mL, EDTA for Dnase inhibition added) at 37 °C. For creation of fresh and old NETs, PMN were stimulated with PMA for 3 h as described above. Subsequently, cells were washed with PBS and old NETs were created by additional incubation of the NET-forming cells in fresh PMN culture medium or pooled human serum (1:10 in PBS) for 4 h at 37 °C in the presence or absence of 10 U/μL RNase inhibitor. Afterward, NET content was harvested as described above. For isolation of naRNA, 10 U/μL RNase inhibitor was added to the PMNs during NET formation. After the above-described washing process, PMA or mock NETs were resuspended in 300 μL of ML buffer and RNA isolation was performed according to the manufacturer's instructions (NucleoSpin miRNA isolation kit, Macherey-Nagel, 740971.50). The RNA was eluted in 50 μL RNase Dnase-free H₂O and the concentration was determined with a Nanodrop Spectrophotometer. For isolation of whole PMN RNA, untreated PMNs were directly lysed, and the RNA was isolated from the cells according to the manufacturer's instructions.

## Preparation of human primary stem cell-derived PMNs

Stem cells derived from human healthy donors were prepared and differentiated as described (Sioud, 2020). In brief, bone marrow was diluted in PBS, carefully layered on Ficoll-Paque medium (density: 1.077 g/mL) and centrifuged at $500 \times g$ for 25 min at RT without brakes. The interphase layer containing the mononuclear cell fraction was transferred to a new tube and washed twice with 30 mL ice-cold PBS by centrifugation at $300 \times g$ for 8 min at 8 °C. Further, the cells were resuspended, counted, and isolated using Human CD34 MicroBead Kit (Miltenyi) for magnetic beads-based isolation of CD34⁺ cells from BM-MNCs. Afterward, the number of

CD34⁺ HPSCs was determined and cultured in CD34⁺ culture medium (Stemline II Hematopoietic Stem Cell Expansion medium supplemented with 10% FCS, 1% L-glutamine, 1% penicillin/streptomycin, and a human recombinant cytokine cocktail consisting of 20 ng/mL IL-3, 20 ng/mL IL-6, 20 ng/mL TPO, 50 ng/mL SCF, and 50 ng/mL FLT-3L) at a density of $2 \times 10^5$/mL at 37 °C and 5% CO₂. The medium was replaced every second day and the cells were cultured for 14 days. During the differentiation process, the cells were treated with 100 μM 5-ethynyluridine for 14 days for subsequent click chemistry labeling of endogenous RNA or were left untreated as negative controls. To verify differentiation, cell morphology was assessed using Cytospin assay. In brief, the Cytoclip™ slide clips were loaded by fitting the filter card, the sample chamber, and the glass slide. An assemble Cytoclip™ slide clip was then placed in the slide clip support plate of the cytospin centrifuge. $2 \times 10^4$ cells from liquid culture differentiation were pipetted into Cytofunnel™ and centrifuged for 3 min at $200 \times g$. The cytospin slides were stained for 5 min in May-Grünwald stain, rinsed shortly with ddH₂O, and then stained for 10 min in Wright-Giemsa stain. Afterward, the slides were rinsed shortly with ddH₂O, and cell morphology was determined using a microscope. To further verify differentiation, flow cytometric analysis was performed using antibodies specific for the following hematopoietic/myeloid markers: CD45 (leukocyte marker), CD34 (HSPC marker), CD33 (promyelocyte marker), CD11b (myeloid cell marker), CD14 (monocyte marker), and CD15 and CD16 (neutrophil markers). Neutrophil percentage was determined by gating on neutrophils as follows: CD45⁺CD11b⁺CD15⁺, or CD45⁺CD11b⁺CD16⁺, or CD45⁺CD15⁺CD16⁺ cells.

## BlaER1 cell culture, transdifferentiation, and stimulation

BlaER1 cells (WT, $Unc93b^{-/-}$, and $Tlr8^{-/-}$), a kind gift of Holger Heine, Borstel, Germany (Vierbuchen et al, 2017), were cultured, transdifferentiated, and stimulated for 18 h with the respective stimuli as described (Herster et al, 2020). In brief, $1 \times 10^6$ cells/well were seeded in a 6-well plate and differentiated with 10 ng/mL hIL-3, 10 ng/mL hMCSF, and 150 nM β-estradiol in complete RPMI-1640 (PANBiotech, P04-18525) for 7 days. Afterward, $5 \times 10^4$ differentiated cells were reseeded in a 96-well plate, followed by 1 h resting. Cells were treated with the respective stimuli (LPS at 0.1 μg/mL, R848 at 5 μg/mL, TL8 at 100 ng/mL or ssRNA+LL37 complex as described above) and mock or PMA NETs with or without RNase inhibitor) in complete medium in a total volume of 125 μL/well for 18 h. After stimulation, the cells were centrifuged for 5 min at 1200 rpm, the supernatant was transferred into a new plate and stored at −80 °C until the ELISA was performed.

## THP-1 cell culture, differentiation, and stimulation

THP-1 cells (THP-1 cells were a kind gift from Thomas Zillinger, Bonn, Germany (Coch et al, 2019)), were cultured in complete RPMI-1640 (Sigma, R8758-24X500ML) medium. For differentiation into macrophage-like cells, $5 \times 10^4$ cells/well were seeded in a 96-well plate, treated with 300 ng/mL PMA and incubated for 16 h at 37 °C and 5% CO₂. The next day, the cells were washed three times with PBS and fresh medium was added, followed by 48 h of resting. Subsequently, the medium was removed, exchanged by medium containing 200 U/mL IFN-γ (Sigma, I-3265), and the cells

were incubated for 6 h. After repeated washing and medium exchange, cells were treated with the respective stimuli (PMA (25 µg/mL) + Ionomycin (0.375 µg/mL), LPS (0.1 µg/mL), R848 (5 µg/mL), TL8 (40 ng/mL), ssRNA+LL37 complex (as described above), mock NETs + RNase inhibitor (1:50 dilution), PMA NETs (1:50 dilution) and PMA NETs + RNase inhibitor (1:50 dilution)) in complete medium in a total volume of 125 µL/well for 18 h. After stimulation, the cells were centrifuged for 5 min at 1200 rpm, the supernatant was transferred into a new plate and stored at −80 °C until the ELISA was performed.

## N/TERT-1 keratinocyte cell culture and stimulation

N/TERT-1 cells (a kind gift from Prof. James Rheinwald (Dickson et al, 2000)) were cultured for less than ten passages in complete CnT-07 medium (CELLnTEC, CnT-07). Two days prior to stimulation, a total amount of $2 \times 10^4$ cells/well was seeded in a 96-well plate and incubated at 37 °C and 5% $CO_2$. The medium was renewed, and the cells were stimulated in a total volume of 125 µL/well of the respective stimuli diluted in medium (PMA (25 µg/mL) + Ionomycin (0.375 µg/mL), TL8 (200 ng/mL), ssRNA+LL37 complex (as previously described), as well as mock and PMA NETs with and without RNase inhibitor at indicated dilutions). After 24 h stimulation, the cells were centrifuged at 1200 rpm for 5 min and the supernatant was stored in a new plate at −80 °C until further usage.

## MyD88 knockdown in N/TERT-1 keratinocytes

N/TERT-1 keratinocytes were grown as described and a total amount of $2.5 \times 10^5$ cells/well was seeded in a 6-well plate. The next day, cells were washed, medium without supplements was added and the cells were transfected using HiPerFect transfection reagent (Qiagen, 301705) and the respective siRNA (3.5 nM, Qiagen) or scrambled negative control (3.5 nM, Qiagen). After incubation for 16–18 h the medium was removed, and the cells were washed twice with PBS before reseeding of the cells in a 96-well plate ($5 \times 10^4$ cells/well). The cells are rested again for 24 h before stimulation with the respective stimuli as described above for IL-8 ELISA.

## Primary normal human epidermal keratinocyte (NHEK) cell culture and stimulation

NHEK cells (Normal Human Epidermal Keratinocytes (NHEK) single juvenile donor, proliferating, PromoCell, C-12002) were grown and stimulated in Keratinocyte Growth Medium 2 (PromoCell, C-20111). For stimulation for ELISA, a total amount of $2 \times 10^4$ cells/well was seeded in a 96-well plate and incubated at 37 °C and 5% $CO_2$ two days prior to stimulation. For stimulation, the medium was renewed with basal medium (PromoCell, C-20211) containing 1.7 mM $CaCl_2$ (Roth, CN93.1) and stimuli were added and diluted in medium in a final volume of 125 µL (MDP+Ionomycin as described above, $Pam_2CSK4$ (2.54 µg/mL), ssRNA (34.4 ng/mL), R848 (20 µg/mL), TL8 (200 ng/mL), ssRNA+LL37 complex, as well as mock NETs + RNase inhibitor (1:25 dilution) and PMA NETs with and without RNase inhibitor or RNase A or Dnase I digest (1:25 dilution)). After 24 h stimulation, the cells were centrifuged at 1200 rpm for 5 min and the supernatant was stored in a new plate at −80 °C until further usage. For qPCR analysis, a total amount of $2.5 \times 10^5$ cells/well was seeded in a 6-well plate and incubated at 37 °C and 5% $CO_2$ two days prior to

stimulation. For stimulation, the medium was renewed, and stimuli were added and diluted in medium in a final volume of 750 µL/well (ssRNA (34.4 ng/mL), CpG (2.5 µM), ssRNA+LL37 complex, as well as mock NETs + RNase inhibitor (1:25 dilution) and PMA NETs with and without RNase inhibitor or RNase A or Dnase I digest (1:25 dilution)). After 24 h stimulation, the medium was removed, and the cells were lysed in 350 µL RLT buffer. RNA isolation and qPCR was performed as described below. mRNA levels (CT values) of RNase- or Dnase-treated samples were indistinguishable from other samples.

## NOD2, TRIF, MAVS, and NLRP1 knockdown in NHEK

NHEK were grown as described and a total amount of $2.25 \times 10^5$ cells/well was seeded in a 6-well plate. The next day, cells were washed, medium without supplements was added and the cells were transfected using HiPerFect transfection reagent (Qiagen, 301705) and the respective siRNA (3.5 nM, Qiagen) or scrambled negative control (3.5 nM, Qiagen). After incubation for 16–18 h the medium was removed, and the cells were washed twice with PBS before reseeding of the cells in a 96-well plate ($4 \times 10^4$ cells/well). The cells are rested again for 24 h before stimulation with the respective stimuli as described above for IL-8 and IL-1β ELISA.

## Preparation of 3D human skin equivalent

The 3D human skin equivalent was prepared as described (Bitschar et al, 2020). Briefly, primary fibroblasts were seeded on collagen and incubated in FF medium for five days. Subsequently, primary keratinocytes were added to the wells and airlifting was performed on day 12. On day 22, the 3D skin model was stimulated with NET content (25 µL/well for one 3D construct grown in a 12-well chamber) or respective water control for 24 h. Afterward, supernatant was harvested for ELISA, RNA was isolated for qPCR analysis and H&E staining was performed.

## NK-92 MI cell culture and stimulation

NK-92 MI cells (kindly provided by Melanie Märklin, University Hospital Tübingen) were cultured in IMDM-Medium (Lonza, 12-722F). For stimulation, a total amount of $1 \times 10^5$ cells/well were seeded in a volume of 200 µL and rested for 2 h at 37 °C and 5% $CO_2$. The cells were stimulated with LPS (100 ng/mL), CpG (2.5 µM) + DOTAP (25 µg/mL), R848 (5 µg/mL), TL8 (100 ng/mL), ssRNA (1.6 µg/mL) + DOTAP (50 µg/mL), ssRNA+LL37 complex (as described above), or NET content (1:100 dilution) for 24 h and afterward centrifuged for 5 min at 1500 rpm. Subsequently, the supernatant was transferred into a new plate and stored at −80 °C until further usage.

## Flow cytometry

After PMN isolation, the purity and activation status of the cells was determined by flow cytometry as described (Herster et al, 2020). In brief, 200 µL of cells were transferred into a 96-well plate (U-bottom) and centrifuged for 5 min at $448 \times g$. Afterward, blocking was performed using pooled human serum diluted 1:10 in PBS for 15 min at 4 °C. After washing, the samples were stained for 20 min at RT in the dark and fixed (4% PFA in PBS) after repeated washing for 10 min at RT in the dark. After an additional washing

step, the cell pellets were resuspended in 300 μL PBS and measurements were performed on a FACS Canto II (BD Bioscience, Diva software). Analysis was performed using FlowJo V10 analysis software.

## RNA-sequencing analysis of naRNA

Isolated naRNA was analyzed for quality control using the Agilent 4200 TapeStation system. Subsequently, the RNA was sequenced according to NEBNext® Ultra™ II Directional RNA Library Prep Kit for Illumina® using the protocol for use with rRNA Depleted FFPE RNA. The data was quantified using Salmon Version1.5.0 and tximport was used to obtain the transcript-level quantification. Data have been deposited in the NCBI Gene Expression Omnibus under accession number GSE253440. For transcript classification, GENECODE annotation was performed, and transcripts per million (TPM) values were aggregated for each biotype.

## Fluorescence microscopy of fixed human or murine primary neutrophils

500 μL of $1.6 \times 10^6$ cells/mL of human blood PMNs, and $2 \times 10^6$ cells/mL murine BM-PMNs were seeded in a 24-well plate containing poly-L-lysine-coated glass coverslips (Electron Microscopy Sciences, 72292-04) and rested for 30 min before stimulation at 37 °C and 5% $CO_2$ with the according stimuli (as described above) for 3 h (human) or 16 h (murine), respectively, as adapted from Brinkmann et al, 2004 (Brinkmann et al, 2004). After stimulation, the cells were carefully washed with PBS and fixed with fixation buffer (Biolegend, 420801) for 10 min at RT in the dark. Afterward, the cells were blocked with PBS containing 0.1% heat-inactivated diethylpyrocarbonate (DEPC) (Roth, K028.2), 10% chicken serum (Normal Chicken Serum Blocking Solution S-3000, Biozol/Vectorlabs, VEC-S-3000-20), 0.1% saponin (Applichem, A4518.0100), as well as 10 U/μL RNase inhibitor for 2 h at RT. The primary antibodies (rRNA Y10b, hLL37, ASC (MS F-9) see Reagent Table) were diluted 1:50 (rRNA Y10b and LL37) or 1:200 (ASC) in blocking buffer and subsequently incubated for 2 h at RT. Afterward, the cells were washed three times with PBS containing 0.1% heat inactivated DEPC and incubated with the secondary antibodies (see Reagent Table) in a 1:500 dilution in blocking buffer for 1 h. After repeated washing, the cells were incubated with SYTO RNAselect (Thermo Fisher; 50 μM) for 10 min to stain RNA and/or Hoechst 33342 (Thermo Fisher; 1 μg/mL) for 5 min to stain nuclear DNA. Although high specificity to RNA has been confirmed in multiple studies (Li et al, 2006; Wu et al, 2020; Zhou et al, 2015) and the line plot analysis in Fig. EV5C–E supports this, a very low binding to DNA cannot be fully excluded. Secondary antibodies alone did not yield any significant staining under identical staining and acquisition conditions. The coverslips were mounted (ProLong™ Diamond Antifade Mountant, Thermo Fisher, P36961) on glass slides and left to dry overnight at RT in the dark. Subsequently, the samples were stored at 4 °C before microscopy using a Zeiss LSM800 Confocal microscope (40× or 63× magnification with z-stack acquisition, AiryScan mode) and image analysis using ImageJ-Win64 and Zen Blue3 software was performed. For IF of ultrathin sections, coverslips were coated with a carbon layer, glow-discharged and UV-treated overnight before cells were seeded.

After incubation, the cells were fixed in 2.5% glutaraldehyde in PBS for 2 h at room temperature followed by incubation at 4 °C. Samples were prepared by the progressive lowering of temperature method (PLT). Therefore, the fixed cells were incubated in ethanol for infiltration of Lowicryl HM20, followed by UV polymerization and subsequent sectioning into 50–60 nm ultrathin slices. Immunolabeling was performed by incubation with anti-rRNA Y10b and anti-LL37 primary antibodies for 1 h at RT. Afterward, the cells were incubated with goat anti-mouse Cy3 and goat anti-rabbit AF488 in blocking buffer for 1 h at RT. Following immunolabeling, the samples were treated with 1% uranyl acetate for 5 min at RT before analysis using a Zeiss Axioplan microscope (63× objective, Olympus SIS cell software) and image/line plot analysis was performed using ImageJ-Win64. The scale bar shown in the first image of one panel is the same for all images of the respective panel, if not stated differently.

## Quantification of NET formation

To quantify the formation of NETs, microscopy images were obtained using a Zeiss LSM800 Confocal microscope with a 40× objective and 3×3 tiles acquisition. Three images per sample of three biological replicates were taken. To quantify NET formation by using NET-related signal dispersion of rRNA and DNA signal, ImageJ-Win64 software was used, and a threshold (Triangle threshold) was applied as originally described (Zack et al, 1977). Particles were analyzed with a ROI (region of interest) manager (size (micron$^2$): 100-infinity (pixel units); circularity 0.00–1.00) and average size and number of particles (ROIs) were assessed. In NETs, RNA and DNA signals showed up in a greater number and smaller size, making the usage of the ratio suitable as a measurement of NET-related signal dispersion. In the case of using the DNA signal only to assess the extent of NET formation, ImageJ-Win64 software was used to create a PNG image and a grid with 8×8 tiles was manually applied to the images (Appendix Fig. S2). Tiles containing extracellular DNA (Hoechst signal) or RNA (rRNA Y10 b-AF647 signal) structures were manually counted in a blinded manner as NET-positive tiles. This and other methods cannot exclude overlapping NETs; hence for highly stimulatory conditions the actual number of NETs may be higher than what was quantified.

## Live fluorescence microscopy of enzymatic digest of human NETs

500 μL of $1.6 \times 10^6$ cells/mL of human blood PMNs were seeded in a 4-well glass bottom microscopy cell culture dish (Greiner, 627871) and rested for 30 min before stimulation at 37 °C and 5% $CO_2$ with PMA (600 nM) for 3 h. After stimulation, the medium was carefully removed and the cells were washed with PBS before adding fresh culture medium (RPMI culture medium without phenol red (Sigma-Aldrich, R7509) + 10% FBS (heat inactivated at 60 °C, sterile filtered, TH Geyer, 11682258)). Hoechst 33342 (Thermo Fisher, 1 μg/mL) to stain nuclear DNA and SYTO RNAselect Green fluorescent dye (Thermo Fisher, 50 μM) to stain naRNA was added to the cells, as well as RNase A (Thermo Fisher, EN0531; 100 μg/mL) was added between time point 0 and 5 min. Live cell imaging was performed using a Zeiss LSM800 Confocal microscope with a 63× objective and z-stack acquisition, taking an

image every 5 min for 30–60 min, respectively. Image analysis and video creation was performed using ImageJ-Win64.

## Click chemistry of primary, stem cell-derived PMNs, and fluorescence microscopy

500 µL of $1.6 \times 10^6$ cells/mL of human stem cell-derived PMNs treated with 5-ethynyluridine or left untreated were seeded in a 24-well plate containing poly-L-lysine-coated glass coverslips and rested for 30 min before stimulation with PMA (600 nM) at 37 °C and 5% $CO_2$ for 12 h. After stimulation, the cells were washed and permeabilized with ice-cold acetone (Applichem, A1582.2500PE) 1:1 methanol (Honeywell, 32213-2.5L) for 5 min at RT. Subsequently, the click chemistry (reagents see Reagent Table) labeling of endogenous RNA was performed as described (Presolski et al, 2011). Briefly, in a total volume of 500 µL, 2 µL of AF546-Azide, a pre-mixture of 1 mM $CuSO_4$ and 1.25 mM THPTA, 5 mM aminoguanidine-hydrochloride and 5 mM Na-ascorbate in PBS were added to the cells in a 24-well plate. The wells were sealed with plastic foil and incubated for 1 h at RT while shaking in the dark. For negative controls, 5-ethynyluridine untreated cells incubated with complete click chemistry reagents and 5-ethynyluridine treated cells incubated with PBS and AF546-Azide only were used. No significant signals were observed. After the incubation, the cells were washed three times for 5 min with PBS and counterstained with rRNA Y10b-AF647 (see Reagent Table) at 1:50 in PBS for 2 h at RT in the dark. After repeated washing, the cells were incubated with Hoechst 33342 to stain nuclear DNA and mounted as described above. Imaging and analysis were performed as previously described.

## Electron microscopy

For electron microscopy, 500 µL of $1.6 \times 10^6$ cells/mL of human blood-derived PMNs were seeded in a 24-well plate containing coverslips which were pre-coated with 0.01% poly-L-lysine (Sigma, A-005-C) for 15 min at 37 °C for scanning electron microscopy (SEM). For transmission electron microscopy (TEM), coverslips were coated with a carbon layer, glow-discharged, and UV-treated overnight before cells were seeded. Cells were rested for 30 min at 37 °C and 5% $CO_2$ and subsequently stimulated with 1200 nM PMA or left untreated for 3 h. Afterward, the cells were fixed in 2.5% glutaraldehyde in PBS for 2 h at room temperature followed by incubation at 4 °C. For SEM, samples were post-fixed with 1% osmium tetroxide for 1 h on ice. Subsequently, samples were dehydrated in a graded ethanol series followed by critical point drying (CPD300, Leica Microsystems) with $CO_2$. Finally, the cells were sputter-coated with a 4-nm thick layer of platinum (CCU-010, Safematic) and examined with a field emission scanning electron microscope (Regulus 8230, Hitachi High Technologies) at an accelerating voltage of 3 kV. For antibody labeling, cells treated as described above fixed in 4% formaldehyde in PBS for 1–2 h at room temperature and 4 °C overnight. After washing and blocking (0.2% gelatin in PBS) samples were incubated for 1 h at room temperature with rRNA Y10b as the primary antibody in blocking buffer and for 1 h at RT with goat anti-mouse antibodies coupled to 6 nm gold in blocking buffer (Jackson ImmunoResearch, code number 115-195-166). Samples labeled with 6 nm gold were further silver enhanced. Following immunolabeling, the samples were treated with 1% uranyl acetate for 5 min at RT, dehydrated and critical point dried as before. Samples were sputter-coated with a 5 nm thick layer of carbon (CCU-010, Safematic) and analyzed in the SEM with an accelerating voltage of 5 kV. For TEM, samples were prepared by the PLT method as described above for IF of ultrathin sections. Samples were placed on grids (Cu, Ni) and subsequently sectioned into 50–60 nm ultrathin slices. Immunolabeling was performed by incubation with anti-rRNA Y10b as the first primary antibody in blocking buffer at 4 °C overnight, followed by incubation with anti-LL37 as the second primary antibody for 1 h at RT. Afterward, the cells were incubated with goat anti-mouse coupled to 6 nm gold and goat anti-rabbit 12 nm gold secondary antibodies in blocking buffer for 1 h at RT. Following immuno-labeling, the samples were treated with 1% uranyl acetate for 5 min at RT before analysis in the TEM.

## Transient transfection of HEK293T cells

HEK293T cells were transiently transfected with the respective TLR8, TLR7, TLR9, and NF-κB reporter plasmids as described (Colak et al, 2014) (see Reagent Table for plasmids) using X-tremeGENE™ HP DNA Transfection Reagent (Merck, 6366236001). A total amount of $5 \times 10^4$ cells/well were seeded in a 24-well plate one day prior to transfection. For the transfection of one well, 100 ng of the according TLR plasmid, 100 ng of the firefly luciferase NF-κB reporter and 10 ng *Renilla* luciferase control reporter was mixed in Opti-MEM™ Reduced Serum Medium (Thermo Fisher, 31985062) in a total volume of 50 µL. After 15 min incubation at RT, the transfection mix was added to the cells and the cells were incubated for 48 h. Prior to subsequent stimulation, the medium was changed to complete DMEM medium (Sigma, D5796-24X500ML), and the cells were incubated with the respective NET content stimuli and controls (R848 (2.5 µg/mL), TL8 (100 ng/mL), CpG (1.25 µM) + DOTAP (25 µg/mL), ssRNA (0.6 µg/mL) + DOTAP (20 µg/mL), various NET contents as indicated (1:50)) for 18 h at 37 °C and 5% $CO_2$. Supernatants were then removed, and the cells frozen briefly at −80 °C. Subsequently, they were used for dual luciferase assay.

## Dual-luciferase reporter assay

The dual luciferase reporter assay for detection NF-κB activation after TLR transfection and subsequent stimulation was performed as described (Herster et al, 2020). In brief, supernatants were removed from the cells after stimulation and 60 µL/well of 1x passive lysis buffer (E194A, Promega) was added. The plate was then incubated for 15 min at RT on the plate shaker and subsequently stored at −80 °C for at least 15 min to facilitate complete cell lysis. After thawing, the cell solution (60 µL) was transferred into a V-bottom 96-well plate and centrifuged for 10 min at 2500 rpm and 4 °C to pellet cell debris. Ten microliters of supernatant were then transferred into a white microplate and each condition was measured in triplicates using the FLUOstar OPTIMA device (BMG Labtech). Firefly and Renilla luciferase activity were determined using the Promega Dual luciferase kit. Both enzyme activities were measured for 12.5 s with 24 intervals of 0.5 s, respectively. The data was analyzed by calculating the ratio of the two measured signals, thereby normalizing each firefly luciferase signal to its corresponding Renilla luciferase signal. The

ratios were represented as the relative light units (RLU) of NF-κB activation.

## Extracellular bacterial killing of *S. aureus*

The killing assay of *S. aureus* with human PMNs was performed according to Brinkmann et al, 2004 (Brinkmann et al, 2004). In brief, PMNs were seeded at a density of $2 \times 10^6$ cells/mL and incubated with PMA (600 nM) for 2 h at 37 °C. Afterward, the medium was carefully replaced with serum-free culture medium, containing 2% heat-inactivated pooled human serum with cytochalasin D (10 µg/mL) and incubated further for 15 min before infection with bacteria. Cytochalasin D treatment did not affect NETs and this concentration was effective in blocking bacterial phagocytosis. To investigate whether naRNA of NETs was important in extracellular killing, samples were either treated with RNase A (Thermo Fisher, EN0531; 100 µg/mL) or DNase I (Thermo Fisher, EN0521; 1 U/10 µL) for 2 h during NET formation, or after NET formation during the 30 min bacterial killing process. Samples were centrifuged at $700 \times g$ for 10 min and incubated at 37 °C and 5% $CO_2$ for 30 min. Bacterial killing was measured as percentages of control values (bacteria incubated alone in media without neutrophils).

## *S. aureus* colonization after priming of keratinocytes with NET content

To investigate potential priming effects of synthetic RNA on keratinocytes modulating colonization of *S. aureus*, primary human keratinocytes isolated from foreskin and maintained in CnT-07 medium (CELLnTEC) were expanded and differentiated for three days in presence of 2 mM $CaCl_2$. Subsequently, cells were starved without supplements or $CaCl_2$ for 24 h and primed in the meantime with ssRNA40 (103.2 µg/well). Afterward, the keratinocytes were infected with *S. aureus* (USA300 WT LAC, MOI30) for 1 h before the supernatant was removed. The cells were then washed with PBS and fresh medium was added, as well as starvation medium with antibiotics. Gentamicin (10 µg/mL) and amphotericin B (0.25 µg/mL) were added to kill excessive bacteria which did not adhere or internalize. After additional 23 h incubation, supernatant was removed, and cells were washed before trypsinization with 0.05% Trypsin/EDTA. Detachment was stopped via addition of DMEM medium containing 10% FCS, cells were spun down and once washed with PBS. For CFU analysis, cells were lysed in 1 mL 0.1% Triton-X-100 for 1 h at room temperature, plated on tryptic soy agar plates (Sigma, 22091) overnight was performed and CFU was determined.

## In vivo analysis of naRNA DAMP effects

To investigate the effect of NETs with or without RNase inhibitors and the respective TLR signaling, 20 µL of NET content and the respective controls were injected into the ears of C57BL/6 and $Tlr13^{-/-}$ mice intradermally on day 0. Afterward, as a measure of inflammation, the ear thickness was assessed using a manual caliper (0.01–10 mm, Peacock, Tokyo, Japan) until day 4.

## Neutrophil infiltration in vivo fluorescence imaging

The in vivo experiment for investigating neutrophil infiltration was performed as described (Herster et al, 2020). Briefly, LysM$^{EGFP/+}$

mice were injected intradermally with 20 µL of PMA or mock NET content or respective controls. LysM$^{EGFP/+}$ mice were then anesthetized with inhalation isoflurane and in vivo fluorescence imaging was performed using the IVIS Lumina II imaging system (Caliper). EGFP fluorescence was measured using excitation (465 nm), emission (515–575 nm), and exposure time (0.5 s). Data are quantified as total radiant efficiency ([photons/s]/[µW/cm²]) within a circular region of interest using Living Image software (Caliper).

## Imiquimod model of psoriatic skin inflammation

To analyze the effect of RNA signaling in an in vivo model for psoriasis, the well-established imiquimod mouse model was used (Gilliet et al, 2004). C57BL/6 and $Tlr13^{-/-}$ mice were used and, briefly, 70 µL (62.5 mg) of imiquimod (5%, Taro Pharmaceuticals Industries, Hawthorne, NY) was applied daily to both sides of a mouse ear for 5 consecutive days (day 0 to 4). Ear thickness was measured with a manual caliper (0.01–10 mm, Peacock, Tokyo, Japan) before imiquimod application. A day after the last application of imiquimod (day 5), full-thickness ear skin was excised with surgical scissors for histologic analysis.

## Histologic analysis of mouse ear skin

Ear skin specimens were collected by excising the entire ear with surgical scissors, fixed in formalin (10%), and embedded in paraffin. 4 µm thickness sections were mounted onto glass slides and stained with H&E by the Johns Hopkins Reference Histology Laboratory. To measure epidermal thickness, 100 epidermal thickness measurements per mouse were averaged from images taken at ×200 magnification (Hamamatsu Nanozoomer) using ImageJ software (NIH).

## Immunofluorescence microscopy of murine ear skin

Formalin-fixed, paraffin-embedded ear skin samples from IMQ treated WT or *Tlr13* KO mice were deparaffinized and rehydrated using Roti Histol (Roth, 6640.1) and decreasing concentrations of ethanol (100%, 100%, 95%, 80%, and 70%). After rinsing in dd$H_2$O, antigen retrieval was performed by boiling for 35 min in citrate buffer (0.1 M, pH = 6). The skin tissue was then washed 3 times for 5 min with PBS. Blocking was performed using chicken serum (1:10 in PBS) overnight at 4 °C. The primary antibody was added for 2 h at RT. After 3 washes, the secondary antibody was added for 1 h at RT in the dark. Thereafter, the samples were washed again and Hoechst 33342 (ThermoFisher, 1 µg/ml) was added for 10 min at RT in the dark. Then, 3 washes were performed before using ProLong Diamond Antifade Mountant (Thermo Fisher, P36965) for mounting. The samples were left to dry overnight at RT in the dark at 4 °C before confocal microscopy analysis. For imaging, a Zeiss LSM800 Confocal microscope with an oil 40× objective and 55 tiles acquisition was used. Three images per sample of 7 WT and 7 *Tlr13* KO biological replicates were taken. ImageJ-Win64 software was used to create TIFF images, which were auto-thresholded and then 10–70 µm ROIs automatically counted in each channel. The number of ROIs per image was plotted.

## ELISA

To measure cytokine release of human and murine neutrophils, BlaER-1, THP-1, N/TERT-1, NHEK, murine keratinocytes, NK-92 MI, PBMCs and 3D human skin equivalent after stimulation with NET content and respective controls, ELISA Kits for hIL-8 (ELISA MAX™ Deluxe Set Human IL-8, Biolegend, 431504), hIL-6 (ELISA MAX™ Deluxe Set Human IL-6, Biolegend, 430504), IFN-γ (ELISA MAX™ Deluxe Set Human IFN-γ, Biolegend, 430104), and human IL-1β (Biolegend, 437015) were used according to the manufacturer's instructions. Samples were assessed in triplicates.

## qPCR analysis of *IL-8*, *IL17C*, *IL36G*, and *LL37* expression of NHEK and 3D human skin equivalent

To investigate *IL-8*, *IL17*C, *IL36G*, and *LL37* expression of NHEK cells and 3D human skin equivalent after stimulation with NET contents, qPCR analysis was performed. First, total RNA was isolated using the RNeasy Mini Kit (Qiagen, 74106) for animal tissues and cells. For cDNA preparation, the High-Capacity RNA-to-cDNA-Kit (Thermo Fisher, 4387406) was used according to the manufacturer's instructions. For the qPCR, the TaqMan™ system was used. Briefly, a master mix of TaqMan™ Universal Mastermix II (Thermo Fisher, 4440040) and TaqMan™ Gene Expression Assay (Thermo Fisher, 4448892) was prepared according to the manufacturer's instructions. For one reaction, 5.5 μL master mix and 4.5 μL of the respective cDNA were mixed and the qPCR was run. Analysis in triplicates was performed using QuantStudio Real-Time-PCR software version 1.3 (Thermo Fisher).

## Isolation and stimulation of murine keratinocytes from mouse tail

Basically, mouse keratinocytes from tails were isolated as described (Lorscheid et al, 2019). In brief, tails were incubated in 5 U/mL Dispase solution (Stem Cell, 07913) overnight at 4 °C. Next day, epidermis and dermis were separated, followed by a 15 min incubation of the epidermis with Trypsin/EDTA at RT. Digestion was stopped with DMEM medium containing 10% FCS. After centrifugation, cells were seeded in collagen-coated plates using K-SFM medium (Gibco Thermo Fisher, 17005042) in the presence of 0.05 M CaCl$_2$. At 80% confluency, cells were starved overnight followed by stimulation for 1 h with either ssRNA, mock control NETs, or PMA-activated NET content.

## qPCR analysis of murine *IL17C*, *Cxcl10*, *Cxcl1*, *Cxcl2*, and *Cxcl5* expression of primary murine keratinocytes

To investigate *IL17C*, *Cxcl10*, *Cxcl1*, *Cxcl2*, and *Cxcl5* expression of primary murine keratinocytes, qPCR analysis was performed. Total RNA was extracted from keratinocytes with QIAzol (QIAGEN, Hilden, Germany, 79306) according to manufacturer's instructions. Contaminating genomic DNA was removed by treatment with DNAse I (Thermo Fisher Scientific, Waltham, MA, USA, EN0523) in the presence of Ribonuclease Inhibitor (Thermo Fisher Scientific, EO0382). For reverse transcription cDNA was synthesized from 2 μg total RNA using Random hexamer primer (Thermo Fisher Scientific, SO142) and Revert Aid reverse transcriptase. cDNA reaction was performed for 1 h at 42 °C. Relative gene expression

was quantified by real-time PCR using the Green master mix (Genaxxon, Ulm, Germany, M3023.0500) and self-designed primers. Real-time PCR analysis was performed on CFX384 (Bio-Rad) using the following PCR conditions: initial denaturation 15 min at 95 °C, followed by 40 cycles of 95 °C for 15 s and 60 °C for 45 s. Relative mRNA levels were calculated by normalization to reference gene *Actin* using the 2$^{-\Delta\Delta Ct}$ method.

## Statistics

Experimental data were analyzed using Excel 2019 (Microsoft) and/or GraphPad Prism 8, microscopy data with ImageJ-Win64 or ZenBlue3 software, flow cytometry data with FlowJo V10. Normal distribution in each group was always tested using the Shapiro–Wilk test first for the subsequent choice of a parametric (ANOVA, Student's t-test for normally distributed data) or non-parametric (Mann–Whitney U) test as indicated in figure legends. *p*-values ($\alpha = 0.05$) were then calculated as indicated in the figure legends using Prism. Multiple testing was always corrected for in Prism. Values <0.05 were generally considered statistically as significant and denoted by * throughout even if considerably lower. Comparisons made to unstimulated control, unless indicated otherwise, were denoted by brackets.

## Data availability

All sourced data has been submitted to the journal, RNAseq data have been deposited in the NCBI Gene Expression Omnibus under accession number GSE253440.

The source data of this paper are collected in the following database record: biostudies:S-SCDT-10_1038-S44319-024-00150-5.

## Peer review information

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

## Acknowledgements

We gratefully acknowledge Jim Rheinwald, Holger Heine, Austin Chang, Thomas Zillinger for the provision of reagents, respectively, and Jon Kagan, Alexander Dalpke and Libera Lo Presti for helpful scientific and editorial comments. We further acknowledge Bettina Danker, Mark Helm and Martina Christina Schmidt-Dengler for very helpful technical and scientific support. We thank all voluntary healthy donors of biomaterials for participating in the study. The study was supported by the Volkswagenstiftung Momentum grant "InnatelyHuman" (to ANRW), the Deutsche Forschungsgemeinschaft (German Research Foundation, DFG) grants CRC TR156 "The skin as an immune sensor and effector organ – Orchestrating local and systemic immunity" (to FB, CG, JF, JS, BS, BF, DK, and ANRW, specifically INST 35/1259-2 and INST 35/1259-3), DFG project We-4195/18-1 (to ANRW and AS), NIH grants R01AI146177, R01AR073665, and R01AR069502 (to NKA). NKA has received previous grant support from Pfizer and Boehringer Ingelheim and was a paid consultant for Janssen Pharmaceuticals. VNCL was specifically supported by the Sao Paulo Research Foundation (FAPESP) grant 2021/13049-4, BS by DFG research project SCHI510/12-1. Infrastructural funding was provided by the University of Tübingen, the University Hospital Tübingen and the DFG Clusters of Excellence "iFIT – Image-Guided and Functionally Instructed Tumor Therapies" (EXC 2180, to AW, PE, BS, and MWL), "CMFI – Controlling Microbes to Fight Infection (EXC 2124 to AW and BS). We gratefully acknowledge support by the Open Access Publication Funds of the University of Tübingen and the Medical Faculty Tübingen Library.

## Author contributions

**Francesca Bork**: Conceptualization; Data curation; Formal analysis; Supervision; Validation; Investigation; Visualization; Writing—original draft; Writing—review and editing. **Carsten L Greve**: Formal analysis; Investigation; Visualization; Writing—review and editing. **Christine Youn**: Formal analysis; Investigation; Visualization; Writing—review and editing. **Sirui Chen**: Formal analysis; Investigation; Visualization; Writing—review and editing. **Vinícius N C Leal**: Formal analysis; Investigation; Visualization; Writing—review and editing. **Yu Wang**: Project administration; Writing—review and editing. **Berenice Fischer**: Resources; Formal analysis; Validation; Investigation; Writing—review and editing. **Masoud Nasri**: Formal analysis; Investigation; Writing—review and editing. **Jule Focken**: Formal analysis; Investigation; Writing—review and editing. **Jasmin Scheurer**: Formal analysis; Investigation; Writing—review and editing. **Pujan Engels**: Formal analysis; Investigation; Writing—review and editing. **Marissa Dubbelaar**: Investigation; Writing—review and editing. **Katharina Hipp**: Formal analysis; Investigation; Visualization; Methodology; Writing—review and editing. **Baher Zalat**: Formal analysis; Investigation; Writing—review and editing. **Andras Szolek**: Software; Formal analysis; Writing—review and editing. **Meng-Jen Wu**: Formal analysis; Investigation. **Birgit Schittek**: Supervision; Funding acquisition; Writing—review and editing. **Stefanie Bugl**: Resources; Writing—review and editing. **Thomas A Kufer**: Resources; Methodology; Writing—review and editing. **Markus W Löffler**: Resources; Project administration; Writing—review and editing. **Mathias Chamaillard**: Resources; Methodology; Writing—review and editing. **Julia Skokowa**: Supervision; Methodology; Writing—review and editing. **Daniela Kramer**: Supervision; Funding acquisition; Methodology; Writing—review and editing. **Nathan K Archer**: Formal analysis; Supervision; Validation; Writing—review and editing. **Alexander N R Weber**: Conceptualization; Resources; Formal analysis; Supervision; Funding acquisition; Validation; Visualization; Writing—original draft; Project administration; Writing—review and editing.

Source data underlying figure panels in this paper may have individual authorship assigned. Where available, figure panel/source data authorship is listed in the following database record: biostudies:S-SCDT-10_1038-S44319-024-00150-5.

## Disclosure and competing interests statement

NKA has received previous grant support from Pfizer and Boehringer Ingelheim and was a paid consultant for Janssen Pharmaceuticals. MWL is an inventor of patents owned by Immatics Biotechnologies and has acted as a speaker and paid consultant for Boehringer Ingelheim. All other authors declare no competing interests.

# Expanded View Figures

▶

**Figure EV1.   Controls, IF microscopy of murine bone marrow-derived neutrophils and of stem cell-derived PMNs, 3D reconstruction of naRNA in NETs, controls of electron microscopy and analysis of isolated naRNA.**

(A) Confocal microscopy of unstimulated or PMA (600 nM) stimulated primary human PMNs after 3 h and stained for naRNA (anti-rRNA Y10b, magenta) and DNA (Hoechst 33342, white). Complete staining and secondary antibody controls only ($n = 3$ biological replicates, representative images, scale bar: 10 μm, white arrows indicate NETs). (B) Confocal microscopy of unstimulated primary human PMNs (control to Fig. 1A) after 3 h and stained for naRNA (anti-rRNA Y10b, magenta) and DNA (Hoechst 33342, white, $n = 3$ biological replicates, representative images, scale bar: 10 μm, white arrows indicate NETs). (C) Confocal microscopy of primary murine BM-PMNs of C57BL/6 WT mice stimulated as indicated for 16 h and stained as in (B) ($n = 3$ biological replicates, representative images, scale bar: 10 μm, white arrows indicate NETs). (D) Confocal microscopy of primary human stem cells differentiated in vitro with/without 100 μM 5-ethynyluridine (5-EU), click-labeled with a fluorescent dye (yellow, total RNA), and stained for naRNA (anti-rRNA Y10b, magenta) and DNA (Hoechst 33342, white, $n = 3$ biological replicates, representative images, scale bar: 10 μm, 2 μm in cropped image, white arrows indicate NETs). (E) Brightfield microscopy analysis of control cytospun of primary human stem cell-derived PMNs shown in (A) ($n = 3$ biological replicates, representative images, scale bar: 10 μm). (F) FACS analysis of cells shown in (D) and (E) ($n = 3$ biological replicates, representative data of one biological replicate shown). (G) As in (B) showing 3D image reconstruction of NETs from z-stacks created with ZenBlue3 ($n = 3$ biological replicates, representative images, scale bar as indicated). (H) Scanning electron microscopy of PMA-treated human primary PMNs showing only secondary antibody staining (no primary antibody) control of Fig. 1C ($n = 1$ biological replicate, representative data; the image on the right is a composite image with signals from secondary electron and backscattered electron detectors for topography and additional material information, respectively). (I) Agilent TapeStation quantification of naRNA isolated from mock or PMA NETs (from $n = 4–6$ biological replicates, combined data, each dot represents one biological replicate). Data information: In (I), data are presented as mean + SD. *$p < 0.05$ according to Mann–Whitney test. Please note that the panel shown in B also appears in Fig. EV2C as these two experiments were carried out simultaneously or were part of the same experiment, and hence control conditions (e.g., unstimulated) are identical. Source data are available online for this figure.

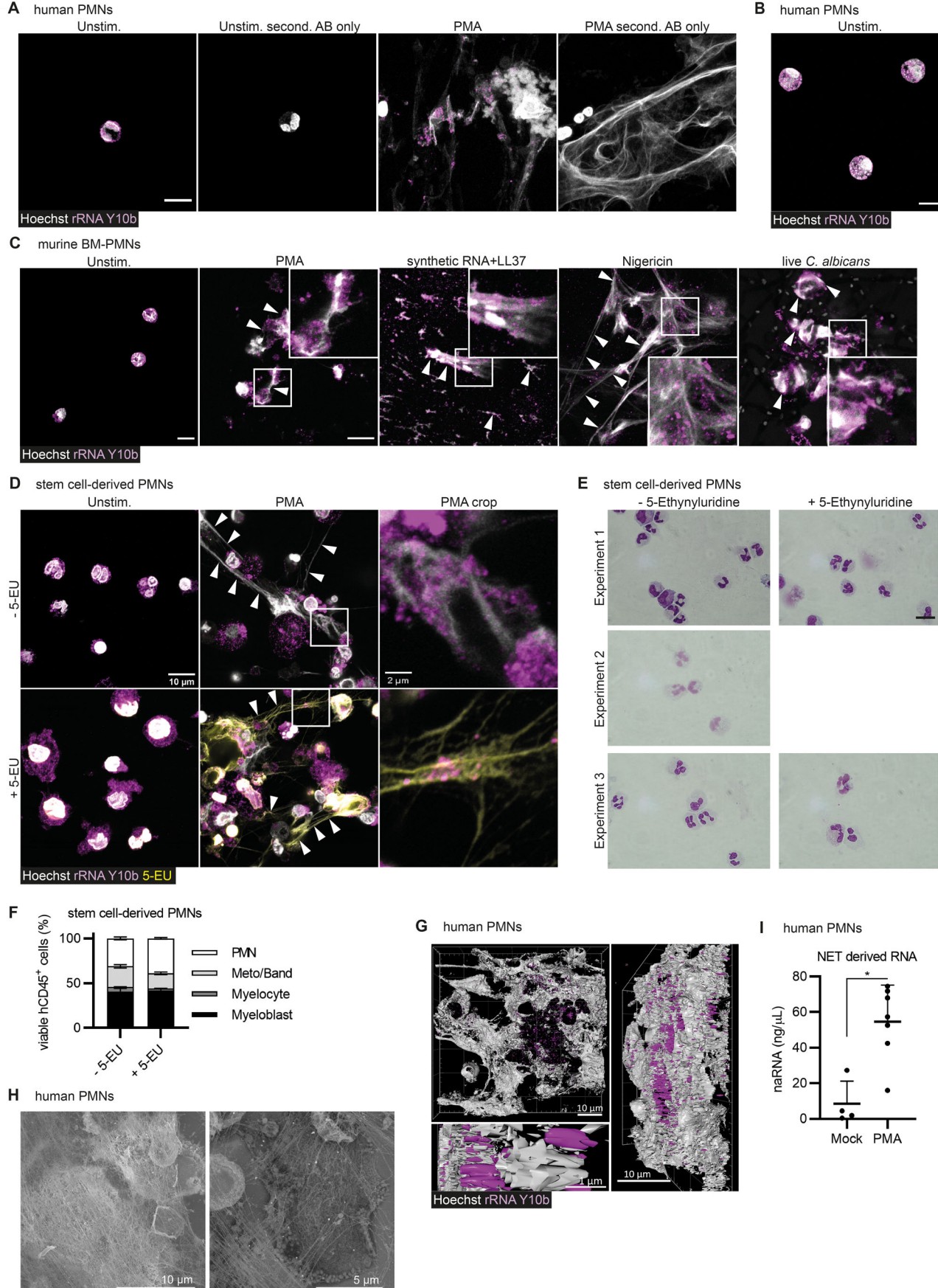

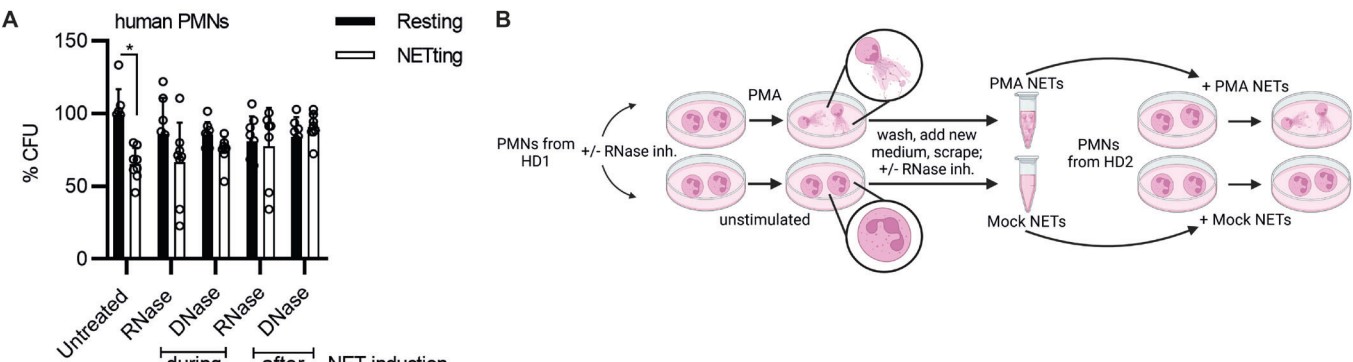

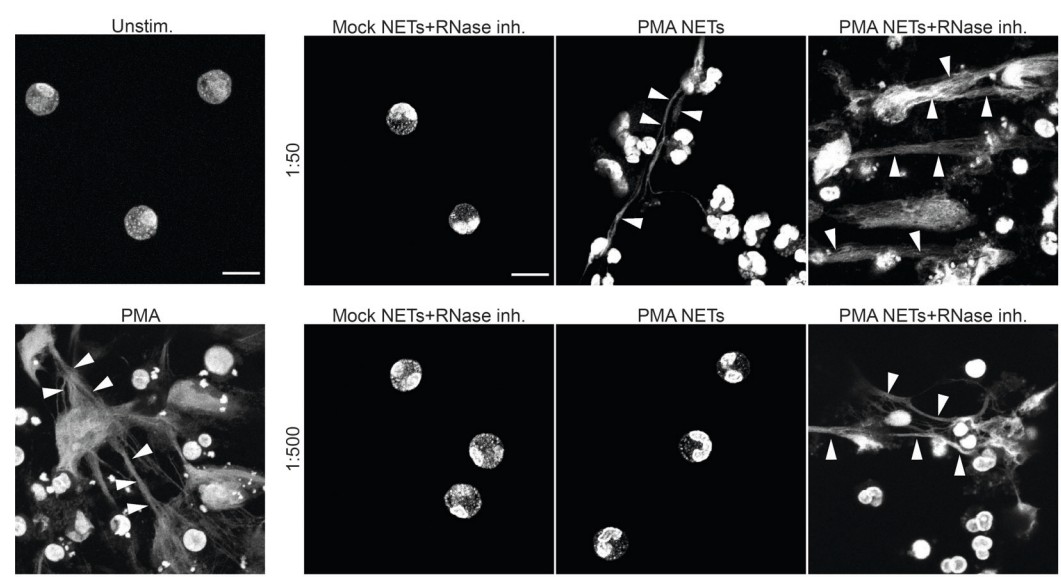

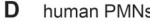

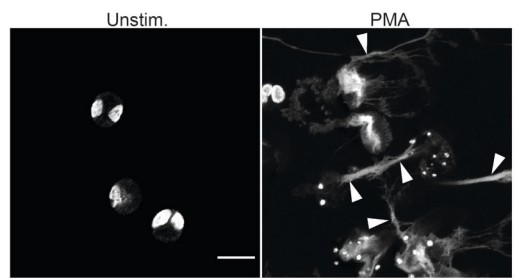

**Figure EV2. Antibacterial effect of NETs on live *S. aureus*, isolation of NET content and controls of IF microscopy.**

(A) Extracellular bactericidal activity of human PMNs/NETs after infection with *S. aureus* and treatment with RNase A and DNase I during or after formation of PMA-induced NETs ($n = 8$ biological replicates, combined data). (B) Workflow for NET content preparation from one donor and transfer to naive human primary PMNs from a second donor (created with BioRender.com). (C) Confocal microscopy of primary human PMNs stimulated for 3 h with PMA (600 nM) or NET content (harvested with/without RNase inhibitor and diluted 1:50 or 1:500), and then stained for NETs/DNA (Hoechst 33342, $n = 9$ biological replicates, representative images, scale bar: 10 μm, white arrows indicate NETs). (F) As in (C) but with/without pre-digestion of NET content with RNase A (controls to Fig. 1F, $n = 3$ biological replicates, representative images, scale bar: 10 μm, white arrows indicate NETs). (E) Confocal microscopy of unstimulated or PMA-stimulated (3 h) primary human PMNs (controls to Fig. 1H), subsequently stained for DNA (Hoechst 33342, white, $n = 9$ biological replicates, representative images, scale bar: 10 μm, white arrows indicate NETs). Data information: In (A), data are presented as mean + SD. *$p < 0.05$ according to one-way ANOVA. Please note that the panel shown in (C) also appears in Fig. EV1B as these two experiments were carried out simultaneously or were part of the same experiment, and hence control conditions (e.g., unstimulated) are identical. Source data are available online for this figure.

**A** human PMNs

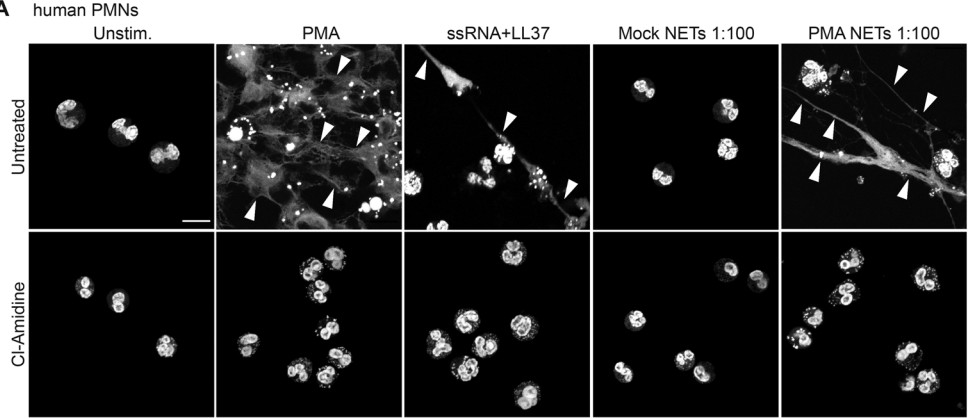

**B** murine BM-PMNs

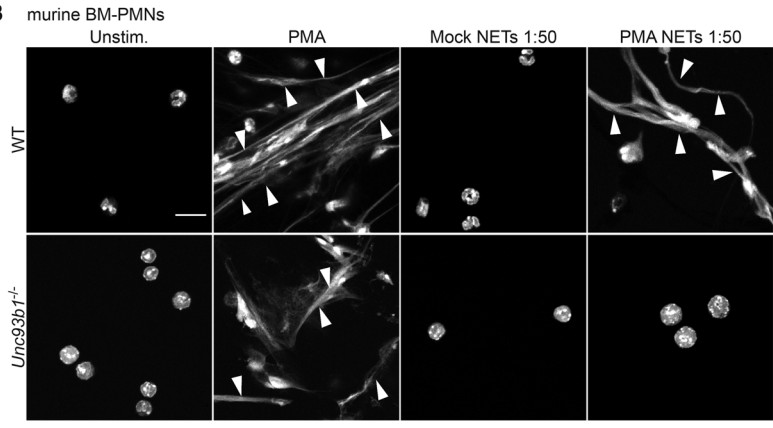

**C** murine BM-PMNs

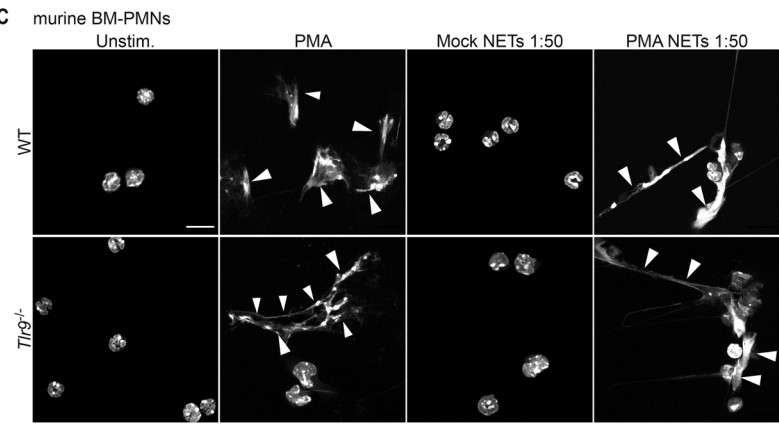

**D** human PMNs

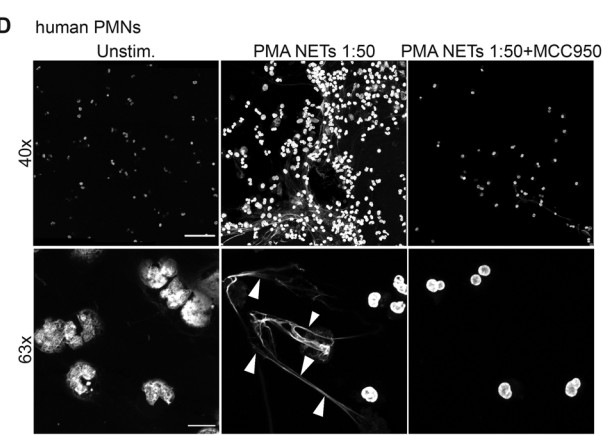

**Figure EV3. Inhibition of PAD4 in human PMNs during NET formation assay, *Unc93b1*$^{-/-}$ and *Tlr9*$^{-/-}$ BM-PMN stimulation with human NETs and inhibition of NLRP3 in human PMNs during NET formation assay.**

(**A**) Confocal microscopy of primary human PMNs, stimulated for 3 h in the presence or absence of the pan-PAD-inhibitor Cl-amidine (200 μM), and subsequently stained for DNA (Hoechst 33342, white) ($n = 3$ biological replicates, representative images; scale bar 10 μm, white arrows indicate NETs). (**B**) Confocal microscopy of primary C57BL/6 WT or *Unc93b1*$^{-/-}$ murine BM-PMNs stimulated for 16 h as indicated in (**A**) ($n = 3$ biological replicates WT, $n = 1$ *Unc93b1*$^{-/-}$ biological replicate, representative images, scale bar: 10 μm, white arrows indicate NETs). (**C**) Confocal microscopy of primary C57BL/6 WT or *Tlr9*$^{-/-}$ murine BM-PMNs stimulated for 16 h as indicated in (**A**) ($n = 3$ biological replicates, representative images, scale bar: 10 μm, white arrows indicate NETs). (**D**) As in (**A**) in the presence or absence of the NLRP3-inhibitor MCC950 (10 μM, $n = 3$ biological replicates, representative images; scale bar 50 μm for 40× and 10 μm for 63×, white arrows indicate NETs). Data information: Please note that selected panels in (**A**), (**B**), and (**D**) also appear in Fig. E2B, (**D**) and (**H**), respectively, as these two experiments were carried out simultaneously or were part of the same experiment, and hence control conditions (e.g., unstimulated) are identical. Source data are available online for this figure.

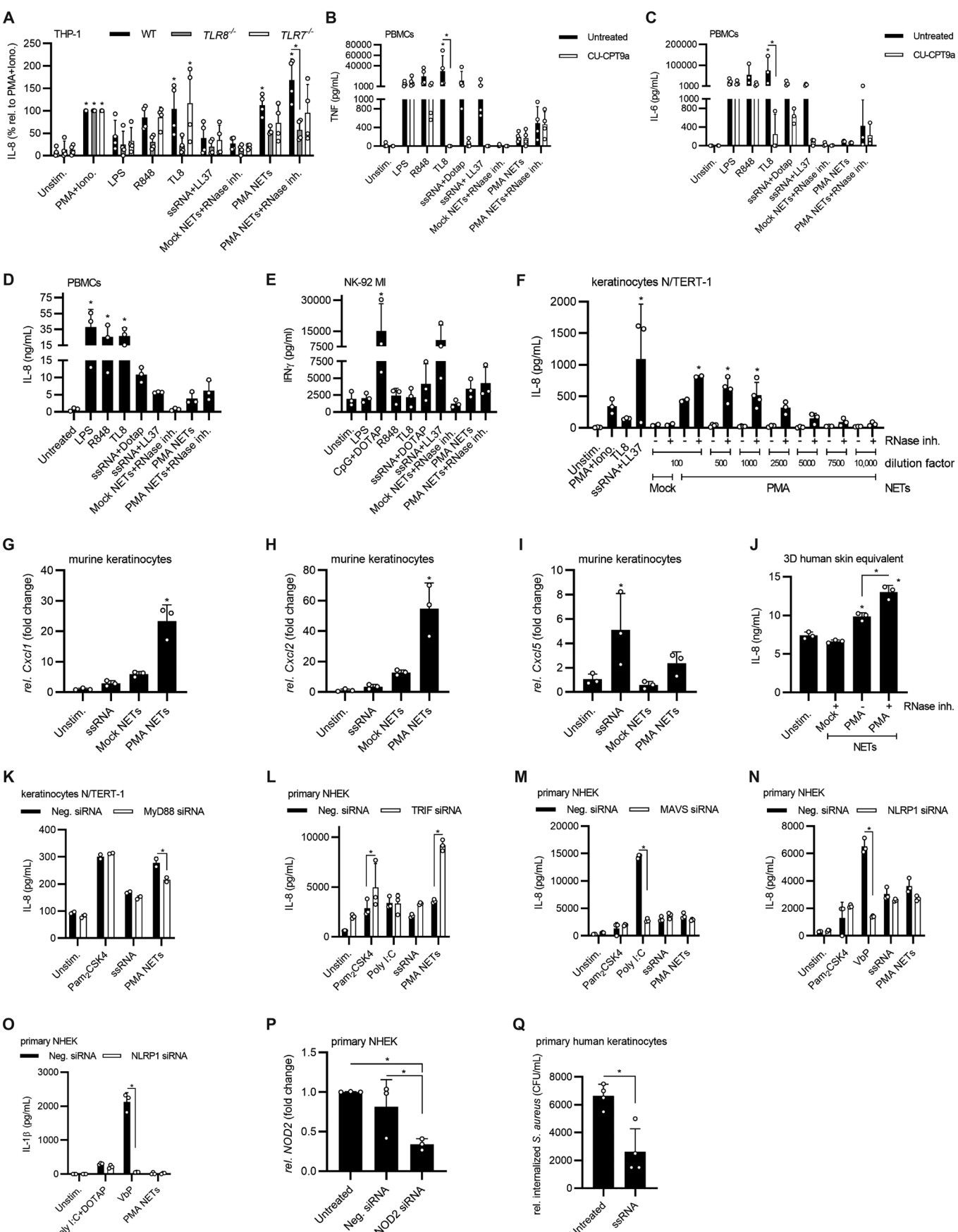

**Figure EV4. Immune responses of PBMCs and human and murine keratinocytes to NETs.**

(A) Levels of IL-8, as measured by triplicate ELISA in WT, *TLR8*$^{-/-}$, and *TLR7*$^{-/-}$ THP-1 cells stimulated as indicated for 18 h. Values were normalized to PMA+ionomycin control (n = 4 biological replicates, combined data, each dot represents one biological replicate). (B–D) Levels of TNF (B, *n* = 4 biological replicates), IL-6 (C, *n* = 3 biological replicates), and IL-8 (D, *n* = 3 biological replicates), as measured by triplicate ELISA in primary human PBMCs stimulated as indicated with/without CU-CPT9a for 24 h (combined data, each dot represents one biological replicate). (E) Levels of IFN-γ, as measured by triplicate upon release from NK-92 MI cells stimulated as indicated for 24 h (*n* = 3 biological replicates, combined data, each dot represents one biological replicate). (F) Levels of IL-8, as measured by triplicate ELISA upon release from N/TERT-1 keratinocytes stimulated as indicated for 24 h (*n* = 3 biological replicates, combined data, each dot represents one biological replicate). (G–I) Fold changes in the expression of (G) *Cxcl1*, (H) *Cxcl2*, or (I) *Cxcl5* in murine C57BL/6 WT keratinocytes stimulated as indicated for 1 h. qPCR was performed in triplicate and fold changes were calculated relative to untreated control (*n* = 3 biological replicates, combined data, each dot represents one biological replicate). (J) Levels of IL-8 as measured by triplicate ELISA upon release from NHEK 3D human skin equivalent constructs stimulated as indicated for 24 h (*n* = 3 biological replicates, representative of one biological replicate is shown, each dot represents one technical replicate). (K) Levels of IL-8, as measured by triplicate ELISA in N/TERT-1 keratinocytes stimulated as indicated with/without MyD88 siRNA knockdown for 24 h (*n* = 1 biological replicate, each dot represents one technical replicate). (L–O) Levels of IL-8 (L–N) or IL-1β (O), as measured by ELISA upon release from primary human normal keratinocytes (NHEK) stimulated as indicated with/without TRIF (L), MAVS (M) or NLRP1 (N, O) siRNA knockdown for 24 h (*n* = 1 biological replicate, each dot represents one technical replicate). (P) Fold changes in the expression of *NOD2* in primary human normal keratinocytes (NHEK) after NOD2 siRNA knockdown. qPCR was performed in triplicate and fold changes were calculated relative to unstimulated control (control to Fig. 3G, *n* = 3 biological replicates, combined data, each dot represents one biological replicate). (Q) In vitro colonization assay (CFU) of primary human keratinocytes primed as indicated for 24 h and subsequently exposed to *S. aureus* for 1 h (*n* = 4 biological replicates, combined data, each dot represents one technical replicate). Data information: In (A–Q), data are presented as mean + SD. In (A–P), *p < 0.05 according to one-way ANOVA. In (Q), *p < 0.05 according to Student's t-test. Source data are available online for this figure.

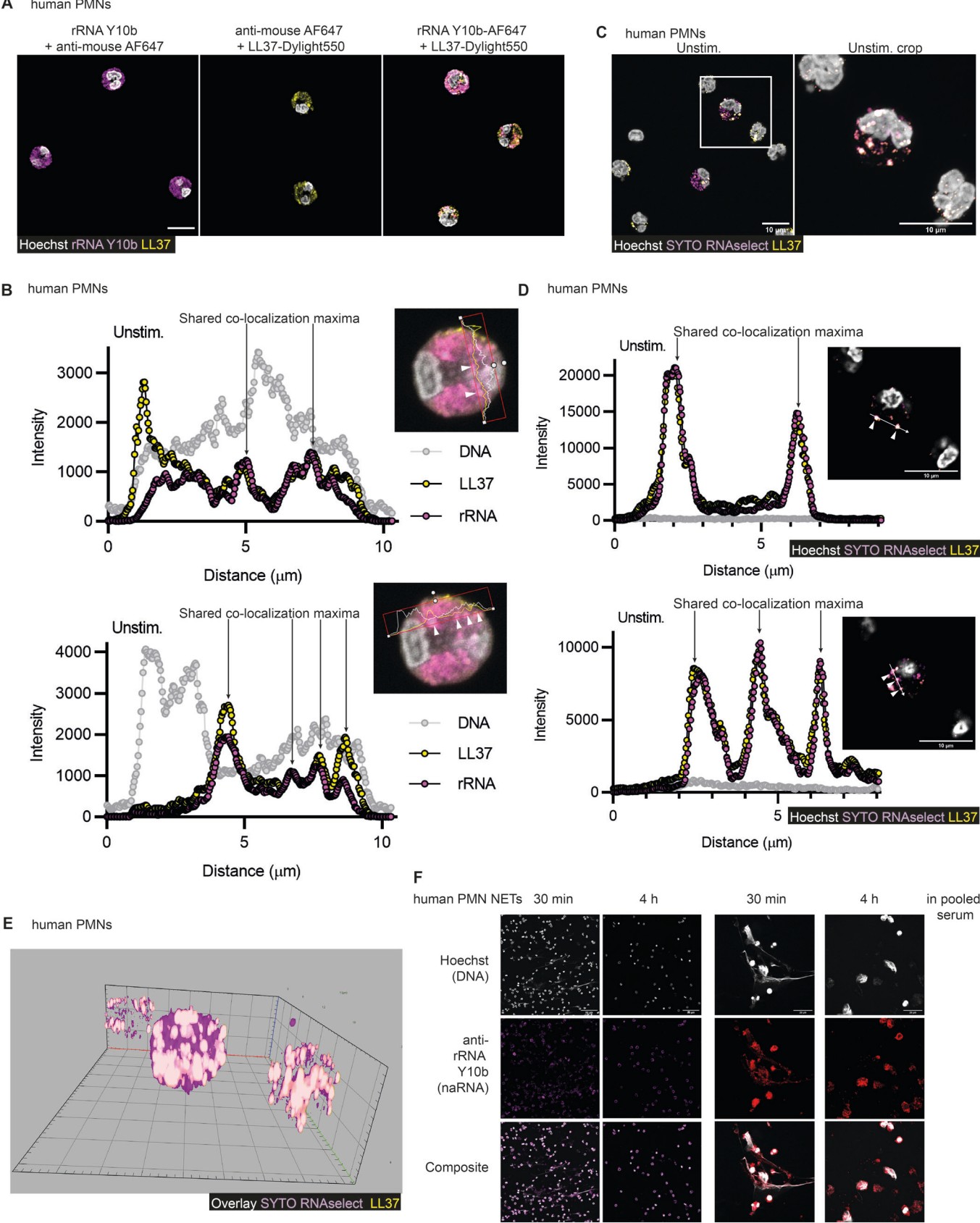

◀ **Figure EV5. Pre-association of naRNA and LL37 in resting and NET-releasing healthy donor neutrophils and graphical abstract.**

(A) Confocal microscopy of primary human PMNs left untreated and stained for RNA only (mouse anti-human rRNA Y10b + anti-mouse AF647, magenta), secondary antibody control for RNA and staining for LL37 (anti-mouse AF647 (magenta) + rabbit anti-human LL37-Dylight550 (yellow)) or counterstaining for rRNA Y10b and LL37 (mouse anti-human rRNA Y10b-AF647 (magenta) + rabbit anti-human LL37-Dylight550 (yellow)) and DNA (Hoechst 33342, white, $n = 3$ biological replicates, representative images, scale bar 10 μm). Controls for Figs. 5A,C and EV5B. (B) Confocal microscopy with line plot analysis of primary human PMNs stimulated as indicated for 3 h and stained for naRNA (anti-rRNA Y10b, magenta), LL37 (anti-hLL37-DyLight550, yellow) and DNA (Hoechst 33342, white, $n = 3$ biological replicates, representative images). The line plot analysis of LL37, RNA, and DNA staining was performed using ZenBlue3 software. One to two different line plots from the same representative image are shown. Additional examples of images shown in Fig. 5C. (C) Confocal microscopy of primary human PMNs left untreated and stained for naRNA (SYTO RNAselect, magenta), LL37 (anti-hLL37-DyLight550, yellow), and DNA (Hoechst 33342, white, $n = 3$ biological replicates, representative images, scale bar 10 μm). (D) Line plot analysis of LL37, RNA, and DNA staining of (A). The analysis was performed using ZenBlue3 software. Three different line plots from the same representative image are shown (scale bar 10 μm). Areas of intensity overlap show up as white. (E) 3D reconstructions of z-stacks from (A). (F) Confocal microscopy of PMA-induced NETs from primary human PMNs incubated for 30 min or 4 h with human serum and stained for DNA (Hoechst 33342, white) and naRNA (anti-rRNA Y10b, magenta or red). Lower magnification (left, scale bar = 50 μm) and higher magnification (right, scale bar = 20 μm) for one presentative of $n = 2$ biological replicates shown. Data information: Please note that selected panels in (D) also appear in Fig. 5C and D, respectively, as these two experiments were carried out simultaneously or were part of the same experiment. Source data are available online for this figure.

