## [Peer Review File · EMBO Reports]

naRNA-LL37 composite DAMPs define sterile NETs as self-limiting drivers of inflammation

Francesca Bork, Carsten Greve, Christine Youn, Sirui Chen, Vinícius Nunes Cordeiro Leal, Yu Wang, Berenice Fischer, Masoud Nasri, Jule Focken, Jasmin Scheurer, Pujan Engels, Marissa Dubbelaar, Katharina Hipp, Baher Zalat, Andras Szolek, Meng-Jen Wu, Birgit Schitteck, Stefanie Bugl, Thomas Kufer, Markus Löffler, Mathias Chamaillard, Julia Skokowa, Daniela Kramer, Nathan Archer, and Alexander Weber

Corresponding author(s): Alexander Weber (alexander.weber@uni-tuebingen.de)

Review Timeline:

Submission Date:	24th Aug 23
Editorial Decision:	22nd Sep 23
Revision Received:	9th Jan 24
Editorial Decision:	6th Feb 24
Appeal Received:	21st Feb 24
Editorial Decision:	22nd Mar 24
Revision Received:	16th Apr 24
Accepted:	18th Apr 24

Editor: Achim Breiling

Transaction Report:

Dear Prof. Weber,

Thank you for the submission of your research manuscript to EMBO reports. I have now received the reports from the three referees that were asked to evaluate your study, which can be found at the end of this email.

As you will see, the referees think that the findings are of interest. However, they have several comments, concerns, and suggestions, indicating that a major revision of the manuscript is necessary to allow publication of the study in EMBO reports. As the reports are below, and all the referee concerns need to be addressed, I will not detail them here.

Given the constructive referee comments, I would like to invite you to revise your manuscript with the understanding that all referee concerns must be addressed in the revised manuscript and/or in a detailed point-by-point response. Acceptance of your manuscript will depend on a positive outcome of a second round of review. It is EMBO reports policy to allow a single round of revision only and acceptance of the manuscript will therefore depend on the completeness of your responses included in the next, final version of the manuscript.

- 1) a .docx formatted version of the final manuscript text (including legends for main figures, EV figures and tables), but without the figures included. Figure legends should be compiled at the end of the manuscript text.
- 2) individual production quality figure files as .eps, .tif, .jpg (one file per figure), of main figures (up to 8) and EV figures. Please upload these as separate, individual files upon re-submission.

- 3) a complete author checklist, which you can download from our author guidelines (<https://www.embopress.org/page/journal/14693178/authorguide>). Please insert page numbers in the checklist to indicate where the requested information can be found in the manuscript. The completed author checklist will also be part of the RPF.

- 4) a complete author checklist, which you can download from our author guidelines

(<https://www.embopress.org/page/journal/14693178/authorguide>). Please insert page numbers in the checklist to indicate where the requested information can be found in the manuscript. The completed author checklist will also be part of the RPF.

5) that primary datasets produced in this study (e.g. RNA-seq, ChIP-seq, structural and array data) are deposited in an appropriate public database. If no primary datasets have been deposited, please also state this in a dedicated section (e.g. 'No primary datasets have been generated and deposited'), see below.

The accession numbers and database should be listed in a formal "Data Availability" section (placed after Materials & Methods) that follows the model below. This is now mandatory (like the COI statement). Please note that the Data Availability Section is restricted to new primary data that are part of this study. This section is mandatory. As indicated above, if no primary datasets have been deposited, please state this in this section

Data availability

8) Regarding data quantification and statistics, please make sure that the number "n" for how many independent experiments were performed, their nature (biological versus technical replicates), the bars and error bars (e.g. SEM, SD) and the test used to calculate p-values is indicated in the respective figure legends (also for potential EV figures and all those in the final Appendix). Please also check that all the p-values are explained in the legend, and that these fit to those shown in the figure. Please provide statistical testing where applicable. Please avoid the phrase 'independent experiment', but clearly state if these were biological or technical replicates. Please also indicate (e.g. with n.s.) if testing was performed, but the differences are not significant. In case n=2, please show the data as separate datapoints without error bars and statistics. See also: <http://www.embopress.org/page/journal/14693178/authorguide#statisticalanalysis>

9) Please add scale bars of similar style and thickness to all the microscopic images, using clearly visible black or white bars (depending on the background). Please place these in the lower right corner of the images themselves. Please do not write on or near the bars in the image but define the size in the respective figure legend.

10) Please also note our reference format:

12) We now use CRediT to specify the contributions of each author in the journal submission system. CRediT replaces the author contribution section. Please use the free text box to provide more detailed descriptions and do not provide your final manuscript text file with an author contributions section. See also our guide to authors:

<https://www.embopress.org/page/journal/14693178/authorguide#authorshipguidelines>

13) We would encourage you to use 'Structured Methods', our new Materials and Methods format. According to this format, the Materials and Methods section should include a Reagents and Tools Table (listing key reagents, experimental models, software and relevant equipment and including their sources and relevant identifiers) followed by a Methods and Protocols section in which we encourage the authors to describe their methods using a step-by-step protocol format with bullet points, to facilitate the adoption of the methodologies across labs. More information on how to adhere to this format as well as downloadable templates (.doc or .xls) for the Reagents and Tools Table can be found in our author guidelines (section 'Structured Methods'):

14) Please add up to five keywords to the title page order the manuscript sections like this, using these names:
Title page - Abstract - Keywords - Introduction - Results - Discussion - Materials and Methods - Data availability section - Acknowledgements - Disclosure and Competing Interests Statement - References - Figure legends - Expanded View Figure legends

I look forward to seeing a revised version of your manuscript when it is ready. Please let me know if you have questions or comments regarding the revision.

Yours sincerely,

Referee #1:

The authors report that NET-associated RNA (naRNA) drove further NET formation in naive PMNs involving a TLR8-NLRP3-caspase-1-gasdermin D-dependent inflammasome pathway. Moreover, keratinocytes responded to naRNA with expression of psoriasis-related genes (IL17, IL36) via atypical NOD2-RIPK signalling. Moreover, naRNA drove skin inflammation in vivo that was drastically ameliorated by genetic ablation of RNA sensing. The naRNA-LL37 complex was pre-stored as 'DAMP signal' in neutrophil granules and its activity after release was transient and self-limiting.

The manuscript lacks novelty, as most of the main points in this report have been previously published. For instance, Lande et al. demonstrated in Nature 2007 that the cathelicidin peptide (LL37) forms complexes with extracellular human RNA, and it was elucidated by Ganguly et al. in J. Exp. Med. 2010 that LL37, through TLR8, induces self-activation. Furthermore, the concept of extracellular RNA as a potential activator of NETs was previously documented by Smolarz et al. in Front. Cell Infect. Microbiol. 2021. Additionally, the manuscript fails to cite prior work, which reported NETs activating the NLRP3 inflammasome (Hu et al., Arthr. Res. Ther. 2019). It seems that the authors show bias in their selection of references, neglecting key publications directly related to this topic. Those uncited references should have been incorporated into the manuscript and discussed thoroughly. Furthermore, there is a tendency to reference review articles instead of original research articles. Regarding the few novel points presented in this manuscript, it is essential to note that the data presented are preliminary and do not substantiate conclusions or the title of the manuscript. A more detailed comments are provided below.

Major points:

1. Figure 1B is not described in the manuscript.
2. Human neutrophils were treated with 600 nM PMA for 3-4 hours to induce NETs. In the original publication (Brinkmann et al., Science 2004), the authors used 25 nM PMA and observed DNA release within 5-30 min. Is RNA extrusion requiring higher PMA concentration and extended incubation time? If so, the authors should have performed PMA concentration and time-dependent experiments (Fig. 1C, and throughout the manuscript where PMA stimulation was performed).
3. According to Fig. 1D, the NET associated RNA (naRNA) is equal to total cell RNA (whole RNA). It is plausible that neutrophils could extrude 100% of total neutrophil RNA unless the cells are completely lysed due to excessive PMA concentration and extended incubation time.
4. The RNA dye (Syto RNAselect kit) also binds to DNA, according to information of the manufacturer. Therefore, the experiment shown in Suppl. Movie S2 should have been performed in the presence of DNase I in order to rule out the presence of DNA as part of extracellular traps.
5. Some images contain far less cells (Fig. H upper panel and S2D-E) to be conclusive. The scale bar should be given for all microscopic images.
6. The NF- κ B results shown in Figure 2A have low statistical significance. Considering that the experiments are performed in an artificial set up (TLRs over-expression in HEK293 cells), one would expect more clear differences.
7. How were the cells stained in Fig. 1B and 1D? If naRNA or ssRNA were used as stimuli, the dye that sustained the nucleic

acids should have depicted the residual RNA. To quantify the DNA/RNA association to NETs (Fig. 2C, 2E and 2F), the authors should have collected the supernatant of the stimulated human neutrophils and measured the nucleic acid concentration.

8. The images for WT and *tlr3*^{-/-} neutrophils (Fig.2D) treated with "Mock NETs" or "PMA NETs" contain much less cells compared to PMA-treated images. One could depict images with comparable numbers of cells per image.

9. The authors relied mainly on the effect of inhibitors to make their conclusion regarding the involvement of "NLRP3-Caspase-1-gasdermin D pathway" for naRNA induced NET formation. More specifically, the authors used disulfiram (inhibitor of pyroptosis by blocking gasdermin D pore formation) to block the IL-1 β cytokine production using ssRNA + LL37 (Fig. 2G) and concluded that "a unique TLR8-NLRP3-caspase-1-gasdermin D-dependent inflammasome pathway" is involved for the naRNA induced NET formation. Recently, Stojkov et al. (Sci. Signal. 2023) published that GSDMD-knockout neutrophils are capable of NET formation. The same is true for the usage of Cl-amidine (PAD4 inhibitor). Not all stimuli, known to form NETs, require PAD4 (Fig. 2C) (Kenny et al. Elife 2017).

10. In Fig. 2I, instead of using a genetically modified macrophage-like cell line (BlaER1), the authors could have used bone marrow derived macrophages from WT or TLR8^{-/-} or UNC93B1^{-/-} mice.

11. In Fig. 3B-G and 3J-M as well in Fig. S4G, H, I, P, the results of qPCR are shown as "rel...(fold change)". It is not clear to what value this quantification was correlated.

12. The modulation of the immune response upon LL37-RNA had been previously reported (Kato et al. Inflammation, 2023 and Adase et al. J. Biol. Chem., 2016), making the results shown in Fig. 3 partially redundant.

13. The results shown in Fig. 4 lacks any evidence for in vivo NET formation upon intradermal injection of PMA-NETs in the ears of the mice. Therefore, the induction of inflammation is not properly shown and does not support the conclusion of this manuscript.

14. The data in Fig. 5 does not follow the trend of the story in the manuscript. For accurate colocalization of RNA with LL37, it would be more appropriate if the authors use the metabolically labelled primary human hematopoietic stem cells (HSCs) PMNs with 5-EU (as shown in Suppl. Fig. S1D). Immunofluorescent antibody staining in granulocytes could cause nonspecific staining. The isotype control antibody should be included in the experiments shown in Fig. 5.

Referee #2:

Bork et al present the interesting hypothesis that RNA is a major immunostimulatory component of NETs. Since neutrophils and NETs are implicated in many diseases, this finding has important implications for regulation of inflammation.

The authors provide some compelling evidence on presence of RNA in NETs and on the ability of naRNA to activate a variety of cells. One of the strengths of the report is the use of multiple genetic models to reconstruct a potential pathway. The idea of 'composite DAMPs' that are self limiting is an appealing and useful concept.

However some major issues are unaddressed:

My main concern is PMA carryover, specially since this report uses an exceptionally high PMA concentration to induce NETs (50-100 nM is more customary). Although the authors include some good controls, it is essential to repeat this with NETs induced by other stimuli, to prove that sensing of naRNA occurs without use of PMA. The most commonly used stimulus in NET research (after PMA) is calcium ionophore, which should be tested. Alternatively nigericin or ConA-induced NETs. Do these also trigger NETs and macrophage and keratinocyte activation?

Second, 'mock NETs' should be treated with PMA after they are scraped, followed by the same number of washes and this should be tested as negative control.

Neutrophils are very sensitive to chemical inhibitors and treatment often induces apoptosis. Since apoptosis and NETosis are alternate pathways, triggering of apoptosis may nonspecifically block NETs. For instance, Cl amidine is known to trigger apoptosis. All inhibitors should be tested in an apoptosis assay, eg annexin V binding, in neutrophils, at the concentration at which they are used in the manuscript (and in the same medium). This is particularly important for the TLR8 inhibitor.

Is naRNA signaling dependent on de novo transcription? This is an intriguing possibility, since PMA strongly activates transcription in neutrophils (before NETs are released). Transcriptional inhibitors do not block NETs (PMID: 27310721) so this can be tested with actinomycin or similar RNA pol inhibitor.

Figure 4: have these results been repeated or is this a single experiment? Number of repeats should be indicated for all in vivo experiments

Figure 4D-F: this does not directly test the role of naRNA, simply tests the requirement of *tlr13*. There could be various other agonists. As currently written, the section is overinterpreted.

Figure 5: the idea of pre-packaging of RNA in granules is very unexpected. The evidence provided here is not sufficient for this claim. Firstly, it would be useful to see individual channels in the microscopy images, to assess RNA subcellular localization. As currently presented, it is difficult to see granular RNA localisation. Furthermore another reagent is needed to prove that this is not

an artefact of the anti-rRNA Y10b antibody. The TEM evidence provided is also rather poor. More compelling evidence is needed for this claim and additional methods, potentially RNA-protein pulldown, RNA FISH combined with IF, or other suitable technique.

Referee #3:

Authors explore the role of NET associated RNA (naRNA) as a component of neutrophil extracellular traps (NETs) and its immunostimulatory properties in the skin.

Summary of paper:

1. naRNA in NETs: The paper establishes that naRNA is a common component of NETs formed by human polymorphonuclear neutrophils (PMNs) in response to various stimuli, including PMA, LL37-RNA complexes, nigericin, and live pathogens like *Candida albicans*. This is confirmed through confocal microscopy and metabolic labeling experiments.
2. naRNA as a DAMP: The study demonstrates that naRNA serves as a potent damage-associated molecular pattern (DAMP) that can trigger NET formation in primary PMNs. It is proposed that naRNA, in conjunction with LL37, acts as a 'composite DAMP' to activate neutrophils.
3. RNA Sensors and Inflammasome Activation: The research identifies the involvement of RNA sensors TLR8/Tlr13 and the NLRP3 inflammasome in naRNA-mediated NET formation.
4. Immune Cell Activation: The paper shows that naRNA can activate not only neutrophils but also other immune cells and these responses are mediated by RNA sensing.
5. naRNA in Keratinocytes: The study demonstrates that naRNA can activate keratinocytes, leading to the induction of psoriasis-associated genes. This indicates the potential role of naRNA in skin inflammation.
6. In Vivo DAMP Activity: The research extends to in vivo models, where intradermal injection of RNA-stabilized PMA NETs in mice induces skin inflammation. Tlr13-deficient mice show reduced skin inflammation in a psoriasis model.
7. Pre-Association of RNA and LL37: Pre-association of RNA and LL37 within neutrophil compartments occurs before NET extrusion, indicating that neutrophils contain preformed naRNA-LL37 DAMPs.

This is a hard-to-read manuscript mainly due to a poor language style. However, the main limitation of this manuscript is that authors are looking at only one side of biology and that is protected RNA, while they should also prove their concept with RNase-free NETs.

COMMENTS

Major Points:

1. Is the pre-association of RNA and LL37 sufficiently shown? The data of Figure 5C show the colocalization to the granule compartment but do not show a bound complex. To convince, please show a control granule protein of which you know it is not associated with RNA. Or precipitate the RNA/LL37 complex from isolated granules.
2. How is RNA bound to DNA? Via LL37? Figure 1 shows that RNA is associated with DNA strands.
3. The entire manuscript is built on the concept that effects of RNA are shown in the presence of RNase inhibitor. Authors need to complement their experiments by introducing a system in which the effects are abrogated in the absence of RNA. For example, in the animal experiments, authors should inject PMA NETs + RNase vs PMA NETs alone.
4. Regarding your quantification of NETs DNA and RNA, you mention quantification of NETs DNA and RNA in immunofluorescence images but do not explain the units used, such as "rel. ROI number/ROI average size." There's a need for clarification regarding the calculation and presentation of ROIs, and an example of ROIs with and without NETs should be provided.
5. Figure 1G is supposed to complement Figure 1E, but different methods are used to calculate and show the presence of NETs. The paper should present one or both analysis methods for both experiments. Please, standardize your dispersion versus tile counting methodology.
6. To support the claim that naRNA, and not DNA, is the relevant immunostimulatory component for NETs, the paper should include experiments where PMA NETs are treated with DNase and DNase inhibitors.
7. While the in vivo experiments are interesting, they mainly focus on the presence and absence of extracellular RNA rather than NETs-associated extracellular RNA. The paper should address the potential effects of RNase inhibitors on inflammation in an in vivo setting. Although you show 'RNase inhibitor+RPMI' you do not show the RNase inhibitor in the context of inflammation that is not driven by NETs.
8. In Figure 3 pre-digestion of NETs with RNase A abolished the stimulatory effects of NETs on keratinocytes, whereas pre-digestion with DNase I did not. It is hard to believe that the stimulatory effect of NETs on keratinocytes is abrogated by RNase but not by DNase. NETs are DNA scaffolds. If you break them down, the associated RNA should be affected. Please, show in the supernatants the size of DNA fragments (note comments 2 and 12).
9. The paper should address whether RNase treatment affects the degradation of keratinocyte RNA in addition to NET RNA. A key point you make is to show the different stimulatory potential between RNase and DNase treated NETs. In Figure 3b you show that RNase but not DNase abolishes the stimulatory effects of NETs on keratinocytes, however your readout is using

qPCR to quantify the RNA of the psoriasis-associated genes. How do you rule out whether the RNase also degrades the keratinocyte, psoriasis associated RNA? Please, show the protein production in another experiment.

10. When discussing the differences in ear thickness and epidermal thickness in wild-type and Tlr13 KO animals, it would be important to include images of the ears from each group for visual reference.

11. Are RNA-LL37 complexes released during degranulation, outside of NETosis?

12. With regards to fresh and aged/degraded NETs I do not see any characterization of your fresh and aged NETs with serum or medium. What happens to the DNA, RNA and proteins after incubation? Serum contains RNases, DNases, and proteases which all may play a role here.

13. Simplify your language, e.g. avoid purposefully and accidentally and barbed wire roadblocks, etc. Along the same lines, the manuscript title needs to be rephrased.

14. The paragraph discussing the trade-off between DAMP activity and RNA's short-lived nature should be revised or removed from the results section, as it is more of a discussion point.

15. The graphical abstract is not clear.

16. Reference to immune tolerance is not subject of this work.

We thank all referees for the time and consideration they have given to our work and would like to address their comments here.

Point-by-point reply EMBOR-2023-58048-T

Please note that Supplemental figures have been renamed EV figures, i.e. Fig. EV1 is now Fig. EV1 in the revised text version.

Referee #1

The authors report that NET-associated RNA (naRNA) drove further NET formation in naive PMNs involving a TLR8-NLRP3-caspase-1-gasdermin D-dependent inflammasome pathway. Moreover, keratinocytes responded to naRNA with expression of psoriasis-related genes (IL17, IL36) via atypical NOD2-RIPK signaling. Moreover, naRNA drove skin inflammation in vivo that was drastically ameliorated by genetic ablation of RNA sensing. The naRNA-LL37 complex was pre-stored as 'DAMP signal' in neutrophil granules and its activity after release was transient and self-limiting.

Comment #1: The manuscript lacks novelty, as most of the main points in this report have been previously published. For instance, Lande et al. demonstrated in Nature 2007 that the cathelicidin peptide (LL37) forms complexes with extracellular human RNA, and it was elucidated by Ganguly et al. in J. Exp. Med. 2010 that LL37, through TLR8, induces self-activation.

Author reply #1: We concede the point (and state so in the introduction) that RNA has been known to interact with LL37 for TLR activation. However, this work was mainly done with synthetic RNA and natural sources of RNA have not been described in these seminal papers. Our paper described NETs as a plausible and abundant source of LL37 complexing (na)RNA as a TLR activator. The highlight of our work is that this novel source leads to a self-propagating activation loop. Finally, the previous work by Gilliet et al mentioned by referee 1 proposed that RNA-mediated activation of TLRs via LL37 complexation is an "accidental" break of self-nucleic acid tolerance, see (Lande, Chamilos et al. 2015). Instead, we provide compelling evidence that neutrophil RNA-LL37 complexes are abundant composite DAMPs that are configured to activate TLRs, rather than

combine accidentally. This is an entirely new concept. Interestingly, we presented this concept to Michel Gilliet himself at the recent Novel Concepts in Innate Immunity 2023 () meeting, where he was an invited speaker. He was highly intrigued and fascinated by our concept and confirmed it as entirely novel and ground-breaking. Hence, lack of novelty should not be a concern.

Comment #2: Furthermore, the concept of extracellular RNA as a potential activator of NETs was previously documented by Smolarz et al. in *Front. Cell Infect. Microbiol.* 2021.

Author reply #2: That *Candida* leads to NET formation has been known for years and the primary focus of this paper is that *Candida* biofilms contain and release (fungal) nucleic acids that can trigger NET formation from human neutrophils. Smolarz and colleagues go on to delineate the pathway and implicate TLR8. Although this is interesting and fits well with our observation in Fig. 1A, this work does not address or compromise the points made in our work: 1) the authors do not investigate host RNA like we do; 2) they fail to investigate if *Candida*-triggered NETs cause further self-amplification; 3) they do not explore the connection between NET RNA and LL37; 4) They do not show any in vivo relevance for their pathway. Thus, the referee's comment is correct but is unrelated to our work.

Comment #3: Additionally, the manuscript fails to cite prior work, which reported NETs activating the NLRP3 inflammasome (Hu et al., *Arthr. Res. Ther.* 2019).

Author reply #3: NETs and inflammasomes are very active research fields and we sincerely apologize if we have overlooked important work. The referee here cites a clinical study in which cell-free DNA and NET-DNA complexes were quantified in sera of healthy controls and patients suffering from Adult Onset Still's disease. Hu and colleagues did check the activation of the inflammasome by NETs, however, this was only done in immortalized THP-1 cells (i.e. macrophage-like cells), not primary neutrophils. As THP-1 cells, or macrophages in general, do not release RNA and LL37, they do not make any contribution to the concept of NET propagation, whereas neutrophils do. NLRP3 has also been reported to respond to hundreds of stimuli but only for a few a precise pathway has been proposed. As TLR8 was also not implicated by Hu et al., the pathway in this work remains obscure and does not compromise the novelty of our data.

Comment #4: It seems that the authors show bias in their selection of references, neglecting key publications directly related to this topic. Those uncited references should have been incorporated into the manuscript and discussed thoroughly. Furthermore, there is a tendency to reference review articles instead of original research articles.

Author reply #4: In the same way as with the last comment, we apologize for any unintentional bias and are open to suggestions by the referees and editor. Unfortunately, no additional specific references were suggested by referee 1, whilst the ones stated in the preceding comments were discussed (and most already cited). We were also not aware of a strong bias of original papers vs reviews/book chapters: the ratio in our originally submitted paper was 42 original papers to 21 reviews, which appears not unbalanced towards reviews. Moreover, we hold that for summarizing

vast areas of research (as is the case for the NET field), citing review articles is plausible and common practice. We therefore do not think that this comment should be taken as a general substantial criticism of our work and hope the abovementioned figures reassure the referee and Editor.

Comment #5: Regarding the few novel points presented in this manuscript, it is essential to note that the data presented are preliminary and do not substantiate conclusions or the title of the manuscript. A more detailed comments are provided below.

Author reply #5: As pointed out above, leading experts attribute a high degree of novelty to our work, which is also confirmed by the other two referees. Moreover, most of the work includes at least 3 repeats and multiple donors and animals so that, technically speaking, the work is not preliminary but built on solid data that was statistically tested. Microscopy was always quantified. We have sought to address all specific criticisms referee 1 made in the following but disagree with the general statement that the work in general is preliminary and unsubstantiated biologically and conceptually.

Comment #6: Major points. 1. Figure 1B is not described in the manuscript.

Author reply #6: Even if correct, a missing callout to a figure should not be regarded as a major critical point. In fact, the criticism does not apply as Figure 1B was described in Line 125 (see underline) as follows "(Fig. 1A, quantified in B; control of antibody staining in Fig. EV1A, B). To avoid this misunderstanding in future, we have amended the phrasing to "quantified in Fig. 1B".

Comment #7: 2. Human neutrophils were treated with 600 nM PMA for 3-4 hours to induce NETs. In the original publication (Brinkmann et al., Science 2004), the authors used 25 nM PMA and observed DNA release within 5-30 min. Is RNA extrusion requiring higher PMA concentration and extended incubation time? If so, the authors should have performed PMA concentration and time-dependent experiments (Fig. 1C, and throughout the manuscript where PMA stimulation was performed).

Author reply #7: Although PMA > 25 nM has been used in multiple other publications (e.g. (Lim, Kuiper et al. 2011, Remijsen, Vanden Berghe et al. 2011, Parker, Dragunow et al. 2012, Masuda, Shimizu et al. 2017, Neubert, Senger-Sander et al. 2019, Arpinati, Shaul et al. 2020), we agree that this could be viewed as a concern. However, our goal was not to establish stimulation conditions for PMA but to use PMA to robustly generate NETs. 600 nM for several hours seemed suitable to ensure full NETosis in the entire treated PMN population. Nevertheless, to address the comment we generated NETs with 600 nM vs 60 nM PMA, harvested them as before and transferred them to naïve neutrophils. Biological triplicates were performed and quantified. NETs alternatively generated with the ionophore A23187 (suggestion by referee 2) were also analyzed in a similar way. As shown in the reviewer figure below, there is no substantial difference between 600 nM (labeled PMA NETs for consistency with the remaining manuscript) or 60 nM PMA or A23187 NETs, alleviating the reviewers concern (differences in morphology between different fields of view

and/or stimuli are normal). Additionally, 600 nM PMA added to Mock NETs and then washed away before harvesting (also suggested by referee 2), does not lead to activation, as expected by us, excluding carry-over of PMA in PMA NETs. Moreover, it seems important to note that PMA is used only as a tool to generate NETs whereas all the remaining work is done with NETs themselves to observe biological effects. Here PMA is used merely as a positive control, as is common practice. It is also vital to note that all PMA NET effects are truly RNA receptor dependent, whereas PMA alone is not TLR sensitive. This confirms further that carry-over or the PMA concentration used is not relevant **to** the conclusions drawn. As it may lead to confusion, we decided not to include this data in the revised manuscript but have added a comment about PMA concentration to the methods section.

A human PMNs

B

Confocal microscopy of primary human PMNs treated as indicated (A) and quantification thereof (B). Stimuli are as follows:

- unstimulated
- PMA directly (600 nM)
- Mock NETs prepared as before
- PMA NETs prepared as before (i.e. induced using 600 nM PMA)
- Mock NETs exposed to 600 nM PMA before harvesting
- PMA NETs generated with 60 nM PMA
- A23187 NETs (5 μ M) (

All NET preparations were used at 1:100 dilution) stimulated for 3 h. Cells were then stained for DNA (Hoechst 33342, white, n = 1 donor, representative images, scale bar 10 μ m). (B) Quantification of (A) using DNA (Hoechst 33342) signal to quantify NET formation (n = 1 donor, mean+SD, each dot represents the number of NET-positive tiles in one image quantified from three images/condition).

Comment #8: 3. According to Fig. 1D, the NET associated RNA (naRNA) is equal to total cel. RI RNA (whole RNA). It is plausible that neutrophils could extrude 100% of total neutrophil RNA unless the cells are completely lysed due to excessive PMA concentration and extended incubation time.

Author reply #8: We are unsure why the referee’s conclusion is that naRNA and cellular whole RNA are equal as we think the data make exactly the opposite point: In Fig. 1D (below) the composition of naRNA is not identical to that of whole RNA. For example, whereas whole RNA is dominated by rRNA, rRNA is more balanced with e.g. ncRNA or mRNA in naRNA. This is exactly an argument against the referee’s concern that PMA would lead to excessive lysis. We are therefore not sure how to interpret the referee’s concern. In any case, even if (hypothetically speaking) 100% of neutrophil RNA ended up in the NETs, this would still qualify as NET-associated RNA and would not change any of the main conclusions related to naRNA.

Fig. 1D from paper.

Comment #9: 4. The RNA dye (Syto RNaselect kit) also binds to DNA, according to information of the manufacturer. Therefore, the experiment shown in Suppl. Movie S2 should have been performed in the presence of DNase I in order to rule out the presence of DNA as part of extracellular traps.

Author reply #9: As DNA is a primary structural feature of NETs shown in countless publications, we do not understand the plausibility of “ruling out” the presence of DNA in NETs. The

manufacturer (ThermoFisher) states the following: “The SYTO RNaselect green fluorescent cell stain is a cell-permeant nucleic acid stain that selectively stains RNA. Although virtually nonfluorescent in the absence of nucleic acids, SYTO RNaselect stain exhibits bright green fluorescence when bound to RNA (absorption/emission maxima ~490/530 nm), but only a weak fluorescent signal when bound to DNA”. We concede that this does not entirely exclude binding to DNA. Although there may be superior dyes emerging (albeit not commercially available), multiple studies have relied on (or used as a bench-mark) SYTO RNaselect for meaningful and satisfactory results but have not raised concerns about specificity, e.g. (Li, Kim et al. 2006, Zhou, Liu et al. 2015, Herster, Bittner et al. 2020, Wu, Liu et al. 2020). Moreover, if the majority of SYTO RNaselect signal was due to DNA binding, we would expect the signals to overlap. As we show below for a representative line plot, this is not the case. This data has been added to the revised manuscript in the **new Fig. EV5C-E**. Finally, to avoid any doubt about the specificity of SYTO RNaselect, we mostly used anti-rRNA staining throughout and even implemented RNA click-chemistry labeling for further verification. We contend this is a relatively rigorous way of testing and controlling specificity. Please also see **Author reply #29**.

New Fig. EV5E,F: Confocal microscopy and line plots of unstimulated primary human PMN. Cells were stained for DNA (Hoechst 33342, white), RNA (SYTO RNaselect, magenta) and LL37 (antibody staining, yellow) and intensity levels quantified using line plots.

Comment #10: 5. Some images contain far less cells (Fig. H upper panel and S2D-E) to be conclusive.

Author reply #10: In unstimulated conditions there are typically less cells as unstimulated neutrophils do not adhere very strongly to the coverslips and therefore some cells may be lost during preparation of the slips for antibody staining. This is a typical technical limitation that cannot be avoided unless far more cells are seeded in the unstimulated condition. However, we would be

concerned the greater cell density during the stimulation period would falsify results as it could lead to accidental activation. Additionally, after NET induction it is nearly impossible to identify the number of cells that were imaged retrospectively as demarcating features are completely lost. In general, single images are never conclusive therefore multiple images were always acquired, typically in 3x3 tiles, and quantified in an unbiased manner. We therefore opted for seeding and stimulating the same number of cells per condition (which is common practice), imaging larger fields of view with identical objective/zoom and then quantifying from multiple experiments and images per experiment. Please see also **Author reply #25**.

Comment #11: The scale bar should be given for all microscopic images.

Author reply #11: We apologize for the confusion regarding scale bars and that, indeed, in some images scale bars were missing (e.g. Fig. 1F and 5C). Typically, a scale bar was given for at least one image of a given series and applies to all other images of that series. As all images in one series were acquired with identical objectives and zoom, it is not necessary to show a scale bar in each image but we have now clearly stated in the methods that a given scale bar applies to all images of this series. We have clarified this point in the relevant methods sections.

Comment #12: 6. The NF- κ B results shown in Figure 2A have low statistical significance. Considering that the experiments are performed in an artificial set up (TLRs over-expression in HEK293 cells), one would expect more clear differences.

Author reply #12: We appreciate the referee's concern but according to our statistical knowledge, for $\alpha = 5\%$ (which is common practice), $p < 0.05$ is significant and it is informative but not meaningful how much smaller the p-value is. The differences marked with asterisks in Fig. 2A are therefore unrefutably statistically significant. Maybe the reviewer is concerned about the biological significance. Here we concede that the differences may be smaller than expected by the referee; nevertheless, and taking statistical significance as a meaningful measure, we contend that there are real differences between different TLRs pointing to RNA being the decisive stimulant contained in NETs. We agree with the referee that HEK293T cells are an artificial, yet well-accepted system, in the TLR field. That is also why most of our work is done with primary PMN and keratinocytes.

Comment #13: 7. How were the cells stained in Fig. 1B and 1D? If naRNA or ssRNA were used as stimuli, the dye that sustained the nucleic acids should have depicted the residual RNA.

Author reply #13: In Fig. 1B and S1B a quantification of 1A is shown in which naRNA was stained using Hoechst 3334 for DNA (white) and anti-rRNA Y10b for naRNA (magenta). This antibody does not stain ssRNA40, the short synthetic RNA used for stimulation. Fig. 1D is RNAseq not based on staining, so we are not sure if the reviewer refers to this panel or rather Fig. EV1D? In Fig. EV1D the incorporated 5-ethynyluridine (5-EU) was click-labeled with a fluorescent dye (yellow, total RNA), combined with staining for naRNA (anti-rRNA Y10b, magenta) and DNA (Hoechst 33342, white). This information is provided in the figure legend. We apologize but are not sure what is meant by

the remaining comment/question. We are happy to comment if the question is re-phrased more precisely.

Comment #14: To quantify the DNA/RNA association to NETs (Fig. 2C, 2E and 2F), the authors should have collected the supernatant of the stimulated human neutrophils and measured the nucleic acid concentration.

Author reply #14: Quantification was attempted; however, as NETs are very heterogenous and remain attached to the cover slip, quantification in the supernatant is not reliable.

Comment #15: 8. The images for WT and *tlr3*^{-/-} neutrophils (Fig.2D) treated with "Mock NETs" or "PMA NETs" contain much less cells compared to PMA-treated images. One could depict images with comparable numbers of cells per image.

Author reply #15: We presume with Tlr3 the referee refers to Tlr13? Regarding the quantification of unstimulated cells, please see **Author reply #10**.

Comment #16: 9. The authors relied mainly on the effect of inhibitors to make their conclusion regarding the involvement of "NLRP3-Caspase-1-gasdermin D pathway" for naRNA induced NET formation. More specifically, the authors used disulfiram (inhibitor of pyroptosis by blocking gasdermin D pore formation) to block the IL-1b cytokine production using ssRNA + LL37 (Fig. 2G) and concluded that "a unique TLR8-NLRP3-caspase-1-gasdermin D-dependent inflammasome pathway" is involved for the naRNA induced NET formation. Recently, Stojkov et al. (Sci. Signal. 2023) published that GSDMD-knockout neutrophils are capable of NET formation. The same is true for the usage of Cl-amidine (PAD4 inhibitor).

Author reply #16: Indeed, not all stimuli, that are known to induce NET formation, require PAD4 (Fig. 2C) and it is now very clear that NET formation can be dependent and independent of PAD4 (Kenny, Herzig et al. 2017). Since naRNA is found in all types of NETs, whether the process is PAD4-dependent or not seems of lesser importance and hence the specificity of inhibitors was not discussed (but the abovementioned reference added to the revised version). Similarly, contradictory data have been published for GSDMD: those implicating GSDMD (Sollberger, Choidas et al. 2018, Silva, Wanderley et al. 2022) vs showing GSDMD redundancy (Stojkov, Claus et al. 2023). It was therefore outside the scope of solving these issues but since all inhibitors used are very well accepted, we chose to remain silent about these issues and consider it legitimate to draw the conclusions we make for the pathway we studied.

Comment #17: 10. In Fig. 2I, instead of using a genetically modified macrophage-like cell line (BlaER1), the authors could have used bone marrow derived macrophages from WT or TLR8^{-/-} or UNC93B1^{-/-} mice.

Author reply #17: The referee's suggestion is valuable. Unfortunately, TLR8 is dysfunctional in BL6 but replaced by Tlr13 (Eigenbrod and Dalpke 2015). We used *Tlr13* and *Unc93b1* KO cells for neutrophil analysis (Fig. 2D, 4C, 4E and S3, respectively). Since neutrophils and the human system were our main focus and the RNA receptor was obvious in multiple systems (using both inhibitors

and KO cells), we did not perform further experiments in BMDM as this offered limited addition to the story.

Comment #18: 11. In Fig. 3B-G and 3J-M as well in Fig. EV4G, H, I, P, the results of qPCR are shown as "rel...(fold change)". It is not clear to what value this quantification was correlated.

Author reply #18: Typically the fold change is relative to "untreated", unless stated otherwise, and we apologize for any omission. We have checked each legend.

Comment #19: 12. The modulation of the immune response upon LL37-RNA had been previously reported (Kato et al. *Inflammatio*, 2023 and Adase et al. *J. Biol. Chem.*, 2016), making the results shown in Fig. 3 partially redundant

Author reply #19: We thank the referee and have added Kato et al 2023 in the discussion as this may be important. One has to bear in mind that this paper only appeared in June 2023. Our paper was initially put on Biorxiv in July 2022 and then submitted to EMBO Molecular Medicine on 28 June 2023, from where it was transferred to EMBO Reports. Thus, we were not aware of it. At first glance, the paper seems to overlap with our work as the title suggests. However, their focus is on double-stranded RNA working with LL37. That LL37 can work with multiple RNA types is not surprising since interactions are charge-mediated. However, host RNA is rarely double-stranded so that NETs, which contain multiple ssRNA types (Fig. 1D), most likely operate via a different route than the one described by Kato et al. This is suggested by our findings that naRNA sensing is mediated by NOD2 and not RIG-I (ruled out using MAVS siRNA, see Fig. EV4M) as in Kato et al. From the TLR field it is well known that dsRNA and ssRNA are sensed differently by TLR3 vs TLR7/8, so that mechanisms differ, and the differences between NOD2 and RIG-I regarding downstream signaling are certainly greater than amongst different TLRs. Moreover, Kato and colleagues only used poly(I:C), a synthetic dsRNA analog, which lacks any physiological relevance. Our paper, conversely, uses naturally occurring naRNA. Adase JBC 2016 is of course a well-known paper by the Gallo group. Their focus is also on dsRNA and in most of the experiments the authors also rely exclusively on poly(I:C). In a selection of experiments, they use U1 ncRNA, which they suppose can contain dsRNA stretches. They confirm a similar activity as poly(I:C) but do not actually prove that in their hands U1 contains dsRNA. So, most of the claims in both papers are based on poly(I:C) again. Additionally, they report entirely different receptors. We therefore have to strongly disagree with the notion that these two studies make our data in Fig. 3 redundant.

Comment #20: 13. The results shown in Fig. 4 lacks any evidence for in vivo NET formation upon intradermal injection of PMA-NETs in the ears of the mice. Therefore, the induction of inflammation is not properly shown and does not support the conclusion of this manuscript.

Author reply #20: We thank the referee for this comment. We here rely on earlier work, which is cited, showing that IMQ treatment and RNA+LL37 injections lead to NET formation in vivo (Herster, Bittner et al. 2020). We have gone back to the tissue samples and analyzed the IMQ-treated samples for NETs. As shown in the new Fig. 4G and 4H, there is significantly less staining for CitH3,

an accepted marker for NETs in tissue (e.g. (Suzuki, Tsuchiya et al. 2022), confirming that in the IMQ mouse model the lack of RNA sensing restricts NET formation otherwise seen in WT animals. We hope that this additional data addresses the reviewers concern.

Comment #21: 14. The data in Fig. 5 does not follow the trend of the story in the manuscript. For accurate colocalization of RNA with LL37, it would be more appropriate if the authors use the metabolically labelled primary human hematopoietic stem cells (HSCs) PMNs with 5-EU (as shown in Suppl. Fig. EV1D). Immunofluorescent antibody staining in granulocytes could cause nonspecific staining. The isotype control antibody should be included in the experiments shown in Fig. 5.

Author reply #21: As controls were included and shown for many of the other staining experiments (depicted in Fig. EV1A), in our opinion there is no doubt about specificity. As the stem cell approach is extremely cumbersome, we employed it to generally rule out non-specificity but were not able to apply it to all experiments. Moreover, although stem-cell based neutrophils are close to primary cells, we considered it important to work with primary neutrophils to derive the most physiologically relevant data. From all that is known, clearly, the LL37 staining is specific as it is a granular protein (Sorensen, Arnljots et al. 1997). Following the referee's comment, the anti-rRNA staining (for which controls are shown for multiple experiments, see above), has therefore been complemented by SYTO RNaselect staining, which avoids antibodies altogether. Although the referee may not be entirely endorsing this dye (but see **Author reply #9**), the **new Figs. 5D, 5E, EV5C-E** further confirm strong colocalization in granules. Collectively, in our opinion the sum of different analyses supports our conclusions and the descriptions provided put the reader in a position to assess the validity of our claims adequately.

Referee #2

Bork et al present the interesting hypothesis that RNA is a major immunostimulatory component of NETs. Since neutrophils and NETs are implicated in many diseases, this finding has important implications for regulation of inflammation.

Comment #22: The authors provide some compelling evidence on presence of RNA in NETs and on the ability of naRNA to activate a variety of cells. One of the strengths of the report is the use of multiple genetic models to reconstruct a potential pathway. The idea of 'composite DAMPs' that are self-limiting is an appealing and useful concept.

Author reply #22: We thank referee 2 for the enthusiastic feedback.

Comment #23: However some major issues are unaddressed:

My main concern is PMA carryover, specially since this report uses an exceptionally high PMA concentration to induce NETs (50-100 nM is more customary). Although the authors include some good controls, it is essential to repeat this with NETs induced by other stimuli, to prove that sensing of naRNA occurs without use of PMA. The most commonly used stimulus in NET research (after

PMA) is calcium ionophore, which should be tested. Alternatively nigericin or ConA-induced NETs. Do these also trigger NETs and macrophage and keratinocyte activation?

Author reply #23: This is a valid concern which we discussed in **Author reply #7**. As neutrophils are clearly the most sensitive cell type, the fact that they did not respond to PMA-treated Mock NETs is an indication that all the data are dependent on NET RNA, not least because genetic RNA receptor KOs do not respond to NETs any more but do respond to the positive control PMA (at higher concentrations than any carry-over would entail). We therefore did not consider it necessary to repeat the same experiments with macrophages or keratinocytes. We hope the data shown in the referee figure connected to **Author reply #7** above demonstrates that PMA carryover or concentration are not an issue.

Comment #24: Second, 'mock NETs' should be treated with PMA after they are scraped, followed by the same number of washes and this should be tested as negative control.

Author reply #24: We did this and the results show no indication for a non-specific effect, please see **Author reply #7** and **Author reply #23**.

Comment #25: Neutrophils are very sensitive to chemical inhibitors and treatment often induces apoptosis. Since apoptosis and NETosis are alternate pathways, triggering of apoptosis may nonspecifically block NETs. For instance, Cl amidine is known to trigger apoptosis. All inhibitors should be tested in an apoptosis assay, e.g. annexin V binding, in neutrophils, at the concentration at which they are used in the manuscript (and in the same medium). This is particularly important for the TLR8 inhibitor.

Author reply #25: We agree with the reviewer that different types of cell death can be mutually exclusive, but we are not sure which paper the referee is specifically referring to. To our understanding, apoptosis in neutrophils shows distinct morphological features (e.g. apoptotic blebs, see (Douda, Yip et al. 2014) and uniform, unilobar nuclei, see (Gray, Hardisty et al. 2018)) which we do not observe in unstimulated, inhibitor-treated cells. Moreover, the inhibitors do not block or affect NET formation in direct response to PMA. If, as the referee suggests, the inhibitors truly switch the route of cell death to apoptosis (as in the case of Akt inhibitors, (Douda, Yip et al. 2014)), then we would expect differences between PMA without inhibitors vs PMA with Cu-CPT9a, for instance. But we find no evidence. To alleviate the referee's concern we show below higher magnifications of the images contributing to Fig. 2B showing intact PMN morphology under inhibitor treatment. Again, the inhibitor data is confirmed by plausible genetic data from mouse *Tlr13* KO PMN.

Close-up images of inhibitor-treated unstimulated primary PMN from Fig. 2B (stained for DNA using Hoechst, blue, and naRNA using anti-rRNA, red) showing intact morphology and absence of any apoptosis features (typically apoptotic blebs or uniform, unilobar nuclei). This morphology is representative for all experiments using these inhibitors at various concentrations.

Comment #26: Is naRNA signaling dependent on de novo transcription? This is an intriguing possibility, since PMA strongly activates transcription in neutrophils (before NETs are released). Transcriptional inhibitors do not block NETs (PMID: 27310721) so this can be tested with actinomycin or similar RNA pol inhibitor.

Author reply #26: We thank the referee for this suggestion but are not sure how to interpret the referee's comment: Is the question 1) whether transcription in the NET-producing neutrophils is required to ensure NETs contain stimulatory naRNA? Or is the question 2) whether responding neutrophils require de novo transcription when sensing naRNA, signaling via TLR8-NLRP3 and producing NETs as a result? We presume the former and certainly appreciate the intriguing suggestion. Although use of the suggested inhibitors would be a suitable way to test this in either the producing (or, alternatively sensing neutrophils), the details regarding the production of naRNA (question 1), its packaging or further downstream signaling events in sensing neutrophils (question 2) in our opinion are outside the scope of the present manuscript.

Comment #27: Figure 4: have these results been repeated or is this a single experiment? Number of repeats should be indicated for all in vivo experiments

Author reply #27: Yes, there were multiple biological replicates in each of the experiments shown. We apologize if the information stated in figure legend was not entirely clear. We have checked this is clearly stated in the revised manuscript.

Comment #28: Figure 4D-F: this does not directly test the role of naRNA, simply tests the requirement of *tlr13*. There could be various other agonists. As currently written, the section is overinterpreted.

Author reply #28: We concede the point based on the previous data and have amended the interpretation. We did analyze skin samples from IMQ-treated WT and *Tlr13* KO mice and observe less NET formation, see **Author reply #20**. Thus, NET propagation seems affected in *Tlr13* KO. Nevertheless, we have scrutinized this section to avoid overinterpretation of the data.

Comment #29: Figure 5: the idea of pre-packaging of RNA in granules is very unexpected. The evidence provided here is not sufficient for this claim. Firstly, it would be useful to see individual channels in the microscopy images, to assess RNA subcellular localization. As currently presented, it is difficult to see granular RNA localization. Furthermore another reagent is needed to prove that this is not an artefact of the anti-rRNA Y10b antibody. The TEM evidence provided is also rather poor. More compelling evidence is needed for this claim and additional methods, potentially RNA-protein pulldown, RNA FISH combined with IF, or other suitable technique.

Author reply #29: We refer to **Author reply #21** and are happy to provide individual channels (below) and have also included SYTO RNaselect staining to avoid antibody staining (please see **new Fig. 5D, E and S6**). Furthermore, orthogonal projections by extensive z-stack analysis and 3D reconstructions were added:

New Fig. 5D: Immunofluorescence analysis of unstimulated PMN stained with SYTO RNaselect staining combined with anti-LL37 staining and line plot analysis. Further examples are shown in Fig. EV5C.

New Fig. 5E: Orthogonal projections of unstimulated PMN stained with SYTO RNaselect staining combined with anti-LL37 staining. 3D reconstruction shown in new Fig. EV5E.

New Fig. EV5E: 3D reconstruction of z-stack analysis of unstimulated PMN stained with SYTO RNaselect staining combined with anti-LL37 staining.

Merged channels	LL37	RNA	DNA

Single channel images for examples of unstimulated PMN stained with SYTO RNaselect and anti-LL37 antibodies.

We understand the reviewer's request for additional experimental verification. Typically, co-localization studies by immunofluorescence and additional dual label EM are considered valid demonstrations. The immunofluorescence on ultrathin sections (50-60 nm, Figure 5D) indicates that both signals come from a very small volume. In Fig. 5E (electron microscopy) RNA and LL37 immuno-gold signals are shown to be within 200 nm proximity. Bearing in mind that unavoidable electrostatic repulsion and steric hindrance related to antibody staining with gold particles (Dulhunty, Junankar et al. 1993, Flechsler, Heimerl et al. 2020) prevent co-labelling of the same physical complex, we consider it practically impossible to demonstrate a direct interaction using this technique but the existence of both ribosomal RNA and LL37 in the same granule, which is consistent with ribosomes becoming obsolete in mature murine neutrophils (Zhu, Gong et al. 2017), at least strongly suggests an interaction, which we more carefully state in the re-submission. Following the referee's comments, we did consider, and partially tested, additional experimental approaches:

- Immunoprecipitation of LL37 and verification of RNA binding: We already considered this, but the experiment cannot be done by simply lysing PMN and pulling down LL37 using antibodies: due to their complementary charge it cannot be excluded that LL37 and neutrophil RNA, both freely diffusible in a lysate, might not interact post-lysis. Therefore, the method of nitrogen cavitation (Udby and Borregaard 2007) would be necessary to purify intact granules and then precipitate LL37 to check for co-immunoprecipitated RNA. As soon as our grant money allows, we will spend €5.000 to acquire a nitrogen cavitation device and establish this method according to existing protocols we have from a former colleague in Tübingen, Prof. Dominik Hartl, an expert on neutrophils who also fully endorses this present work. However, at the present time, no such device is accessible to us in a setting where freshly isolated neutrophils could be fractionated.
- Proximity ligation assay: We have already explored this proximity-based method as an alternative to the immunofluorescence on ultrathin sections presented in Fig. 5F. However, given the size of the 1°+2° antibody sandwich, we would only improve from 200 nm in Fig. 5F to 80 nm (see e.g. (Klasener, Maity et al. 2014)). We explored the method already nevertheless and could see a signal when using the anti-rRNA Y10b and anti-LL37 primary antibodies, in combination with the Duolink® Proximity Ligation Assay Reagents from Merck and subsequent confocal microscopy analysis. However, neutrophils seem to take up the far-red dye provided with the kit and therefore the background by staining reagents-only is high (although unfortunately not typically shown in publications). To show our efforts we append below an interim analysis example. However, this could not be optimized further as reagents were on month-long backorder.

human PMNs

Preliminary confocal microscopy analysis of proximity ligation assay (PLA) of (na)RNA and LL37 in primary human PMN. Unstimulated or PMA (600 nM, 3 h) treated cells were fixed and PLA was performed according to the manufacturer's instructions (Duolink® Proximity Ligation Assay, Merck) with anti-rRNA Y10b (mouse, Santa Cruz Biotechnology, sc-33678) and anti-LL37 (rabbit, LSBio, LS-B6696-500) primary antibodies and subsequent PLUS (mouse, Merck, DUO92001-30RXN) and MINUS (rabbit, Merck, DUO92005-30RXN) secondary antibody probes. Amplification was performed using the Duolink® In Situ Detection Reagents Far red (Merck, DUO92013-30RXN).

- RNA FISH combined with IF is indeed a method that may be suitable and one could envisage using an rRNA probe. This would allow visualizing naRNA but to show LL37, again, antibody

staining would be necessary. As the method is not established and we dedicated time to PLA, we were unable to use this method within the period of time allocated to revisions.

In summary, we provide additional images and additional SYTO RNaselect staining, orthogonal projects and 3D reconstruction to corroborate our findings but have also tuned down our statements. But to explore the co-localization in greater detail – owed to high technical requirements to do so – would, in our opinion have to be considered future work.

Referee #3

Authors explore the role of NET associated RNA (naRNA) as a component of neutrophil extracellular traps (NETs) and its immunostimulatory properties in the skin.

Summary of paper:

1. naRNA in NETs: The paper establishes that naRNA is a common component of NETs formed by human polymorphonuclear neutrophils (PMNs) in response to various stimuli, including PMA, LL37-RNA complexes, nigericin, and live pathogens like *Candida albicans*. This is confirmed through confocal microscopy and metabolic labeling experiments.
2. naRNA as a DAMP: The study demonstrates that naRNA serves as a potent damage-associated molecular pattern (DAMP) that can trigger NET formation in primary PMNs. It is proposed that naRNA, in conjunction with LL37, acts as a 'composite DAMP' to activate neutrophils.
3. RNA Sensors and Inflammasome Activation: The research identifies the involvement of RNA sensors TLR8/Tlr13 and the NLRP3 inflammasome in naRNA-mediated NET formation.
4. Immune Cell Activation: The paper shows that naRNA can activate not only neutrophils but also other immune cells and these responses are mediated by RNA sensing.
5. naRNA in Keratinocytes: The study demonstrates that naRNA can activate keratinocytes, leading to the induction of psoriasis-associated genes. This indicates the potential role of naRNA in skin inflammation.
6. In Vivo DAMP Activity: The research extends to in vivo models, where intradermal injection of RNA-stabilized PMA NETs in mice induces skin inflammation. Tlr13-deficient mice show reduced skin inflammation in a psoriasis model.
7. Pre-Association of RNA and LL37: Pre-association of RNA and LL37 within neutrophil compartments occurs before NET extrusion, indicating that neutrophils contain preformed naRNA-LL37 DAMPs.

Comment #30: This is a hard-to-read manuscript mainly due to a poor language style.

Author reply #30: We thank the reviewer for this feedback. We were surprised because members of our team are English native speakers (e.g. N. Archer, Baltimore). We nevertheless contacted our Cluster of Excellence's scientific writer, Libera Lo Presti, a former editor for *Nature Microbiology*,

for additional editorial input and hope to have amended the style to a level satisfactory to the referee.

Comment #31: However, the main limitation of this manuscript is that authors are looking at only one side of biology and that is protected RNA, while they should also prove their concept with RNase-free NETs.

Author reply #31: We concede to the referee that in some of the in vivo work, we also used protected RNA in Fig. 4A-C. However, for some of the in vivo work (Fig. 4A) and practically all of the in vitro work, we used also non-protected RNA, showing significant differences. We think that the higher response is not due to the presence of the RNase inhibitor but rather a higher starting concentration of RNA in the PMA NETs. In our opinion, the main conclusions can be derived from data based on non-protected RNA and are not compromised by this experimental setup. Moreover, it would be outside the scope of this revision and problematic with regards to 3R guidelines to repeat all in vivo experiments in Fig. 4 with non-protected RNA. Considering the referee's comment we have amended the wording in this section and have added a general comment at the beginning of the discussion regarding the use of RNase inhibitors.

Comment #32: Major Points:

1. Is the pre-association of RNA and LL37 sufficiently shown? The data of Figure 5C show the colocalization to the granule compartment but do not show a bound complex. To convince, please show a control granule protein of which you know it is not associated with RNA. Or precipitate the RNA/LL37 complex from isolated granules.

Author reply #32: As discussed above in **Author reply #29** LL37 is a known granule protein (Sorensen, Arnljots et al. 1997) and further staining with SYTO RNaselect was provided (new Fig. EV5C-E). The technical limitations regarding the immunoprecipitation are discussed above..

Comment #33: 2. How is RNA bound to DNA? Via LL37? Figure 1 shows that RNA is associated with DNA strands.

Author reply #33: This is an intriguing question and one could speculate that LL37 functions as a connector (due to some of its positive charges connecting to the DNA backbone) or there are potentially partial areas of hybridization with the DNA backbone. Apart from the fact that it seems virtually impossible to analyze this aspect experimentally, we considered it not essential for the main conclusions drawn and hope that the referee agrees that this may be part of follow-up work in which the pre-association of naRNA and LL37 and the tethering to the NET is investigated in detail.

Comment #34: 3. The entire manuscript is built on the concept that effects of RNA are shown in the presence of RNase inhibitor. Authors need to complement their experiments by introducing a system in which the effects are abrogated in the absence of RNA. For example, in the animal experiments, authors should inject PMA NETs + RNase vs PMA NETs alone.

Author reply #34: We respectfully disagree with the reviewer that, quote, “the entire manuscript” is based on stabilized RNA as outlined in **Author reply #31**. Moreover, as intended by the referee, we use RNase digestion to show that in unprotected NETs, removal of the RNA abrogates the effects. This was already included in Figs. 1G, 4B-D and is further corroborated by genetic removal of RNA receptors and use of inhibitors. We considered the proposed additional in vivo experiment, but discarded the idea as: 1) Fig. 4A already shows a side-by-side comparison of PMA NETs with stabilized and non-stabilized PMA NETs; and 2) because finding the right conditions would require extensive titration of all reagents not only the stimulus, but also natural RNase (Probst, Brechtel et al. 2006) and endogenous RNase inhibitor (Abtin, Eckhart et al. 2009) levels in the skin as well as potential effects of RNases on TLR activation (Greulich, Wagner et al. 2019) have to be taken into account. The use of an RNase inhibitor as shown in Figs. 4A-C, primarily during the preparation of the NETs, avoided some of the complications arising from use of an active enzyme. In summary, the limitations of our approach have been highlighted to the reader to acknowledge the referee’s concerns at the beginning of the discussion: “Our conclusions largely focus on in vitro observations using primary human cells. As in vivo responses to NETs prepared without RNase inhibitor are relatively small, further work will be required to delineate the complex interplay between naRNA, endogenous RNases and endogenous RNase inhibitors more fully”.. We hope that this acknowledges the limitations of our study appropriately.

Comment #35: 4. Regarding your quantification of NETs DNA and RNA, you mention quantification of NETs DNA and RNA in immunofluorescence images but do not explain the units used, such as "rel. ROI number/ROI average size." There's a need for clarification regarding the calculation and presentation of ROIs, and an example of ROIs with and without NETs should be provided.

Author reply #35: We appreciate the referee’s concern and have amended the methods description accordingly. As the selection of ROIs is indeed challenging due to the erratic shape of the NETs in vitro we now harmonized quantification of in vitro results throughout the manuscript as “tiles showing NETs”, i.e. showing diffuse DNA signal. An example is shown below. In brief, an 8 by 8 grid is superimposed on each image (all images taken with the same settings, including objective, zoom and resolution) and then scored manually. A tile was counted as NET positive whenever a DNA (Hoechst) signal was clearly detected in shape of a fiber outside of a cell with a destroyed nucleus. A tile was counted as negative when only intact cells were observed within this respective area. An example is provided below:

Example of 8x8 tile counting for NETs. Left: original confocal image with DNA visualized by Hoechst staining. Right: scored tiles used for quantification labelled in white.

Comment #36: 5. Figure 1G is supposed to complement Figure 1E, but different methods are used to calculate and show the presence of NETs. The paper should present one or both analysis methods for both experiments. Please, standardize your dispersion versus tile counting methodology.

Author reply #36: As outlined in **Author reply #35**, we have standardized the methodology to exclusively use tile counting.

Comment #37: 6. To support the claim that naRNA, and not DNA, is the relevant immunostimulatory component for NETs, the paper should include experiments where PMA NETs are treated with DNase and DNase inhibitors.

Author reply #37: This control experiment also occurred to us and DNase treatment was indeed included in Fig. EV3C, see Line 244. Moreover, since genetics are more reliable than DNA digestion, we used *Tlr9* KO BM-PMN cells to exclude an involvement of DNA. On the other hand, KOs for RNA receptors were used, collectively providing a very clear picture that RNA, and not DNA, is responsible for the observed effects.

Comment #38: 7. While the in vivo experiments are interesting, they mainly focus on the presence and absence of extracellular RNA rather than NETs-associated extracellular RNA.

Author reply #38: As there is no unequivocal marker to distinguish between these two types of RNA but we can clearly show that NETs contain immunostimulatory RNA, we consider it acceptable to argue that the presence of NETs in vivo entails naRNA-mediated effects, especially if confirmed by the effects of an RNA receptor (*Tlr13*) KO. We conducted additional staining of skin samples from WT and *Tlr13* KO IMQ-treated animals for the PMN marker MPO and the NET marker CitH3. This shows a clear reduction of both signals in *Tlr13* KO. An example as well as an unbiased quantification are provided in the new Fig. 4G, H. We trust this addresses this point as it both shows the presence of the neutrophils and NET in the tissue as a consequence of IMQ treatment as well as their reduction in *Tlr13* KO.

Comment #39: The paper should address the potential effects of RNase inhibitors on inflammation in an in vivo setting. Although you show 'RNase inhibitor+RPMI' you do not show the RNase inhibitor in the context of inflammation that is not driven by NETs.

Author reply #39: Please see **Author reply #31**. Due to the abovementioned complexity, the exact role of endogenous RNases and endogenous RNase inhibitors will require a dedicated research effort outside the scope of the present work. We have added this as a limitation to the present work as described in the abovementioned reply.

Comment #40: 8. In Figure 3 pre-digestion of NETs with RNase A abolished the stimulatory effects of NETs on keratinocytes, whereas pre-digestion with DNase I did not. It is hard to believe that the stimulatory effect of NETs on keratinocytes is abrogated by RNase but not by DNase. NETs are DNA

scaffolds. If you break them down, the associated RNA should be affected. Please, show in the supernatants the size of DNA fragments (note comments 2 and 12).

Author reply #40: As discussed above in **Author reply #14**, quantification is not technically straightforward. Moreover, we would like to point out that the DNase digestion was done on already harvested NETs, so naRNA contained in the NETs is not removed, even if it is released from the DNA strands. Therefore, it would be present when added to the keratinocytes. Moreover, we rule out the involvement of DNA by alternative methods, e.g. KD of MyD88 and STING, which would be expected to affect TLR9 or cGAS sensing pathways. These results were consistent with no effect of DNase treatment.

Comment #41: 9. The paper should address whether RNase treatment affects the degradation of keratinocyte RNA in addition to NET RNA. A key point you make is to show the different stimulatory potential between RNase and DNase treated NETs. In Figure 3b you show that RNase but not DNase abolishes the stimulatory effects of NETs on keratinocytes, however your readout is using qPCR to quantify the RNA of the psoriasis-associated genes. How do you rule out whether the RNase also degrades the keratinocyte, psoriasis associated RNA? Please, show the protein production in another experiment.

Author reply #41: We are not able to provide protein data as there are no remaining samples and the experiment cannot be repeated easily. But we provide below a figure showing plotted Ct values for the housekeeping gene *TBP*. Evidently, there is no visual or statistically significant difference in CT values, and hence mRNA levels, of the housekeeper for samples in which the PMA NETs had been RNase or DNase pre-treated . This is also not surprising since the RNase was added after NET harvest and this preparation was diluted 1:50 into to tissue culture media along with the contained NETs to stimulate the keratinocytes. This high dilution of RNase in the media is unlikely to be sufficient to significantly digest intracellular RNA in live keratinocytes after harvesting of the cells for RNA isolation. The stimulant and any RNase contained in the media were also aspirated and the cells washed before they were harvested and lysed for RNA isolation. Thus, the likelihood of RNase in the stimulus to carry over into lysates and hence RNA preps, to then have an effect on the prepared RNA is very low. In any case, the housekeeper analysis unequivocally rules this out.

Graph showing housekeeper *TBP* mRNA CT values from qPCR experiments in Fig. 3B-C, grouped by 'all values' except those from wells in which PMA NETs had been RNase or DNase pre-treated, those from wells with 'RNase pre-treated' PMA NETs and those from wells with 'DNase pre-treated' PMA NETs. Differences were non-significant according to Kruskal-Wallis non-parametric ANOVA test with Dunn's correction for multiple testing.

Comment #42: 10. When discussing the differences in ear thickness and epidermal thickness in wild-type and *Tlr13* KO animals, it would be important to include images of the ears from each group for visual reference.

Author reply #42: Cross sections were included for Fig. 4F as common place for quantifying inflammation in the imiquimod model. We further added IF staining to highlight neutrophil infiltration and NET formation (new Fig. 4G and 4H) following the suggestion, see **Author reply #20**. Unfortunately, photographic images of the entire ear were not taken at the time as this is not typically done, and we are not permitted by law to use further animals for the sole purpose of acquiring such new images. We hope that by adding the IF analysis there is sufficient information to assess skin inflammation.

Comment #43: 11. Are RNA-LL37 complexes released during degranulation, outside of NETosis?

Author reply #43: This is an intriguing possibility that we could address in future work not directly related to the present manuscript focusing on NETs. So far, we have no evidence that release of LL37-naRNA would precede or would be decoupled from NET formation in any way.

Comment #44: 12. With regards to fresh and aged/degraded NETs I do not see any characterization of your fresh and aged NETs with serum or medium. What happens to the DNA, RNA and proteins after incubation? Serum contains RNases, DNases, and proteases which all may play a role here.

Author reply #44: Indeed, our assumption is that serum contains all of these and that these may be physiological ways in which the inflammatory potential of NETs is reduced over time. We provide new immunofluorescence staining of fresh vs aged NETs in the **new Fig. EV5F**. Moreover, we have

now not only tested fresh vs aged NETs on keratinocytes (Fig. 5G) but also on primary PMN (**new Fig. 5F**), showing the same effect. Although there may be multiple ways in which NETs lose their reactivity, these clearly relate to RNA, consistent with our findings that NETs activate primary PMN and keratinocytes via RNA receptors and RNA dependently.

New Fig. EV5F: Confocal microscopy of PMA-induced NETs from primary human PMNs incubated for 30 min or 4 h with human serum and stained for DNA (Hoechst 33342, white) and naRNA (anti-rRNA Y10b, magenta or red). Lower magnification (left, scale bar = 50 μ m) and higher magnification (right, scale bar = 20 μ m) for one representative of n=2 biological replicates shown.

New Fig. 5F: Quantification of NET release by primary human PMNs using confocal microscopy. Human PMNs were stimulated with fresh or old NETs generated by incubation in PMN culture medium or human serum treatment, respectively. Afterwards PMNs were stained for DNA (Hoechst 33342) and quantified as illustrated above (n= 2 donors, mean+SD, each dot represents the number of NET-positive tiles in one image quantified from three images/condition, *p<0.05 according to Kruskal-Wallis test with Dunn's correction)

Comment #45: 13. Simplify your language, e.g. avoid purposefully and accidentally and barbed wire roadblocks, etc. Along the same lines, the manuscript title needs to be rephrased.

Author reply #45: We thank the reviewer for this comment but have also had positive responses to explain biological phenomena in this way. Referee 1 and 2 apparently did not have negative feedback about style. Regarding this and the title we have nevertheless sought feedback from our Cluster's scientific writer, herself previously an editor at *Nature Microbiology*, and by the *EMBO Reports* editor. The revised manuscript text and title are the result of this consultation and hopefully agreeable to referee 3.

Comment #46: 14. The paragraph discussing the trade-off between DAMP activity and RNA's short-lived nature should be revised or removed from the results section, as it is more of a discussion point.

Author reply #46: We thank referee 3 for flagging this up and have amended the text accordingly.

Comment #47: 15. The graphical abstract is not clear.

Author reply #47: We appreciate the feedback and will confer with the editor about the final layout.

Comment #48: 16. Reference to immune tolerance is not subject of this work.

Author reply #48: As outlined in **Author reply #1**, LL37 interacting with RNA and DNA has been viewed to act as a breaker of tolerance to self-nucleic acids in disease (Lande, Chamilos et al. 2015). We show that it interacts with RNA to act as a DAMP in healthy human donor neutrophils. Hence, the concept of immune tolerance to self-nucleic acids requires revision. This is what we sought to express in the title. Again, we will seek the editor's advice on the final phrasing of the title.

References cited in point-by-point reply

Abtin, A., L. Eckhart, M. Mildner, M. Ghannadan, J. Harder, J. M. Schroder and E. Tschachler (2009). "Degradation by stratum corneum proteases prevents endogenous RNase inhibitor from blocking antimicrobial activities of RNase 5 and RNase 7." *J Invest Dermatol* **129**(9): 2193-2201.

Arpinati, L., M. E. Shaul, N. Kaisar-Iluz, S. Mali, S. Mahroum and Z. G. Fridlender (2020). "NETosis in cancer: a critical analysis of the impact of cancer on neutrophil extracellular trap (NET) release in lung cancer patients vs. mice." *Cancer Immunol Immunother* **69**(2): 199-213.

Douda, D. N., L. Yip, M. A. Khan, H. Grasemann and N. Palaniyar (2014). "Akt is essential to induce NADPH-dependent NETosis and to switch the neutrophil death to apoptosis." *Blood* **123**(4): 597-600.

Dulhunty, A. F., P. R. Junankar and C. Stanhope (1993). "Immunogold labeling of calcium ATPase in sarcoplasmic reticulum of skeletal muscle: use of 1-nm, 5-nm, and 10-nm gold." *J Histochem Cytochem* **41**(10): 1459-1466.

Eigenbrod, T. and A. H. Dalpke (2015). "Bacterial RNA: An Underestimated Stimulus for Innate Immune Responses." *J Immunol* **195**(2): 411-418.

Flechsler, J., T. Heimerl, C. Pickl, R. Rachel, Y. D. Stierhof and A. Klingl (2020). "2D and 3D immunogold localization on (epoxy) ultrathin sections with and without osmium tetroxide." *Microsc Res Tech* **83**(6): 691-705.

Gray, R. D., G. Hardisty, K. H. Regan, M. Smith, C. T. Robb, R. Duffin, A. Mackellar, J. M. Felton, L. Paemka, B. N. McCullagh, C. D. Lucas, D. A. Dorward, E. F. McKone, G. Cooke, S. C. Donnelly, P. K. Singh, D. A. Stoltz, C. Haslett, P. B. McCray, M. K. B. Whyte, A. G. Rossi and D. J. Davidson (2018). "Delayed neutrophil apoptosis enhances NET formation in cystic fibrosis." *Thorax* **73**(2): 134-144.

Greulich, W., M. Wagner, M. M. Gaidt, C. Stafford, Y. Cheng, A. Linder, T. Carell and V. Hornung (2019). "TLR8 Is a Sensor of RNase T2 Degradation Products." *Cell* **179**(6): 1264-1275 e1213.

Herster, F., Z. Bittner, N. K. Archer, S. Dickhofer, D. Eisel, T. Eigenbrod, T. Knorpp, N. Schneiderhan-Marra, M. W. Loffler, H. Kalbacher, T. Vierbuchen, H. Heine, L. S. Miller, D. Hartl, L. Freund, K. Schakel, M. Heister, K. Ghoreschi and A. N. R. Weber (2020). "Neutrophil extracellular trap-associated RNA and LL37 enable self-amplifying inflammation in psoriasis." *Nat Commun* **11**(1): 105.

Kenny, E. F., A. Herzig, R. Kruger, A. Muth, S. Mondal, P. R. Thompson, V. Brinkmann, H. V. Bernuth and A. Zychlinsky (2017). "Diverse stimuli engage different neutrophil extracellular trap pathways." *Elife* **6**.

Klasener, K., P. C. Maity, E. Hobeika, J. Yang and M. Reth (2014). "B cell activation involves nanoscale receptor reorganizations and inside-out signaling by Syk." *Elife* **3**: e02069.

Lande, R., G. Chamilos, D. Ganguly, O. Demaria, L. Frasca, S. Durr, C. Conrad, J. Schroder and M. Gilliet (2015). "Cationic antimicrobial peptides in psoriatic skin cooperate to break innate tolerance to self-DNA." *Eur J Immunol* **45**(1): 203-213.

Li, Q., Y. Kim, J. Namm, A. Kulkarni, G. R. Rosania, Y. H. Ahn and Y. T. Chang (2006). "RNA-selective, live cell imaging probes for studying nuclear structure and function." *Chem Biol* **13**(6): 615-623.

Lim, M. B., J. W. Kuiper, A. Katchky, H. Goldberg and M. Glogauer (2011). "Rac2 is required for the formation of neutrophil extracellular traps." *J Leukoc Biol* **90**(4): 771-776.

Masuda, S., S. Shimizu, J. Matsuo, Y. Nishibata, Y. Kusunoki, F. Hattanda, H. Shida, D. Nakazawa, U. Tomaru, T. Atsumi and A. Ishizu (2017). "Measurement of NET formation in vitro and in vivo by flow cytometry." *Cytometry A* **91**(8): 822-829.

Neubert, E., S. N. Senger-Sander, V. S. Manzke, J. Busse, E. Polo, S. E. F. Scheidmann, M. P. Schon, S. Kruss and L. Erpenbeck (2019). "Serum and Serum Albumin Inhibit in vitro Formation of Neutrophil Extracellular Traps (NETs)." *Front Immunol* **10**: 12.

Parker, H., M. Dragunow, M. B. Hampton, A. J. Kettle and C. C. Winterbourn (2012). "Requirements for NADPH oxidase and myeloperoxidase in neutrophil extracellular trap formation differ depending on the stimulus." *J Leukoc Biol* **92**(4): 841-849.

Probst, J., S. Brechtel, B. Scheel, I. Hoerr, G. Jung, H. G. Rammensee and S. Pascolo (2006). "Characterization of the ribonuclease activity on the skin surface." *Genet Vaccines Ther* **4**: 4.

Remijsen, Q., T. Vanden Berghe, E. Wirawan, B. Asselbergh, E. Parthoens, R. De Rycke, S. Noppen, M. Delforge, J. Willems and P. Vandenabeele (2011). "Neutrophil extracellular trap cell death requires both autophagy and superoxide generation." *Cell Res* **21**(2): 290-304.

Silva, C. M. S., C. W. S. Wanderley, F. P. Veras, A. V. Goncalves, M. H. F. Lima, J. E. Toller-Kawahisa, G. F. Gomes, D. C. Nascimento, V. V. S. Monteiro, I. M. Paiva, C. Almeida, D. B. Caetite, J. C. Silva, M. I. F. Lopes, L. P. Bonjorno, M. C. Giannini, N. B. Amaral, M. N. Benatti, R. C. Santana, L. E. A. Damasceno, B. M. S. Silva, A. H. Schneider, I. M. S. Castro, J. C. S. Silva, A. P. Vasconcelos, T. T. Goncalves, S. S. Batah, T. S. Rodrigues, V. F. Costa, M. C. Pontelli, R. B. Martins, T. V. Martins, D. L. A. Esposito, G. C. M. Cebinelli, B. A. L. da Fonseca, L. O. S. Leiria, L. D. Cunha, E. Arruda, H. I. Nakaia, A. T. Fabro, R. D. R. Oliveira, D. S. Zamboni, P. Louzada-Junior, T. M. Cunha, J. C. F. Alves-Filho and F. Q. Cunha (2022). "Gasdermin-D activation by SARS-CoV-2 triggers NET and mediate COVID-19 immunopathology." *Crit Care* **26**(1): 206.

- Sollberger, G., A. Choidas, G. L. Burn, P. Habenberger, R. Di Lucrezia, S. Kordes, S. Menninger, J. Eickhoff, P. Nussbaumer, B. Klebl, R. Kruger, A. Herzig and A. Zychlinsky (2018). "Gasdermin D plays a vital role in the generation of neutrophil extracellular traps." Sci Immunol **3**(26).
- Sorensen, O., K. Arnljots, J. B. Cowland, D. F. Bainton and N. Borregaard (1997). "The human antibacterial cathelicidin, hCAP-18, is synthesized in myelocytes and metamyelocytes and localized to specific granules in neutrophils." Blood **90**(7): 2796-2803.
- Stojkov, D., M. J. Claus, E. Kozlowski, K. Oberson, O. P. Scharen, C. Benarafa, S. Yousefi and H. U. Simon (2023). "NET formation is independent of gasdermin D and pyroptotic cell death." Sci Signal **16**(769): eabm0517.
- Suzuki, K., M. Tsuchiya, S. Yoshida, K. Ogawa, W. Chen, M. Kanzaki, T. Takahashi, R. Fujita, Y. Li, Y. Yabe, T. Aizawa and Y. Hagiwara (2022). "Tissue accumulation of neutrophil extracellular traps mediates muscle hyperalgesia in a mouse model." Sci Rep **12**(1): 4136.
- Udby, L. and N. Borregaard (2007). "Subcellular fractionation of human neutrophils and analysis of subcellular markers." Methods Mol Biol **412**: 35-56.
- Wu, Y., Y. Liu, C. Lu, S. Lei, J. Li and G. Du (2020). "Quantitation of RNA by a fluorometric method using the SYTO RNASelect stain." Anal Biochem **606**: 113857.
- Zhou, B., W. Liu, H. Zhang, J. Wu, S. Liu, H. Xu and P. Wang (2015). "Imaging of nucleolar RNA in living cells using a highly photostable deep-red fluorescent probe." Biosens Bioelectron **68**: 189-196.
- Zhu, Y., K. Gong, M. Denholtz, V. Chandra, M. P. Kamps, F. Alber and C. Murre (2017). "Comprehensive characterization of neutrophil genome topology." Genes Dev **31**(2): 141-153.

Dear Prof. Weber,

Thank you for the submission of your revised research manuscript to EMBO reports. I have now received the comments from the three referees that were asked to re-assess the study, which you will find below.

I am sorry to say that the evaluation of your revised manuscript is not a positive one. As you will see, all three referees still have remaining concerns, indicating that previous points have not been adequately addressed and that part of the data in the revised manuscript still do not adequately support the conclusions drawn. Moreover, referees #1 and #2 still indicate novelty concerns.

Given these remaining concerns, the fact that you already had a chance to revise the study, and that we allow a single round of major revision only, I am afraid that I cannot offer to publish the manuscript at this point. I am sorry that this decision emerges as the outcome of a lengthy review process, but given that all three referees are still not convinced by the current set of data, I have no other option but to reject your manuscript.

I nevertheless hope, that the referee comments will be helpful in your continued work in this area, and I thank you once more for your interest in our journal.

Yours sincerely

Referee #1:

The authors' response to this Reviewer's comments is deemed inadequate, with critical concerns remaining unaddressed and the supporting evidence for their claims require further substantiation. In response to the lack of novelty, the authors argue, ignoring the fact that the topic has been previously extensively published by others, such as Lande et al. 2007, Ganguly et al., J. Exp Med 2009, Smolarz et al., Front Cell Infect Microbiol, 2021. In addition, the authors did not address all points previously raised, and many of the provided answers are not convincing as they are not supported by actual data:

1. Insufficient evidence for GSDMD-dependent NET formation:

The authors' claim that naRNA induces NET formation through a unique TLR8-NLRP3-caspase-1-gasdermin D-dependent inflammasome pathway lacks clear evidence [Authors' reply #16]. Activation of GSDMD should have been demonstrated in human and mouse neutrophils, ideally using GSDMD^{-/-} mouse neutrophils, to establish that NETs induced by naRNA is GSDMD-dependent. Moreover, to demonstrate the involvement of GSDMD in this process, the authors could have provided immunoblots showing the cleaved form of GSDMD (N-GSDMD) as active form upon activation with naRNA-induced NETs. As previously mentioned (Reviewer comments #9), this would be a solid proof of the proposed signaling pathway to elucidate the role of GSDMD.

2. Inadequate consideration of neutrophil activation conditions:

The authors followed the suggested advice of the Reviewer to reduce the concentration of the PMA; however, a 4-hour stimulation is not appropriate for NET formation. NETs can occur within minutes after PMA treatment (Brinkmann et al., Science 2004). Furthermore, the authors should have provided data on neutrophil viability under the same activation conditions.

3. Failure to address concerns about potential dsDNA presence that was quantified as RNA associated NETs:

The authors neglected comments regarding the confirmation of dsDNA vs. RNA in activated samples [Comment #14]. To validate the DNA/RNA association with NETs, the authors should have collected supernatants from stimulated human neutrophils and measured nucleic acid concentrations in the presence and absence of DNase enzyme.

Referee #2:

The authors have clarified some of the concerns that I had but technical issues still remain:

Comment 23:

The response to this comment is not satisfactory. There seems to be a clear difference in the number of NETs induced with A23-treated NETs. The quantification method is inappropriate: the authors state: (n = 1 donor, mean+SD, each dot represents the number of NET-positive tiles in one image quantified from three images/condition).

Quantification of tiles with NETs is misleading, a tile would be counted positive even if there is one NET per tile, indeed the A23 image looks very different from the PMA-NET image. There is an obvious difference, further strengthening my claim that this is a misleading quantification.

Furthermore, Hoescht does not discriminate between intracellular and extracellular DNA, so it cannot be used to quantify NETs. It appears that there is a biased calling out of what is a NET - the field has advanced beyond that and appropriate quantification techniques have been developed.

N=1 is also not convincing, as neutrophils can exhibit lots of heterogeneity in responses.

Comment 24: same issues with the assay as above, however here the image at least looks like it supports the authors' claims. It should be repeated with an appropriate NET assay, and multiple biological replicates.

Comment 25: neutrophils are sensitive and apoptosis is easily triggered by chemicals. Data using chemical inhibitors is hard to interpret without providing evidence that the inhibitors are not inducing apoptosis. My concern for this data has not been alleviated in the absence of this important control. Authors should provide AnnexinV binding data for apoptosis.

Comment 26: apologies for the lack of clarity. My question was whether naRNA is de novo transcribed. PMA is a strong inducer of transcription and it is interesting to know if naRNA is made in the PMA transcriptional burst. Although an important question, I agree with the authors that these data are not necessary for publication.

Comment 29: the inclusion of the SytoRNA select reagent is helpful and the fact that microscopy experiment is split into different channels is also commendable. However my suggestion was to provide the same split channels for the anti-rRNA Y10b, so that the evidence can be considered collectively. Can the authors rule out nascent LL37 and ribosomal RNA here? To prove that RNA is found in granules, the authors should provide evidence of colocalization of other granule components. Cathelicidin/LL37 is considered to be localised to secondary granules, so lactoferrin or NGAL would be appropriate markers.

In response to the comment by reviewer number 1, I did additional novelty search and found the following publication by the same group:

<https://pubmed.ncbi.nlm.nih.gov/31913271/>

Some of the data from this previous report appears to be repackaged here, which reduces my enthusiasm and makes me more likely to agree with reviewer 1 that this report lacks novelty

Referee #3:

1. Overall, authors provide verbal but few satisfactory experimental responses to the reviewers' comments (looking at reviewers 1 and 2)
2. Regarding our comments, authors clarified well the quantification of NETs by counting tiles and harmonizing all other techniques for NET quantification.
3. Authors respond well to differential RNase digestion of naRNA versus keratinocyte RNA, but authors should also look at protein concentrations.
4. However, both the discussion on immune tolerance, the paper title and central figure remain unclear, and may be not be clarifiable with the editor's help.
5. Please explain why the concept of immune tolerance to self-nucleic acids requires revision.

In summary, one of the challenges for a scientist is to relate concepts in a simple and clear manner, and this is not achieved in this work.

** As a service to authors, EMBO Press provides authors with the ability to transfer a manuscript that one journal cannot offer to publish to another journal, without the author having to upload the manuscript data again. To transfer your manuscript to another EMBO Press journal using this service, please click on
Link Not Available

**A) Detailed appeal EMBOR-2023-58048-T**

We thank the editor and all referees for the time and consideration they have given to our work. Hereby, we would like to appeal against this decision to reject our work after revision based on three main reasons (this **section A**). We ground this based on the most recent comments after revision (**section B**) and the point-by-point reply to the original referee comments (**section C**):

1) The suggested lack of novelty is clearly contradicted by the metrics of our paper

Since it was initially placed on *Biorxiv* in July 2022, our work has experienced considerable interest with >1,300 full text PDF downloads and first citations – despite the fact that further novel observations have been added since then, which are not even included in the *Biorxiv* version:

- Identification of the NLRP3 inflammasome as a critical mediator of a novel, non-canonical TLR8-to-NET pathway in primary neutrophils
- Induction of a psoriasis gene signature by RNA sensing in primary keratinocytes
- Identification of NOD2 as the receptor for naRNA in keratinocytes. That NOD2 can act in this manner is unexpected and will hope up entirely new questions.
- Experimental evidence for the self-restricting nature of the naRNA-LL37 DAMP due to RNA degradation; so far, no self-restricting DAMPs have been described

These points significantly delineate this present manuscript from our previous work; additionally

- Previous work did not show that naRNA is a canonical component of all NETs,
- Previous work hinted to an activity of NETs that might be RNA-related but only here do we show – in vitro and in vivo – that naRNA is the single active ingredient in the way that NETs propagate inflammation
- Only this new work shows a clear mechanistic pathway, e.g. the involvement of NLRP3, in neutrophils and describes a new RNA-related sensing pathway in keratinocytes via NOD2.

- The concepts of a pre-formed composite DAMP and its transient nature are entirely new and has not been described elsewhere

It is therefore the current submission that shows that naRNA is not only relevant for certain NETs but is a general neutrophil property activating concrete effector pathways which will completely change thinking about NETs and thus get cited.

As an indication of this the work has been cited twice in its pre-print form by:

- Bian F, Yan D, Wu X, Yang, C. A Biological Perspective of TLR8 Signaling in Host Defense and Inflammation. *Infectious Microbes and Disease* 2023, 5(2), 44-55.
- Yang C and Yuan R. TLRs and other molecules signaling crosstalk in diseases. Book chapter in “Thirty years of Toll-like receptor’s discovery” (Kumar V, editor); IntechOpen 2024

Finally, albeit ‘non-metric’, the response to talks given about this work in Cambridge (UK), Worcester (USA) and Boston (USA) and in-depth discussions with experts in innate immunity (Jonathan Kagan, Michel Gilliet, Kate Fitzgerald) and neutrophils (Denisa Wagner) have confirmed to us the novelty and technical rigor of our work. Most probably, these leaders in the field would confirm this, if asked.

Based on this, we suggest that metric and verifiable indicators of novelty and interest in the community are at critical odds with the conclusions of the reviewers. We contend that our published work should be shared with the community upon publication and hence appeal against “lack of novelty” as a criterion for rejection.

2) We have made efforts to adequately alleviate the initially raised technical concerns , but were answered with concerns that should have been raised first time or lacking concrete reference to specific literature

We appreciate the reviewers’ attention to technical details and their initial comments as we ourselves like to set a high technical standard for own work. Multiple cellular models, using both inhibitors and genetic approaches, are evidence of this. One referee concern was the stimulation conditions for inducing NETs with PMA¹ to generate naRNA containing stimulants. Specifically, we discussed this in author replies #7, #23 and #24 (section C) and performed additional experiments depicted in the rebuttal, addressing all three specific requests, (i) namely use of a lower PMA concentration (comment #7), (ii) incubation of mock NETs with PMA post-NET-formation and washing (comment #24), and (iii) using NETs induced by non-PMA stimuli (revision comment #23). We are surprised that the reviewers were not satisfied with these control experiments, even

¹ As an aside, we would like to add that PMA was only used as the most commonly used NET trigger to generate NETs containing naRNA; PMA-triggered NET release, despite having been studied in hundreds of papers, was not studied here as PMA is entirely non-physiological as it plays no role in mammalian physiology; rather we studied the biology of one of the few physiologically relevant NET triggers, namely RNA+LL37. Although the referee’s concern regarding stimulation conditions is legitimate, one also needs to bear in mind that the debate is merely about PMA as a “tool” to produce NETs.

though all concerns were alleviated and even though in the literature there is evidence for using >25 mM PMA and/or stimulation for >30 min, e.g. (Gray, Lucas et al. 2013, Flores, Dohrmann et al. 2016, Masuda, Shimizu et al. 2017, de Bont, Koopman et al. 2018, Shrestha, Ito et al. 2019, Yang, Luo et al. 2019) to list a few, highlighting there is a range of valid stimulation conditions that lead to PMA-mediated NET induction. That NETs induced by different accepted stimuli are different in shape can be seen in Fig. 1A and e.g. (Gray, Lucas et al. 2013, Sosa-Luis, Rios-Rios et al. 2021). Differences in shape is therefore not an objective argument against the validity of our findings as referee 2 argues (re-revision comment #5). Since these were reviewer experiments (that were not shown in the paper), we did not perform the experiments three times, but clearly our methods and results fall within what is seen in the field if one would take an unbiased and comprehensive view. Another more important issue with the referee's comments is that, despite requests on our side, they failed to back up their judgements with appropriate literature. We provide several examples:

- Only referee 3 had a question regarding the NET quantification (comment #35) and asked why one panel showed a different quantification method (comment #36), which we responded to by outlining our procedure and providing examples (revision author reply #35 and #36). Reviewer 3 states "authors clarified well the quantification of NETs by counting tiles and harmonizing all other techniques for NET quantification (Re-review comment #12 in Section B). But now referee 2 (Re-review comment #5 and #6 in Section B) criticizes that alternative "appropriate quantification techniques have been developed" and an "appropriate NET assay" should have been used. Ironically, the referee 2 concedes that the "image at least looks like it supports the authors' claims". However, which published quantification method or NET assay the referee is referring to is not specified and thus no objective reference point is provided.
- Referee 2 was concerned about neutrophil apoptosis (original comment #25 in section C). We argued the point providing 4 references and close-up images why apoptosis can be excluded (revision author reply #25). The referee maintains his point after revision (Re-review comment #7 in Section B) but, as before, fails to provide concrete evidence that would substantiate that his/her remaining concern relates to a specific paper.
- Referee 1 repeatedly quoted literature that, upon close inspection, does not nullify our claims (see revision author replies #2, #3, #16 and #19 in section C). The referee does not seem to have studied the details of these papers before using them as a basis for criticism.

In summary, providing objective literature citations would have helped us to better revise our work and understand why our revision is supposedly inadequate. However, we have the impression that regarding technical issues, goal posts were somewhat shifted. Moreover, requests and criticisms were made without clear reference to previous published work as an unbiased and verifiable basis for criticism. Especially for criticisms that lead to a rejection after adequate efforts have been made, we – and surely a journal like EMBO reports – would expect those to be "grounded" on published

work. Whilst we are, of course, willing to reflect some of these concerns even more explicitly in our discussion (e.g. the limitations paragraph) of a final manuscript version, we find this a justified basis for appealing against the decision to reject.

3) Repeatedly minor comments have unnecessarily been inflated to major comments

This applies especially to reviewer 1 and even in the first revision. In the first set of reviews, the following points were proposed as major criticisms:

- The ratio of reviews to original papers cited (which was adequate as outlined in revision author reply #4 in section C).
- One missing call-out of one figure panel (which was in fact not even missing as detailed in revision author reply #6).
- Scale bars missing for approx. 2-3 panels out of a total of >20 (revision author reply #11).
- Clarification of the relative quantification of PCR (revision author reply #18).
- Not addressing the role of GSDMD in NET formation (Re-review comment #2 in section B and comment #16 in section C). As this is clearly a controversial issue (as we detailed in author reply #16 of section C) we consider it a minor point; the referee could have easily recommended to tone down a reference of GSDMD instead of making it a major point for rejection in re-review comment #2.

In our opinion inflation of minor issues to major criticisms makes it difficult to dismiss an overly critical bias in referee 1, which subsequently also affected the judgment of the other two referees. As peer review should be open-minded, we propose bias or an overcritical stance as another reason for an appeal.

We would like to clearly state that we respect the referees as colleagues and have sought to take all justified criticism to heart as a result. Moreover, we appreciate their time spent on evaluating our work, even if we disagree with their recommendation to reject our work. We hope that, given the concerns outlined above, another arbiter can evaluate both our work and how it was reviewed.

B) Re-review comments of reviewers for EMBOR-2023-58048-T (31.01.24)

We here do not discuss these individual comments in detail as they have largely been discussed above or were already commented on in the initial revision (please refer to section C). Nevertheless, we are willing to amend wording that may have been slightly overstated (e.g. by removing the section on GSDMD) or unclear.

Referee #1

Re-review comment #1: The authors' response to this Reviewer's comments is deemed inadequate, with critical concerns remaining unaddressed and the supporting evidence for their

claims require further substantiation. In response to the lack of novelty, the authors argue, ignoring the fact that the topic has been previously extensively published by others, such as Lande et al. 2007, Ganguly et al., J. Exp Med 2009, Smolarz et al., Front Cell Infect Microbiol, 2021. In addition, the authors did not address all points previously raised, and many of the provided answers are not convincing as they are not supported by actual data:

Author reply: please see author replies #2, #3, #16 and #19 in section C below.

Re-review comment #2: 1. Insufficient evidence for GSDMD-dependent NET formation: The authors' claim that naRNA induces NET formation through a unique TLR8-NLRP3-caspase-1-gasdermin D-dependent inflammasome pathway lacks clear evidence [Authors' reply #16]. Activation of GSDMD should have been demonstrated in human and mouse neutrophils, ideally using GSDMD^{-/-} mouse neutrophils, to establish that NETs induced by naRNA is GSDMD-dependent. Moreover, to demonstrate the involvement of GSDMD in this process, the authors could have provided immunoblots showing the cleaved form of GSDMD (N-GSDMD) as active form upon activation with naRNA-induced NETs. As previously mentioned (Reviewer comments #9), this would be a solid proof of the proposed signaling pathway to elucidate the role of GSDMD.

Author reply: please see author reply 16 in section C below.

Re-review comment #3: 2. Inadequate consideration of neutrophil activation conditions: The authors followed the suggested advice of the Reviewer to reduce the concentration of the PMA; however, a 4-hour stimulation is not appropriate for NET formation. NETs can occur within minutes after PMA treatment (Brinkmann et al., Science 2004). Furthermore, the authors should have provided data on neutrophil viability under the same activation conditions.

Author reply: Even though NETs can occur as early as 15-30 min, our goal was to generate a culture in which 100% of neutrophils had formed NETs. Longer time points have also been used by (Gray, Lucas et al. 2013, Flores, Dohrmann et al. 2016, Masuda, Shimizu et al. 2017, de Bont, Koopman et al. 2018, Shrestha, Ito et al. 2019, Yang, Luo et al. 2019).

Re-review comment #4: 3. Failure to address concerns about potential dsDNA presence that was quantified as RNA associated NETs: The authors neglected comments regarding the confirmation of dsDNA vs. RNA in activated samples [Comment #14]. To validate the DNA/RNA association with NETs, the authors should have collected supernatants from stimulated human neutrophils and measured nucleic acid concentrations in the presence and absence of DNase enzyme.

Author reply: see author reply #14 in section C below.

Referee #2

The authors have clarified some of the concerns that I had but technical issues still remain:

Re-review comment #5: Comment 23: The response to this comment is not satisfactory. There seems to be a clear difference in the number of NETs induced with A23-treated NETs. The quantification method is inappropriate: the authors state: (n = 1 donor, mean+SD, each dot

represents the number of NET-positive tiles in one image quantified from three images/condition). Quantification of tiles with NETs is misleading, a tile would be counted positive even if there is one NET per tile, indeed the A23 image looks very different from the PMA-NET image. There is an obvious difference, further strengthening my claim that this is a misleading quantification. Furthermore, Hoescht does not discriminate between intracellular and extracellular DNA, so it cannot be used to quantify NETs. It appears that there is a biased calling out of what is a NET - the field has advanced beyond that and appropriate quantification techniques have been developed. N=1 is also not convincing, as neutrophils can exhibit lots of heterogeneity in responses.

Author reply: discussed in detail above in section A, part 2).

Re-review comment #6: Comment 24: same issues with the assay as above, however here the image at least looks like it supports the authors' claims. It should be repeated with an appropriate NET assay, and multiple biological replicates.

Author reply: discussed in detail above in section A, part 2).

Re-review comment #7: Comment 25: neutrophils are sensitive and apoptosis is easily triggered by chemicals. Data using chemical inhibitors is hard to interpret without providing evidence that the inhibitors are not inducing apoptosis. My concern for this data has not been alleviated in the absence of this important control. Authors should provide Annexin V binding data for apoptosis.

Author reply: discussed in detail above in section A, part 2) and author reply #25.

Re-review comment #8: Comment 26: apologies for the lack of clarity. My question was whether naRNA is de novo transcribed. PMA is a strong inducer of transcription and it is interesting to know if naRNA is made in the PMA transcriptional burst. Although an important question, I agree with the authors that these data are not necessary for publication.

Author reply: as not considered necessary for publication this comment is not discussed further here.

Re-review comment #9: Comment 29: the inclusion of the SytoRNA select reagent is helpful and the fact that microscopy experiment is split into different channels is also commendable. However my suggestion was to provide the same split channels for the anti-rRNA Y10b, so that the evidence can be considered collectively. Can the authors rule out nascent LL37 and ribosomal RNA here? To prove that RNA is found in granules, the authors should provide evidence of colocalization of other granule components. Cathelicidin/LL37 is considered to be localized to secondary granules, so lactoferrin or NGAL would be appropriate markers.

Author reply: Unfortunately, the referee had not specified for which panels single channel images are of interest. Of course this could be provided for all images shown. LL37 was used as a well-known granule marker and EM confirms the granular shape. Of course, further amendments would have been possible, especially if e.g. lactoferrin or NGAL had been specified in the initial comment #29.

Re-review comment #10: In response to the comment by reviewer number 1, I did additional novelty search and found the following publication by the same group: <https://pubmed.ncbi.nlm.nih.gov/31913271/> Some of the data from this previous report appears to be repackaged here, which reduces my enthusiasm and makes me more likely to agree with reviewer 1 that this report lacks novelty.

Author reply: discussed in detail above in section A, part 1).

Referee #3

Re-review comment #11: 1. Overall, authors provide verbal but few satisfactory experimental responses to the reviewers' comments (looking at reviewers 1 and 2).

Author reply: several new experiment panels and reviewer figures were added and are described in the 25-page point-by-point reply of section C.

Re-review comment #12: 2. Regarding our comments, authors clarified well the quantification of NETs by counting tiles and harmonizing all other techniques for NET quantification.

Author reply: discussed in detail above in section A, part 2).

Re-review comment #13: 3. Authors respond well to differential RNase digestion of naRNA versus keratinocyte RNA, but authors should also look at protein concentrations.

Author reply: discussed in detail in author reply #41 in section C below; of note, frequently studies report mRNA levels without always validating protein levels.

Re-review comment #14: 4. However, both the discussion on immune tolerance, the paper title and central figure remain unclear, and may be not be clarifiable with the editor's help.

Author reply: we regret not to have entirely convinced referee 3 but had taken the referee's concern seriously by having had the revised paper checked by Dr. Libera Lo Presti, a scientific writing and former *Nature Microbiology* editor.

Re-review comment #15: 5. Please explain why the concept of immune tolerance to self-nucleic acids requires revision. In summary, one of the challenges for a scientist is to relate concepts in a simple and clear manner, and this is not achieved in this work.

Author reply: please see previous author reply above.

C) Original Point-by-point reply EMBOR-2023-58048-T

Please note that Supplemental figures have been renamed EV figures, i.e. Fig. EV1 is now Fig. EV1 in the revised text version.

Referee #1

The authors report that NET-associated RNA (naRNA) drove further NET formation in naive PMNs involving a TLR8-NLRP3-caspase-1-gasdermin D-dependent inflammasome pathway. Moreover, keratinocytes responded to naRNA with expression of psoriasis-related genes (IL17, IL36) via

atypical NOD2-RIPK signaling. Moreover, naRNA drove skin inflammation in vivo that was drastically ameliorated by genetic ablation of RNA sensing. The naRNA-LL37 complex was pre-stored as 'DAMP signal' in neutrophil granules and its activity after release was transient and self-limiting.

Comment #1: The manuscript lacks novelty, as most of the main points in this report have been previously published. For instance, Lande et al. demonstrated in Nature 2007 that the cathelicidin peptide (LL37) forms complexes with extracellular human RNA, and it was elucidated by Ganguly et al. in J. Exp. Med. 2010 that LL37, through TLR8, induces self-activation.

Author reply #1: We concede the point (and state so in the introduction) that RNA has been known to interact with LL37 for TLR activation. However, this work was mainly done with synthetic RNA and natural sources of RNA have not been described in these seminal papers. Our paper described NETs as a plausible and abundant source of LL37 complexing (na)RNA as a TLR activator. The highlight of our work is that this novel source leads to a self-propagating activation loop. Finally, the previous work by Gilliet et al mentioned by referee 1 proposed that RNA-mediated activation of TLRs via LL37 complexation is an “accidental” break of self-nucleic acid tolerance, see (Lande, Chamilos et al. 2015). Instead, we provide compelling evidence that neutrophil RNA-LL37 complexes are abundant composite DAMPs that are configured to activate TLRs, rather than combine accidentally. This is an entirely new concept. Interestingly, we presented this concept to Michel Gilliet himself at the recent Novel Concepts in Innate Immunity 2023 () meeting, where he was an invited speaker. He was highly intrigued and fascinated by our concept and confirmed it as entirely novel and ground-breaking. Hence, lack of novelty should not be a concern.

Comment #2: Furthermore, the concept of extracellular RNA as a potential activator of NETs was previously documented by Smolarz et al. in Front. Cell Infect. Microbiol. 2021.

Author reply #2: That *Candida* leads to NET formation has been known for years and the primary focus of this paper is that *Candida* biofilms contain and release (fungal) nucleic acids that can trigger NET formation from human neutrophils. Smolarz and colleagues go on to delineate the pathway and implicate TLR8. Although this is interesting and fits well with our observation in Fig. 1A, this work does not address or compromise the points made in our work: 1) the authors do not investigate host RNA like we do; 2) they fail to investigate if *Candida*-triggered NETs cause further self-amplification; 3) they do not explore the connection between NET RNA and LL37; 4) They do not show any in vivo relevance for their pathway. Thus, the referee’s comment is correct but is unrelated to our work.

Comment #3: Additionally, the manuscript fails to cite prior work, which reported NETs activating the NLRP3 inflammasome (Hu et al., Arthr. Res. Ther. 2019).

Author reply #3: NETs and inflammasomes are very active research fields and we sincerely apologize if we have overlooked important work. The referee here cites a clinical study in which cell-free DNA and NET-DNA complexes were quantified in sera of healthy controls and patients

suffering from Adult Onset Still's disease. Hu and colleagues did check the activation of the inflammasome by NETs, however, this was only done in immortalized THP-1 cells (i.e. macrophage-like cells), not primary neutrophils. As THP-1 cells, or macrophages in general, do not release RNA and LL37, they do not make any contribution to the concept of NET propagation, whereas neutrophils do. NLRP3 has also been reported to respond to hundreds of stimuli but only for a few a precise pathway has been proposed. As TLR8 was also not implicated by Hu et al., the pathway in this work remains obscure and does not compromise the novelty of our data.

Comment #4: It seems that the authors show bias in their selection of references, neglecting key publications directly related to this topic. Those uncited references should have been incorporated into the manuscript and discussed thoroughly. Furthermore, there is a tendency to reference review articles instead of original research articles.

Author reply #4: In the same way as with the last comment, we apologize for any unintentional bias and are open to suggestions by the referees and editor. Unfortunately, no additional specific references were suggested by referee 1, whilst the ones stated in the preceding comments were discussed (and most already cited). We were also not aware of a strong bias of original papers vs reviews/book chapters: the ratio in our originally submitted paper was 42 original papers to 21 reviews, which appears not unbalanced towards reviews. Moreover, we hold that for summarizing vast areas of research (as is the case for the NET field), citing review articles is plausible and common practice. We therefore do not think that this comment should be taken as a general substantial criticism of our work and hope the abovementioned figures reassure the referee and Editor.

Comment #5: Regarding the few novel points presented in this manuscript, it is essential to note that the data presented are preliminary and do not substantiate conclusions or the title of the manuscript. A more detailed comments are provided below.

Author reply #5: As pointed out above, leading experts attribute a high degree of novelty to our work, which is also confirmed by the other two referees. Moreover, most of the work includes at least 3 repeats and multiple donors and animals so that, technically speaking, the work is not preliminary but built on solid data that was statistically tested. Microscopy was always quantified. We have sought to address all specific criticisms referee 1 made in the following but disagree with the general statement that the work in general is preliminary and unsubstantiated biologically and conceptually.

Comment #6: Major points. 1. Figure 1B is not described in the manuscript.

Author reply #6: Even if correct, a missing callout to a figure should not be regarded as a major critical point. In fact, the criticism does not apply as Figure 1B was described in Line 125 (see underline) as follows "(Fig. 1A, quantified in B; control of antibody staining in Fig. EV1A, B). To avoid this misunderstanding in future, we have amended the phrasing to "quantified in Fig. 1B".

Comment #7: 2. Human neutrophils were treated with 600 nM PMA for 3-4 hours to induce NETs. In the original publication (Brinkmann et al., Science 2004), the authors used 25 nM PMA and observed DNA release within 5-30 min. Is RNA extrusion requiring higher PMA concentration and extended incubation time? If so, the authors should have performed PMA concentration and time-dependent experiments (Fig. 1C, and throughout the manuscript where PMA stimulation was performed).

Author reply #7: Although PMA > 25 nM has been used in multiple other publications (e.g. (Lim, Kuiper et al. 2011, Remijsen, Vanden Berghe et al. 2011, Parker, Dragunow et al. 2012, Masuda, Shimizu et al. 2017, Neubert, Senger-Sander et al. 2019, Arpinati, Shaul et al. 2020)), we agree that this could be viewed as a concern. However, our goal was not to establish stimulation conditions for PMA but to use PMA to robustly generate NETs. 600 nM for several hours seemed suitable to ensure full NETosis in the entire treated PMN population. Nevertheless, to address the comment we generated NETs with 600 nM vs 60 nM PMA, harvested them as before and transferred them to naïve neutrophils. Biological triplicates were performed and quantified. NETs alternatively generated with the ionophore A23187 (suggestion by referee 2) were also analyzed in a similar way. As shown in the reviewer figure below, there is no substantial difference between 600 nM (labeled PMA NETs for consistency with the remaining manuscript) or 60 nM PMA or A23187 NETs, alleviating the reviewers concern (differences in morphology between different fields of view and/or stimuli are normal). Additionally, 600 nM PMA added to Mock NETs and then washed away before harvesting (also suggested by referee 2), does not lead to activation, as expected by us, excluding carry-over of PMA in PMA NETs. Moreover, it seems important to note that PMA is used only as a tool to generate NETs whereas all the remaining work is done with NETs themselves to observe biological effects. Here PMA is used merely as a positive control, as is common practice. It is also vital to note that all PMA NET effects are truly RNA receptor dependent, whereas PMA alone is not TLR sensitive. This confirms further that carry-over or the PMA concentration used is not relevant to the conclusions drawn. As it may lead to confusion, we decided not to include this data in the revised manuscript but have added a comment about PMA concentration to the methods section.

A human PMNs**B**
Confocal microscopy of primary human PMNs treated as indicated (A) and quantification thereof (B). Stimuli are as follows:

- unstimulated
- PMA directly (600 nM)
- Mock NETs prepared as before
- PMA NETs prepared as before (i.e. induced using 600 nM PMA)
- Mock NETs exposed to 600 nM PMA before harvesting
- PMA NETs generated with 60 nM PMA
- A23187 NETs (5 μ M)

All NET preparations were used at 1:100 dilution stimulated for 3 h. Cells were then stained for DNA (Hoechst 33342, white, n = 1 donor, representative images, scale bar 10 μ m). (B) Quantification of (A) using DNA (Hoechst 33342) signal to quantify NET formation (n = 1 donor, mean+SD, each dot represents the number of NET-positive tiles in one image quantified from three images/condition).

Comment #8: 3. According to Fig. 1D, the NET associated RNA (naRNA) is equal to total cel. RI RNA (whole RNA). It is plausible that neutrophils could extrude 100% of total neutrophil RNA unless the cells are completely lysed due to excessive PMA concentration and extended incubation time.

Author reply #8: We are unsure why the referee's conclusion is that naRNA and cellular whole RNA are equal as we think the data make exactly the opposite point: In Fig. 1D (below) the composition of naRNA is not identical to that of whole RNA. For example, whereas whole RNA is dominated by rRNA, rRNA is more balanced with e.g. ncRNA or mRNA in naRNA. This is exactly an argument against the referee's concern that PMA would lead to excessive lysis. We are therefore not sure how to interpret the referee's concern. In any case, even if (hypothetically speaking) 100% of neutrophil RNA ended up in the NETs, this would still qualify as NET-associated RNA and would not change any of the main conclusions related to naRNA.

Fig. 1D from paper.

Comment #9: 4. The RNA dye (Syto RNaselect kit) also binds to DNA, according to information of the manufacturer. Therefore, the experiment shown in Suppl. Movie S2 should have been performed in the presence of DNase I in order to rule out the presence of DNA as part of extracellular traps.

Author reply #9: As DNA is a primary structural feature of NETs shown in countless publications, we do not understand the plausibility of “ruling out” the presence of DNA in NETs. The manufacturer (ThermoFisher) states the following: “The SYTO RNaselect green fluorescent cell stain is a cell-permeant nucleic acid stain that selectively stains RNA. Although virtually nonfluorescent in the absence of nucleic acids, SYTO RNaselect stain exhibits bright green fluorescence when bound to RNA (absorption/emission maxima ~490/530 nm), but only a weak fluorescent signal when bound to DNA”. We concede that this does not entirely exclude binding to DNA. Although there may be superior dyes emerging (albeit not commercially available), multiple studies have relied on (or used as a bench-mark) SYTO RNaselect for meaningful and satisfactory results but have not raised concerns about specificity, e.g. (Li, Kim et al. 2006, Zhou, Liu et al. 2015, Herster, Bittner et al. 2020, Wu, Liu et al. 2020). Moreover, if the majority of SYTO RNaselect signal

was due to DNA binding, we would expect the signals to overlap. As we show below for a representative line plot, this is not the case. This data has been added to the revised manuscript in the **new Fig. EV5C-E**. Finally, to avoid any doubt about the specificity of SYTO RNaselect, we mostly used anti-rRNA staining throughout and even implemented RNA click-chemistry labeling for further verification. We contend this is a relatively rigorous way of testing and controlling specificity. Please also see **Author reply #29**.

New Fig. EV5E,F: Confocal microscopy and line plots of unstimulated primary human PMN. Cells were stained for DNA (Hoechst 33342, white), RNA (SYTO RNaselect, magenta) and LL37 (antibody staining, yellow) and intensity levels quantified using line plots.

Comment #10: 5. Some images contain far less cells (Fig. H upper panel and S2D-E) to be conclusive.

Author reply #10: In unstimulated conditions there are typically less cells as unstimulated neutrophils do not adhere very strongly to the coverslips and therefore some cells may be lost during preparation of the slips for antibody staining. This is a typical technical limitation that cannot be avoided unless far more cells are seeded in the unstimulated condition. However, we would be concerned the greater cell density during the stimulation period would falsify results as it could lead to accidental activation. Additionally, after NET induction it is nearly impossible to identify the number of cells that were imaged retrospectively as demarcating features are completely lost. In general, single images are never conclusive therefore multiple images were always acquired, typically in 3x3 tiles, and quantified in an unbiased manner. We therefore opted for seeding and stimulating the same number of cells per condition (which is common practice), imaging larger fields of view with identical objective/zoom and then quantifying from multiple experiments and images per experiment. Please see also **Author reply #25**.

Comment #11: The scale bar should be given for all microscopic images.

Author reply #11: We apologize for the confusion regarding scale bars and that, indeed, in some images scale bars were missing (e.g. Fig. 1F and 5C). Typically, a scale bar was given for at least one image of a given series and applies to all other images of that series. As all images in one series were acquired with identical objectives and zoom, it is not necessary to show a scale bar in each image but we have now clearly stated in the methods that a given scale bar applies to all images of this series. We have clarified this point in the relevant methods sections.

Comment #12: 6. The NF- κ B results shown in Figure 2A have low statistical significance. Considering that the experiments are performed in an artificial set up (TLRs over-expression in HEK293 cells), one would expect more clear differences.

Author reply #12: We appreciate the referee's concern but according to our statistical knowledge, for $\alpha = 5\%$ (which is common practice), $p < 0.05$ is significant and it is informative but not meaningful how much smaller the p-value is. The differences marked with asterisks in Fig. 2A are therefore unrefutably statistically significant. Maybe the reviewer is concerned about the biological significance. Here we concede that the differences may be smaller than expected by the referee; nevertheless, and taking statistical significance as a meaningful measure, we contend that there are real differences between different TLRs pointing to RNA being the decisive stimulant contained in NETs. We agree with the referee that HEK293T cells are an artificial, yet well-accepted system, in the TLR field. That is also why most of our work is done with primary PMN and keratinocytes.

Comment #13: 7. How were the cells stained in Fig. 1B and 1D? If naRNA or ssRNA were used as stimuli, the dye that sustained the nucleic acids should have depicted the residual RNA.

Author reply #13: In Fig. 1B and S1B a quantification of 1A is shown in which naRNA was stained using Hoechst 3334 for DNA (white) and anti-rRNA Y10b for naRNA (magenta). This antibody does not stain ssRNA40, the short synthetic RNA used for stimulation. Fig. 1D is RNAseq not based on staining, so we are not sure if the reviewer refers to this panel or rather Fig. EV1D? In Fig. EV1D the incorporated 5-ethynyluridine (5-EU) was click-labeled with a fluorescent dye (yellow, total RNA), combined with staining for naRNA (anti-rRNA Y10b, magenta) and DNA (Hoechst 33342, white). This information is provided in the figure legend. We apologize but are not sure what is meant by the remaining comment/question. We are happy to comment if the question is re-phrased more precisely.

Comment #14: To quantify the DNA/RNA association to NETs (Fig. 2C, 2E and 2F), the authors should have collected the supernatant of the stimulated human neutrophils and measured the nucleic acid concentration.

Author reply #14: Quantification was attempted; however, as NETs are very heterogenous and remain attached to the cover slip, quantification in the supernatant is not reliable.

Comment #15: 8. The images for WT and *tlr3*^{-/-} neutrophils (Fig.2D) treated with "Mock NETs" or "PMA NETs" contain much less cells compared to PMA-treated images. One could depict images with comparable numbers of cells per image.

Author reply #15: We presume with Tlr3 the referee refers to Tlr13? Regarding the quantification of unstimulated cells, please see **Author reply #10**.

Comment #16: 9. The authors relied mainly on the effect of inhibitors to make their conclusion regarding the involvement of "NLRP3-Caspase-1-gasdermin D pathway" for naRNA induced NET formation. More specifically, the authors used disulfiram (inhibitor of pyroptosis by blocking gasdermin D pore formation) to block the IL-1b cytokine production using ssRNA + LL37 (Fig. 2G) and concluded that "a unique TLR8-NLRP3-caspase-1-gasdermin D-dependent inflammasome pathway" is involved for the naRNA induced NET formation. Recently, Stojkov et al. (Sci. Signal. 2023) published that GSDMD-knockout neutrophils are capable of NET formation. The same is true for the usage of Cl-amidine (PAD4 inhibitor).

Author reply #16: Indeed, not all stimuli, that are known to induce NET formation, require PAD4 (Fig. 2C) and it is now very clear that NET formation can be dependent and independent of PAD4 (Kenny, Herzig et al. 2017). Since naRNA is found in all types of NETs, whether the process is PAD4-dependent or not seems of lesser importance and hence the specificity of inhibitors was not discussed (but the abovementioned reference added to the revised version). Similarly, contradictory data have been published for GSDMD: those implicating GSDMD (Sollberger, Choidas et al. 2018, Silva, Wanderley et al. 2022) vs showing GSDMD redundancy (Stojkov, Claus et al. 2023). It was therefore outside the scope of solving these issues but since all inhibitors used are very well accepted, we chose to remain silent about these issues and consider it legitimate to draw the conclusions we make for the pathway we studied.

Comment #17: 10. In Fig. 2I, instead of using a genetically modified macrophage-like cell line (BlaER1), the authors could have used bone marrow derived macrophages from WT or TLR8-/- or UNC93B1-/- mice.

Author reply #17: The referee's suggestion is valuable. Unfortunately, TLR8 is dysfunctional in BL6 but replaced by Tlr13 (Eigenbrod and Dalpke 2015). We used *Tlr13* and *Unc93b1* KO cells for neutrophil analysis (Fig. 2D, 4C, 4E and S3, respectively). Since neutrophils and the human system were our main focus and the RNA receptor was obvious in multiple systems (using both inhibitors and KO cells), we did not perform further experiments in BMDM as this offered limited addition to the story.

Comment #18: 11. In Fig. 3B-G and 3J-M as well in Fig. EV4G, H, I, P, the results of qPCR are shown as "rel...(fold change)". It is not clear to what value this quantification was correlated.

Author reply #18: Typically the fold change is relative to "untreated", unless stated otherwise, and we apologize for any omission. We have checked each legend.

Comment #19: 12. The modulation of the immune response upon LL37-RNA had been previously reported (Kato et al. Inflammatio, 2023 and Adase et al. J. Biol. Chem., 2016), making the results shown in Fig. 3 partially redundant

Author reply #19: We thank the referee and have added Kato et al 2023 in the discussion as this may be important. One has to bear in mind that this paper only appeared in June 2023. Our paper was initially put on Biorxiv in July 2022 and then submitted to EMBO Molecular Medicine on 28 June 2023, from where it was transferred to EMBO Reports. Thus, we were not aware of it. At first glance, the paper seems to overlap with our work as the title suggests. However, their focus is on double-stranded RNA working with LL37. That LL37 can work with multiple RNA types is not surprising since interactions are charge-mediated. However, host RNA is rarely double-stranded so that NETs, which contain multiple ssRNA types (Fig. 1D), most likely operate via a different route than the one described by Kato et al. This is suggested by our findings that naRNA sensing is mediated by NOD2 and not RIG-I (ruled out using MAVS siRNA, see Fig. EV4M) as in Kato et al. From the TLR field it is well known that dsRNA and ssRNA are sensed differently by TLR3 vs TLR7/8, so that mechanisms differ, and the differences between NOD2 and RIG-I regarding downstream signaling are certainly greater than amongst different TLRs. Moreover, Kato and colleagues only used poly(I:C), a synthetic dsRNA analog, which lacks any physiological relevance. Our paper, conversely, uses naturally occurring naRNA. Adase JBC 2016 is of course a well-known paper by the Gallo group. Their focus is also on dsRNA and in most of the experiments the authors also rely exclusively on poly(I:C). In a selection of experiments, they use U1 ncRNA, which they suppose can contain dsRNA stretches. They confirm a similar activity as poly(I:C) but do not actually prove that in their hands U1 contains dsRNA. So, most of the claims in both papers are based on poly(I:C) again. Additionally, they report entirely different receptors. We therefore have to strongly disagree with the notion that these two studies make our data in Fig. 3 redundant.

Comment #20: 13. The results shown in Fig. 4 lacks any evidence for in vivo NET formation upon intradermal injection of PMA-NETs in the ears of the mice. Therefore, the induction of inflammation is not properly shown and does not support the conclusion of this manuscript.

Author reply #20: We thank the referee for this comment. We here rely on earlier work, which is cited, showing that IMQ treatment and RNA+LL37 injections lead to NET formation in vivo (Herster, Bittner et al. 2020). We have gone back to the tissue samples and analyzed the IMQ-treated samples for NETs. As shown in the new Fig. 4G and 4H, there is significantly less staining for CitH3, an accepted marker for NETs in tissue (e.g. (Suzuki, Tsuchiya et al. 2022)), confirming that in the IMQ mouse model the lack of RNA sensing restricts NET formation otherwise seen in WT animals. We hope that this additional data addresses the reviewers concern.

Comment #21: 14. The data in Fig. 5 does not follow the trend of the story in the manuscript. For accurate colocalization of RNA with LL37, it would be more appropriate if the authors use the metabolically labelled primary human hematopoietic stem cells (HSCs) PMNs with 5-EU (as shown in Suppl. Fig. EV1D). Immunofluorescent antibody staining in granulocytes could cause nonspecific staining. The isotype control antibody should be included in the experiments shown in Fig. 5.

Author reply #21: As controls were included and shown for many of the other staining experiments (depicted in Fig. EV1A), in our opinion there is no doubt about specificity. As the stem cell approach is extremely cumbersome, we employed it to generally rule out non-specificity but were not able to apply it to all experiments. Moreover, although stem-cell based neutrophils are close to primary cells, we considered it important to work with primary neutrophils to derive the most physiologically relevant data. From all that is known, clearly, the LL37 staining is specific as it is a granular protein (Sorensen, Arnljots et al. 1997). Following the referee's comment, the anti-rRNA staining (for which controls are shown for multiple experiments, see above), has therefore been complemented by SYTO RNaselect staining, which avoids antibodies altogether. Although the referee may not be entirely endorsing this dye (but see **Author reply #9**), the **new Figs. 5D, 5E, EV5C-E** further confirm strong colocalization in granules. Collectively, in our opinion the sum of different analyses supports our conclusions and the descriptions provided put the reader in a position to assess the validity of our claims adequately.

Referee #2

Bork et al present the interesting hypothesis that RNA is a major immunostimulatory component of NETs. Since neutrophils and NETs are implicated in many diseases, this finding has important implications for regulation of inflammation.

Comment #22:The authors provide some compelling evidence on presence of RNA in NETs and on the ability of naRNA to activate a variety of cells. One of the strengths of the report is the use of multiple genetic models to reconstruct a potential pathway. The idea of 'composite DAMPs' that are self-limiting is an appealing and useful concept.

Author reply #22: We thank referee 2 for the enthusiastic feedback.

Comment #23: However some major issues are unaddressed:

My main concern is PMA carryover, specially since this report uses an exceptionally high PMA concentration to induce NETs (50-100 nM is more customary). Although the authors include some good controls, it is essential to repeat this with NETs induced by other stimuli, to prove that sensing of naRNA occurs without use of PMA. The most commonly used stimulus in NET research (after PMA) is calcium ionophore, which should be tested. Alternatively nigericin or ConA-induced NETs. Do these also trigger NETs and macrophage and keratinocyte activation?

Author reply #23: This is a valid concern which we discussed in **Author reply #7**. As neutrophils are clearly the most sensitive cell type, the fact that they did not respond to PMA-treated Mock NETs is an indication that all the data are dependent on NET RNA, not least because genetic RNA receptor KOs do not respond to NETs any more but do respond to the positive control PMA (at higher concentrations than any carry-over would entail). We therefore did not consider it necessary to repeat the same experiments with macrophages or keratinocytes. We hope the data shown in the

referee figure connected to **Author reply #7** above demonstrates that PMA carryover or concentration are not an issue.

Comment #24: Second, 'mock NETs' should be treated with PMA after they are scraped, followed by the same number of washes and this should be tested as negative control.

Author reply #24: We did this and the results show no indication for a non-specific effect, please see **Author reply #7** and **Author reply #23**.

Comment #25: Neutrophils are very sensitive to chemical inhibitors and treatment often induces apoptosis. Since apoptosis and NETosis are alternate pathways, triggering of apoptosis may nonspecifically block NETs. For instance, Cl amidine is known to trigger apoptosis. All inhibitors should be tested in an apoptosis assay, e.g. annexin V binding, in neutrophils, at the concentration at which they are used in the manuscript (and in the same medium). This is particularly important for the TLR8 inhibitor.

Author reply #25: We agree with the reviewer that different types of cell death can be mutually exclusive, but we are not sure which paper the referee is specifically referring to. To our understanding, apoptosis in neutrophils shows distinct morphological features (e.g. apoptotic blebs, see (Douda, Yip et al. 2014) and uniform, unilobar nuclei, see (Gray, Hardisty et al. 2018)) which we do not observe in unstimulated, inhibitor-treated cells. Moreover, the inhibitors do not block or affect NET formation in direct response to PMA. If, as the referee suggests, the inhibitors truly switch the route of cell death to apoptosis (as in the case of Akt inhibitors, (Douda, Yip et al. 2014)), then we would expect differences between PMA without inhibitors vs PMA with Cu-CPT9a, for instance. But we find no evidence. To alleviate the referee's concern we show below higher magnifications of the images contributing to Fig. 2B showing intact PMN morphology under inhibitor treatment. Again, the inhibitor data is confirmed by plausible genetic data from mouse *Tlr13* KO PMN.

Close-up images of inhibitor-treated unstimulated primary PMN from Fig. 2B (stained for DNA using Hoechst, blue, and nRNA using anti-rRNA, red) showing intact morphology and absence of any apoptosis features (typically apoptotic blebs or uniform, unilobar nuclei). This morphology is representative for all experiments using these inhibitors at various concentrations.

Comment #26: Is naRNA signaling dependent on de novo transcription? This is an intriguing possibility, since PMA strongly activates transcription in neutrophils (before NETs are released). Transcriptional inhibitors do not block NETs (PMID: 27310721) so this can be tested with actinomycin or similar RNA pol inhibitor.

Author reply #26: We thank the referee for this suggestion but are not sure how to interpret the referee's comment: Is the question 1) whether transcription in the NET-producing neutrophils is required to ensure NETs contain stimulatory naRNA? Or is the question 2) whether responding neutrophils require de novo transcription when sensing naRNA, signaling via TLR8-NLRP3 and producing NETs as a result? We presume the former and certainly appreciate the intriguing suggestion. Although use of the suggested inhibitors would be a suitable way to test this in either the producing (or, alternatively sensing neutrophils), the details regarding the production of naRNA (question 1), its packaging or further downstream signaling events in sensing neutrophils (question 2) in our opinion are outside the scope of the present manuscript.

Comment #27: Figure 4: have these results been repeated or is this a single experiment? Number of repeats should be indicated for all in vivo experiments

Author reply #27: Yes, there were multiple biological replicates in each of the experiments shown. We apologize if the information stated in figure legend was not entirely clear. We have checked this is clearly stated in the revised manuscript.

Comment #28: Figure 4D-F: this does not directly test the role of naRNA, simply tests the requirement of *tlr13*. There could be various other agonists. As currently written, the section is overinterpreted.

Author reply #28: We concede the point based on the previous data and have amended the interpretation. We did analyze skin samples from IMQ-treated WT and *Tlr13* KO mice and observe less NET formation, see **Author reply #20**. Thus, NET propagation seems affected in *Tlr13* KO. Nevertheless, we have scrutinized this section to avoid overinterpretation of the data.

Comment #29: Figure 5: the idea of pre-packaging of RNA in granules is very unexpected. The evidence provided here is not sufficient for this claim. Firstly, it would be useful to see individual channels in the microscopy images, to assess RNA subcellular localization. As currently presented, it is difficult to see granular RNA localization. Furthermore another reagent is needed to prove that this is not an artefact of the anti-rRNA Y10b antibody. The TEM evidence provided is also rather poor. More compelling evidence is needed for this claim and additional methods, potentially RNA-protein pulldown, RNA FISH combined with IF, or other suitable technique.

Author reply #29: We refer to **Author reply #21** and are happy to provide individual channels (below) and have also included SYTO RNaselect staining to avoid antibody staining (please see **new Fig. 5D, E and S6**). Furthermore, orthogonal projections by extensive z-stack analysis and 3D reconstructions were added:

New Fig. 5D: Immunofluorescence analysis of unstimulated PMN stained with SYTO RNAselect staining combined with anti-LL37 staining and line plot analysis. Further examples are shown in Fig. EV5C.

New Fig. 5E: Orthogonal projections of unstimulated PMN stained with SYTO RNAselect staining combined with anti-LL37 staining. 3D reconstruction shown in new Fig. EV5E.

New Fig. EV5E: 3D reconstruction of z-stack analysis of unstimulated PMN stained with SYTO RNAselect staining combined with anti-LL37 staining.

Merged channels	LL37	RNA	DNA
-----------------	------	-----	-----

Single channel images for examples of unstimulated PMN stained with SYTO RNAselect and anti-LL37 antibodies.

We understand the reviewer's request for additional experimental verification. Typically, co-localization studies by immunofluorescence and additional dual label EM are considered valid demonstrations. The immunofluorescence on ultrathin sections (50-60 nm, Figure 5D) indicates that both signals come from a very small volume. In Fig. 5E (electron microscopy) RNA and LL37 immuno-gold signals are shown to be within 200 nm proximity. Bearing in mind that unavoidable electrostatic repulsion and steric hindrance related to antibody staining with gold particles (Dulhunty, Junankar et al. 1993, Flechsler, Heimerl et al. 2020) prevent co-labelling of the same physical complex, we consider it practically impossible to demonstrate a direct interaction using this technique but the existence of both ribosomal RNA and LL37 in the same granule, which is consistent with ribosomes becoming obsolete in mature murine neutrophils (Zhu, Gong et al. 2017), at least strongly suggests an interaction, which we more carefully state in the re-submission. Following the referee's comments, we did consider, and partially tested, additional experimental approaches:

- Immunoprecipitation of LL37 and verification of RNA binding: We already considered this, but the experiment cannot be done by simply lysing PMN and pulling down LL37 using antibodies: due to their complementary charge it cannot be excluded that LL37 and neutrophil RNA, both freely diffusible in a lysate, might not interact post-lysis. Therefore, the method of nitrogen cavitation (Udby and Borregaard 2007) would be necessary to purify intact granules and then precipitate LL37 to check for co-immunoprecipitated RNA. As soon as our grant money allows, we will spend €5.000 to acquire a nitrogen cavitation device and establish this method according to existing protocols we have from a former colleague in Tübingen, Prof. Dominik Hartl, an expert on neutrophils who also fully endorses this present work. However, at the present time, no such device is accessible to us in a setting where freshly isolated neutrophils could be fractionated.

- Proximity ligation assay: We have already explored this proximity-based method as an alternative to the immunofluorescence on ultrathin sections presented in Fig. 5F. However, given the size of the 1°+2° antibody sandwich, we would only improve from 200 nm in Fig. 5F to 80 nm (see e.g. (Klasener, Maity et al. 2014)). We explored the method already nevertheless and could see a signal when using the anti-rRNA Y10b and anti-LL37 primary antibodies, in combination with the Duolink® Proximity Ligation Assay Reagents from Merck and subsequent confocal microscopy analysis. However, neutrophils seem to take up the far-red dye provided with the kit and therefore the background by staining reagents-only is high (although unfortunately not typically shown in publications). To show our efforts we append below an interim analysis example. However, this could not be optimized further as reagents were on month-long backorder.

Preliminary confocal microscopy analysis of proximity ligation assay (PLA) of (na)RNA and LL37 in primary human PMN. Unstimulated or PMA (600 nM, 3 h) treated cells were fixed and PLA was performed according to the manufacturer's instructions (Duolink® Proximity Ligation Assay, Merck) with anti-rRNA Y10b (mouse, Santa Cruz Biotechnology, sc-33678) and anti-LL37 (rabbit, LSBio, LS-B6696-500) primary antibodies and subsequent PLUS (mouse, Merck, DUO92001-30RXN) and MINUS (rabbit, Merck, DUO92005-30RXN) secondary antibody probes. Amplification was performed using the Duolink® In Situ Detection Reagents Far red (Merck, DUO92013-30RXN).

- RNA FISH combined with IF is indeed a method that may be suitable and one could envisage using an rRNA probe. This would allow visualizing naRNA but to show LL37, again, antibody staining would be necessary. As the method is not established and we dedicated time to PLA, we were unable to use this method within the period of time allocated to revisions.

In summary, we provide additional images and additional SYTO RNaselect staining, orthogonal projects and 3D reconstruction to corroborate our findings but have also tuned down our statements. But to explore the co-localization in greater detail – owed to high technical requirements to do so – would, in our opinion have to be considered future work.

Referee #3

Authors explore the role of NET associated RNA (naRNA) as a component of neutrophil extracellular traps (NETs) and its immunostimulatory properties in the skin.

Summary of paper:

1. naRNA in NETs: The paper establishes that naRNA is a common component of NETs formed by human polymorphonuclear neutrophils (PMNs) in response to various stimuli, including PMA, LL37-RNA complexes, nigericin, and live pathogens like *Candida albicans*. This is confirmed through confocal microscopy and metabolic labeling experiments.
2. naRNA as a DAMP: The study demonstrates that naRNA serves as a potent damage-associated molecular pattern (DAMP) that can trigger NET formation in primary PMNs. It is proposed that naRNA, in conjunction with LL37, acts as a 'composite DAMP' to activate neutrophils.
3. RNA Sensors and Inflammasome Activation: The research identifies the involvement of RNA sensors TLR8/Tlr13 and the NLRP3 inflammasome in naRNA-mediated NET formation.
4. Immune Cell Activation: The paper shows that naRNA can activate not only neutrophils but also other immune cells and these responses are mediated by RNA sensing.
5. naRNA in Keratinocytes: The study demonstrates that naRNA can activate keratinocytes, leading to the induction of psoriasis-associated genes. This indicates the potential role of naRNA in skin inflammation.
6. In Vivo DAMP Activity: The research extends to in vivo models, where intradermal injection of RNA-stabilized PMA NETs in mice induces skin inflammation. Tlr13-deficient mice show reduced skin inflammation in a psoriasis model.
7. Pre-Association of RNA and LL37: Pre-association of RNA and LL37 within neutrophil compartments occurs before NET extrusion, indicating that neutrophils contain preformed naRNA-LL37 DAMPs.

Comment #30: This is a hard-to-read manuscript mainly due to a poor language style.

Author reply #30: We thank the reviewer for this feedback. We were surprised because members of our team are English native speakers (e.g. N. Archer, Baltimore). We nevertheless contacted our Cluster of Excellence's scientific writer, Libera Lo Presti, a former editor for *Nature Microbiology*, for additional editorial input and hope to have amended the style to a level satisfactory to the referee.

Comment #31: However, the main limitation of this manuscript is that authors are looking at only one side of biology and that is protected RNA, while they should also prove their concept with RNase-free NETs.

Author reply #31: We concede to the referee that in some of the in vivo work, we also used protected RNA in Fig. 4A-C. However, for some of the in vivo work (Fig. 4A) and practically all of the in vitro work, we used also non-protected RNA, showing significant differences. We think that the higher response is not due to the presence of the RNase inhibitor but rather a higher starting concentration of RNA in the PMA NETs. In our opinion, the main conclusions can be derived from data based on non-protected RNA and are not compromised by this experimental setup. Moreover, it would be outside the scope of this revision and problematic with regards to 3R guidelines to

repeat all in vivo experiments in Fig. 4 with non-protected RNA. Considering the referee's comment we have amended the wording in this section and have added a general comment at the beginning of the discussion regarding the use of RNase inhibitors.

Comment #32: Major Points:

1. Is the pre-association of RNA and LL37 sufficiently shown? The data of Figure 5C show the colocalization to the granule compartment but do not show a bound complex. To convince, please show a control granule protein of which you know it is not associated with RNA. Or precipitate the RNA/LL37 complex from isolated granules.

Author reply #32: As discussed above in **Author reply #29** LL37 is a known granule protein (Sorensen, Arnljots et al. 1997) and further staining with SYTO RNaselect was provided (new Fig. EV5C-E). The technical limitations regarding the immunoprecipitation are discussed above..

Comment #33: 2. How is RNA bound to DNA? Via LL37? Figure 1 shows that RNA is associated with DNA strands.

Author reply #33: This is an intriguing question and one could speculate that LL37 functions as a connector (due to some of its positive charges connecting to the DNA backbone) or there are potentially partial areas of hybridization with the DNA backbone. Apart from the fact that it seems virtually impossible to analyze this aspect experimentally, we considered it not essential for the main conclusions drawn and hope that the referee agrees that this may be part of follow-up work in which the pre-association of naRNA and LL37 and the tethering to the NET is investigated in detail.

Comment #34: 3. The entire manuscript is built on the concept that effects of RNA are shown in the presence of RNase inhibitor. Authors need to complement their experiments by introducing a system in which the effects are abrogated in the absence of RNA. For example, in the animal experiments, authors should inject PMA NETs + RNase vs PMA NETs alone.

Author reply #34: We respectfully disagree with the reviewer that, quote, "the entire manuscript" is based on stabilized RNA as outlined in **Author reply #31**. Moreover, as intended by the referee, we use RNase digestion to show that in unprotected NETs, removal of the RNA abrogates the effects. This was already included in Figs. 1G, 4B-D and is further corroborated by genetic removal of RNA receptors and use of inhibitors. We considered the proposed additional in vivo experiment, but discarded the idea as: 1) Fig. 4A already shows a side-by-side comparison of PMA NETs with stabilized and non-stabilized PMA NETs; and 2) because finding the right conditions would require extensive titration of all reagents not only the stimulus, but also natural RNase (Probst, Brechtel et al. 2006) and endogenous RNase inhibitor (Abtin, Eckhart et al. 2009) levels in the skin as well as potential effects of RNases on TLR activation (Greulich, Wagner et al. 2019) have to be taken into account. The use of an RNase inhibitor as shown in Figs. 4A-C, primarily during the preparation of the NETs, avoided some of the complications arising from use of an active enzyme. In summary, the limitations of our approach have been highlighted to the reader to acknowledge the referee's

concerns at the beginning of the discussion: “Our conclusions largely focus on in vitro observations using primary human cells. As in vivo responses to NETs prepared without RNase inhibitor are relatively small, further work will be required to delineate the complex interplay between naRNA, endogenous RNases and endogenous RNase inhibitors more fully”.. We hope that this acknowledges the limitations of our study appropriately.

Comment #35: 4. Regarding your quantification of NETs DNA and RNA, you mention quantification of NETs DNA and RNA in immunofluorescence images but do not explain the units used, such as "rel. ROI number/ROI average size." There's a need for clarification regarding the calculation and presentation of ROIs, and an example of ROIs with and without NETs should be provided.

Author reply #35: We appreciate the referee’s concern and have amended the methods description accordingly. As the selection of ROIs is indeed challenging due to the erratic shape of the NETs in vitro we now harmonized quantification of in vitro results throughout the manuscript as “tiles showing NETs”, i.e. showing diffuse DNA signal. An example is shown below. In brief, an 8 by 8 grid is superimposed on each image (all images taken with the same settings, including objective, zoom and resolution) and then scored manually. A tile was counted as NET positive whenever a DNA (Hoechst) signal was clearly detected in shape of a fiber outside of a cell with a destroyed nucleus. A tile was counted as negative when only intact cells were observed within this respective area. An example is provided below:

Example of 8x8 tile counting for NETs. Left: original confocal image with DNA visualized by Hoechst staining. Right: scored tiles used for quantification labelled in white.

Comment #36: 5. Figure 1G is supposed to complement Figure 1E, but different methods are used to calculate and show the presence of NETs. The paper should present one or both analysis methods for both experiments. Please, standardize your dispersion versus tile counting methodology.

Author reply #36: As outlined in **Author reply #35**, we have standardized the methodology to exclusively use tile counting.

Comment #37: 6. To support the claim that naRNA, and not DNA, is the relevant immunostimulatory component for NETs, the paper should include experiments where PMA NETs are treated with DNase and DNase inhibitors.

Author reply #37: This control experiment also occurred to us and DNase treatment was indeed included in Fig. EV3C, see Line 244. Moreover, since genetics are more reliable than DNA digestion,

we used *Tlr9* KO BM-PMN cells to exclude an involvement of DNA. On the other hand, KOs for RNA receptors were used, collectively providing a very clear picture that RNA, and not DNA, is responsible for the observed effects.

Comment #38: 7. While the in vivo experiments are interesting, they mainly focus on the presence and absence of extracellular RNA rather than NETs-associated extracellular RNA.

Author reply #38: As there is no unequivocal marker to distinguish between these two types of RNA but we can clearly show that NETs contain immunostimulatory RNA, we consider it acceptable to argue that the presence of NETs in vivo entails naRNA-mediated effects, especially if confirmed by the effects of an RNA receptor (*Tlr13*) KO. We conducted additional staining of skin samples from WT and *Tlr13* KO IMQ-treated animals for the PMN marker MPO and the NET marker CitH3. This shows a clear reduction of both signals in *Tlr13* KO. An example as well as an unbiased quantification are provided in the new Fig. 4G, H. We trust this addresses this point as it both shows the presence of the neutrophils and NET in the tissue as a consequence of IMQ treatment as well as their reduction in *Tlr13* KO.

Comment #39: The paper should address the potential effects of RNase inhibitors on inflammation in an in vivo setting. Although you show 'RNase inhibitor+RPMI' you do not show the RNase inhibitor in the context of inflammation that is not driven by NETs.

Author reply #39: Please see **Author reply #31**. Due to the abovementioned complexity, the exact role of endogenous RNases and endogenous RNase inhibitors will require a dedicated research effort outside the scope of the present work. We have added this as a limitation to the present work as described in the abovementioned reply.

Comment #40: 8. In Figure 3 pre-digestion of NETs with RNase A abolished the stimulatory effects of NETs on keratinocytes, whereas pre-digestion with DNase I did not. It is hard to believe that the stimulatory effect of NETs on keratinocytes is abrogated by RNase but not by DNase. NETs are DNA scaffolds. If you break them down, the associated RNA should be affected. Please, show in the supernatants the size of DNA fragments (note comments 2 and 12).

Author reply #40: As discussed above in **Author reply #14**, quantification is not technically straightforward. Moreover, we would like to point out that the DNase digestion was done on already harvested NETs, so naRNA contained in the NETs is not removed, even if it is released from the DNA strands. Therefore, it would be present when added to the keratinocytes. Moreover, we rule out the involvement of DNA by alternative methods, e.g. KD of MyD88 and STING, which would be expected to affect TLR9 or cGAS sensing pathways. These results were consistent with no effect of DNase treatment.

Comment #41: 9. The paper should address whether RNase treatment affects the degradation of keratinocyte RNA in addition to NET RNA. A key point you make is to show the different stimulatory potential between RNase and DNase treated NETs. In Figure 3b you show that RNase but not DNase abolishes the stimulatory effects of NETs on keratinocytes, however your readout is using qPCR to

quantify the RNA of the psoriasis-associated genes. How do you rule out whether the RNase also degrades the keratinocyte, psoriasis associated RNA? Please, show the protein production in another experiment.

Author reply #41: We are not able to provide protein data as there are no remaining samples and the experiment cannot be repeated easily. But we provide below a figure showing plotted Ct values for the housekeeping gene *TBP*. Evidently, there is no visual or statistically significant difference in CT values, and hence mRNA levels, of the housekeeper for samples in which the PMA NETs had been RNase or DNase pre-treated. This is also not surprising since the RNase was added after NET harvest and this preparation was diluted 1:50 into to tissue culture media along with the contained NETs to stimulate the keratinocytes. This high dilution of RNase in the media is unlikely to be sufficient to significantly digest intracellular RNA in live keratinocytes after harvesting of the cells for RNA isolation. The stimulant and any RNase contained in the media were also aspirated and the cells washed before they were harvested and lysed for RNA isolation. Thus, the likelihood of RNase in the stimulus to carry over into lysates and hence RNA preps, to then have an effect on the prepared RNA is very low. In any case, the housekeeper analysis unequivocally rules this out.

Graph showing housekeeper *TBP* mRNA CT values from qPCR experiments in Fig. 3B-C, grouped by 'all values' except those from wells in which PMA NETs had been RNase or DNase pre-treated, those from wells with 'RNase pre-treated' PMA NETs and those from wells with 'DNase pre-treated' PMA NETs. Differences were non-significant according to Kruskal-Wallis non-parametric ANOVA test with Dunn's correction for multiple testing.

Comment #42: 10. When discussing the differences in ear thickness and epidermal thickness in wild-type and Tlr13 KO animals, it would be important to include images of the ears from each group for visual reference.

Author reply #42: Cross sections were included for Fig. 4F as common place for quantifying inflammation in the imiquimod model. We further added IF staining to highlight neutrophil

infiltration and NET formation (new Fig. 4G and 4H) following the suggestion, see **Author reply #20**. Unfortunately, photographic images of the entire ear were not taken at the time as this is not typically done, and we are not permitted by law to use further animals for the sole purpose of acquiring such new images. We hope that by adding the IF analysis there is sufficient information to assess skin inflammation.

Comment #43: 11. Are RNA-LL37 complexes released during degranulation, outside of NETosis?

Author reply #43: This is an intriguing possibility that we could address in future work not directly related to the present manuscript focusing on NETs. So far, we have no evidence that release of LL37-naRNA would precede or would be decoupled from NET formation in any way.

Comment #44: 12. With regards to fresh and aged/degraded NETs I do not see any characterization of your fresh and aged NETs with serum or medium. What happens to the DNA, RNA and proteins after incubation? Serum contains RNases, DNases, and proteases which all may play a role here.

Author reply #44: Indeed, our assumption is that serum contains all of these and that these may be physiological ways in which the inflammatory potential of NETs is reduced over time. We provide new immunofluorescence staining of fresh vs aged NETs in the **new Fig. EV5F**. Moreover, we have now not only tested fresh vs aged NETs on keratinocytes (Fig. 5G) but also on primary PMN (**new Fig. 5F**), showing the same effect. Although there may be multiple ways in which NETs lose their reactivity, these clearly relate to RNA, consistent with our findings that NETs activate primary PMN and keratinocytes via RNA receptors and RNA dependently.

New Fig. EV5F: Confocal microscopy of PMA-induced NETs from primary human PMNs incubated for 30 min or 4 h with human serum and stained for DNA (Hoechst 33342, white) and naRNA (anti-rRNA Y10b, magenta or red). Lower magnification (left, scale bar = 50 μ m) and higher magnification (right, scale bar = 20 μ m) for one representative of n=2 biological replicates shown.

New Fig. 5F: Quantification of NET release by primary human PMNs using confocal microscopy. Human PMNs were stimulated with fresh or old NETs generated by incubation in PMN culture medium or human serum treatment, respectively. Afterwards PMNs were stained for DNA (Hoechst 33342) and quantified as illustrated above (n= 2 donors, mean+SD, each dot represents the number of NET-positive tiles in one image quantified from three images/condition, *p<0.05 according to Kruskal-Wallis test with Dunn's correction)

Comment #45: 13. Simplify your language, e.g. avoid purposefully and accidentally and barbed wire roadblocks, etc. Along the same lines, the manuscript title needs to be rephrased.

Author reply #45: We thank the reviewer for this comment but have also had positive responses to explain biological phenomena in this way. Referee 1 and 2 apparently did not have negative feedback about style. Regarding this and the title we have nevertheless sought feedback from our Cluster's scientific writer, herself previously an editor at *Nature Microbiology*, and by the *EMBO Reports* editor. The revised manuscript text and title are the result of this consultation and hopefully agreeable to referee 3.

Comment #46: 14. The paragraph discussing the trade-off between DAMP activity and RNA's short-lived nature should be revised or removed from the results section, as it is more of a discussion point.

Author reply #46: We thank referee 3 for flagging this up and have amended the text accordingly.

Comment #47: 15. The graphical abstract is not clear.

Author reply #47: We appreciate the feedback and will confer with the editor about the final layout.

Comment #48: 16. Reference to immune tolerance is not subject of this work.

Author reply #48: As outlined in **Author reply #1**, LL37 interacting with RNA and DNA has been viewed to act as a breaker of tolerance to self-nucleic acids in disease (Lande, Chamilos et al. 2015). We show that it interacts with RNA to act as a DAMP in healthy human donor neutrophils. Hence, the concept of immune tolerance to self-nucleic acids requires revision. This is what we sought to express in the title. Again, we will seek the editor's advice on the final phrasing of the title.

References cited in point-by-point reply (section C) and appeal (section A)

- Abtin, A., L. Eckhart, M. Mildner, M. Ghannadan, J. Harder, J. M. Schroder and E. Tschachler (2009). "Degradation by stratum corneum proteases prevents endogenous RNase inhibitor from blocking antimicrobial activities of RNase 5 and RNase 7." J Invest Dermatol **129**(9): 2193-2201.
- Arpinati, L., M. E. Shaul, N. Kaisar-Iluz, S. Mali, S. Mahroum and Z. G. Fridlender (2020). "NETosis in cancer: a critical analysis of the impact of cancer on neutrophil extracellular trap (NET) release in lung cancer patients vs. mice." Cancer Immunol Immunother **69**(2): 199-213.
- de Bont, C. M., W. J. H. Koopman, W. C. Boelens and G. J. M. Pruijn (2018). "Stimulus-dependent chromatin dynamics, citrullination, calcium signalling and ROS production during NET formation." Biochim Biophys Acta Mol Cell Res **1865**(11 Pt A): 1621-1629.
- Douda, D. N., L. Yip, M. A. Khan, H. Grasemann and N. Palaniyar (2014). "Akt is essential to induce NADPH-dependent NETosis and to switch the neutrophil death to apoptosis." Blood **123**(4): 597-600.
- Dulhunty, A. F., P. R. Junankar and C. Stanhope (1993). "Immunogold labeling of calcium ATPase in sarcoplasmic reticulum of skeletal muscle: use of 1-nm, 5-nm, and 10-nm gold." J Histochem Cytochem **41**(10): 1459-1466.
- Eigenbrod, T. and A. H. Dalpke (2015). "Bacterial RNA: An Underestimated Stimulus for Innate Immune Responses." J Immunol **195**(2): 411-418.
- Flechsler, J., T. Heimerl, C. Pickl, R. Rachel, Y. D. Stierhof and A. Klingl (2020). "2D and 3D immunogold localization on (epoxy) ultrathin sections with and without osmium tetroxide." Microsc Res Tech **83**(6): 691-705.
- Flores, R., S. Dohrmann, C. Schaal, A. Hakkim, V. Nizet and R. Corriden (2016). "The Selective Estrogen Receptor Modulator Raloxifene Inhibits Neutrophil Extracellular Trap Formation." Front Immunol **7**: 566.
- Gray, R. D., G. Hardisty, K. H. Regan, M. Smith, C. T. Robb, R. Duffin, A. Mackellar, J. M. Felton, L. Paemka, B. N. McCullagh, C. D. Lucas, D. A. Dorward, E. F. McKone, G. Cooke, S. C. Donnelly, P. K. Singh, D. A. Stoltz, C. Haslett, P. B. McCray, M. K. B. Whyte, A. G. Rossi and D. J. Davidson (2018). "Delayed neutrophil apoptosis enhances NET formation in cystic fibrosis." Thorax **73**(2): 134-144.
- Gray, R. D., C. D. Lucas, A. MacKellar, F. Li, K. Hiersemenzel, C. Haslett, D. J. Davidson and A. G. Rossi (2013). "Activation of conventional protein kinase C (PKC) is critical in the generation of human neutrophil extracellular traps." J Inflamm (Lond) **10**(1): 12.
- Greulich, W., M. Wagner, M. M. Gaidt, C. Stafford, Y. Cheng, A. Linder, T. Carell and V. Hornung (2019). "TLR8 Is a Sensor of RNase T2 Degradation Products." Cell **179**(6): 1264-1275 e1213.
- Herster, F., Z. Bittner, N. K. Archer, S. Dickhofer, D. Eisel, T. Eigenbrod, T. Knorpp, N. Schneiderhan-Marra, M. W. Loffler, H. Kalbacher, T. Vierbuchen, H. Heine, L. S. Miller, D. Hartl, L. Freund, K. Schakel, M. Heister, K. Ghoreschi and A. N. R. Weber (2020). "Neutrophil extracellular trap-

associated RNA and LL37 enable self-amplifying inflammation in psoriasis." Nat Commun **11**(1): 105.

Kenny, E. F., A. Herzig, R. Kruger, A. Muth, S. Mondal, P. R. Thompson, V. Brinkmann, H. V. Bernuth and A. Zychlinsky (2017). "Diverse stimuli engage different neutrophil extracellular trap pathways." Elife **6**.

Klasener, K., P. C. Maity, E. Hobeika, J. Yang and M. Reth (2014). "B cell activation involves nanoscale receptor reorganizations and inside-out signaling by Syk." Elife **3**: e02069.

Lande, R., G. Chamilos, D. Ganguly, O. Demaria, L. Frasca, S. Durr, C. Conrad, J. Schroder and M. Gilliet (2015). "Cationic antimicrobial peptides in psoriatic skin cooperate to break innate tolerance to self-DNA." Eur J Immunol **45**(1): 203-213.

Li, Q., Y. Kim, J. Namm, A. Kulkarni, G. R. Rosania, Y. H. Ahn and Y. T. Chang (2006). "RNA-selective, live cell imaging probes for studying nuclear structure and function." Chem Biol **13**(6): 615-623.

Lim, M. B., J. W. Kuiper, A. Katchky, H. Goldberg and M. Glogauer (2011). "Rac2 is required for the formation of neutrophil extracellular traps." J Leukoc Biol **90**(4): 771-776.

Masuda, S., S. Shimizu, J. Matsuo, Y. Nishibata, Y. Kusunoki, F. Hattanda, H. Shida, D. Nakazawa, U. Tomaru, T. Atsumi and A. Ishizu (2017). "Measurement of NET formation in vitro and in vivo by flow cytometry." Cytometry A **91**(8): 822-829.

Neubert, E., S. N. Senger-Sander, V. S. Manzke, J. Busse, E. Polo, S. E. F. Scheidmann, M. P. Schon, S. Kruss and L. Erpenbeck (2019). "Serum and Serum Albumin Inhibit in vitro Formation of Neutrophil Extracellular Traps (NETs)." Front Immunol **10**: 12.

Parker, H., M. Draganow, M. B. Hampton, A. J. Kettle and C. C. Winterbourn (2012). "Requirements for NADPH oxidase and myeloperoxidase in neutrophil extracellular trap formation differ depending on the stimulus." J Leukoc Biol **92**(4): 841-849.

Probst, J., S. Brechtel, B. Scheel, I. Hoerr, G. Jung, H. G. Rammensee and S. Pascolo (2006). "Characterization of the ribonuclease activity on the skin surface." Genet Vaccines Ther **4**: 4.

Remijsen, Q., T. Vanden Berghe, E. Wirawan, B. Asselbergh, E. Parthoens, R. De Rycke, S. Noppen, M. Delforge, J. Willems and P. Vandenabeele (2011). "Neutrophil extracellular trap cell death requires both autophagy and superoxide generation." Cell Res **21**(2): 290-304.

Shrestha, B., T. Ito, M. Kakuuchi, T. Totoki, T. Nagasato, M. Yamamoto and I. Maruyama (2019). "Recombinant Thrombomodulin Suppresses Histone-Induced Neutrophil Extracellular Trap Formation." Front Immunol **10**: 2535.

Silva, C. M. S., C. W. S. Wanderley, F. P. Veras, A. V. Goncalves, M. H. F. Lima, J. E. Toller-Kawahisa, G. F. Gomes, D. C. Nascimento, V. V. S. Monteiro, I. M. Paiva, C. Almeida, D. B. Caetite, J. C. Silva, M. I. F. Lopes, L. P. Bonjorno, M. C. Giannini, N. B. Amaral, M. N. Benatti, R. C. Santana, L. E. A. Damasceno, B. M. S. Silva, A. H. Schneider, I. M. S. Castro, J. C. S. Silva, A. P. Vasconcelos, T. T. Goncalves, S. S. Batah, T. S. Rodrigues, V. F. Costa, M. C. Pontelli, R. B. Martins, T. V. Martins, D. L. A. Esposito, G. C. M. Cebinelli, B. A. L. da Fonseca, L. O. S. Leiria, L. D. Cunha, E. Arruda, H. I. Nakaia,

A. T. Fabro, R. D. R. Oliveira, D. S. Zamboni, P. Louzada-Junior, T. M. Cunha, J. C. F. Alves-Filho and F. Q. Cunha (2022). "Gasdermin-D activation by SARS-CoV-2 triggers NET and mediate COVID-19 immunopathology." Crit Care **26**(1): 206.

Sollberger, G., A. Choidas, G. L. Burn, P. Habenberger, R. Di Lucrezia, S. Kordes, S. Menninger, J. Eickhoff, P. Nussbaumer, B. Klebl, R. Kruger, A. Herzig and A. Zychlinsky (2018). "Gasdermin D plays a vital role in the generation of neutrophil extracellular traps." Sci Immunol **3**(26).

Sorensen, O., K. Arnljots, J. B. Cowland, D. F. Bainton and N. Borregaard (1997). "The human antibacterial cathelicidin, hCAP-18, is synthesized in myelocytes and metamyelocytes and localized to specific granules in neutrophils." Blood **90**(7): 2796-2803.

Sosa-Luis, S. A., W. J. Rios-Rios, A. E. Gomez-Bustamante, M. L. A. Romero-Tlalolini, S. R. Aguilar-Ruiz, R. Baltierrez-Hoyos and H. Torres-Aguilar (2021). "Structural differences of neutrophil extracellular traps induced by biochemical and microbiologic stimuli under healthy and autoimmune milieus." Immunol Res **69**(3): 264-274.

Stojkov, D., M. J. Claus, E. Kozlowski, K. Oberson, O. P. Scharen, C. Benarafa, S. Yousefi and H. U. Simon (2023). "NET formation is independent of gasdermin D and pyroptotic cell death." Sci Signal **16**(769): eabm0517.

Suzuki, K., M. Tsuchiya, S. Yoshida, K. Ogawa, W. Chen, M. Kanzaki, T. Takahashi, R. Fujita, Y. Li, Y. Yabe, T. Aizawa and Y. Hagiwara (2022). "Tissue accumulation of neutrophil extracellular traps mediates muscle hyperalgesia in a mouse model." Sci Rep **12**(1): 4136.

Udby, L. and N. Borregaard (2007). "Subcellular fractionation of human neutrophils and analysis of subcellular markers." Methods Mol Biol **412**: 35-56.

Wu, Y., Y. Liu, C. Lu, S. Lei, J. Li and G. Du (2020). "Quantitation of RNA by a fluorometric method using the SYTO RNASelect stain." Anal Biochem **606**: 113857.

Yang, F., X. Luo, G. Luo, Z. Zhai, J. Zhuang, J. He, J. Han, Y. Zhang, L. Zhuang, E. Sun and Y. He (2019). "Inhibition of NET formation by polydatin protects against collagen-induced arthritis." Int Immunopharmacol **77**: 105919.

Zhou, B., W. Liu, H. Zhang, J. Wu, S. Liu, H. Xu and P. Wang (2015). "Imaging of nucleolar RNA in living cells using a highly photostable deep-red fluorescent probe." Biosens Bioelectron **68**: 189-196.

Zhu, Y., K. Gong, M. Denholtz, V. Chandra, M. P. Kamps, F. Alber and C. Murre (2017). "Comprehensive characterization of neutrophil genome topology." Genes Dev **31**(2): 141-153.

Dear Prof. Weber,

Thank you for the re-submission of your revised manuscript to EMBO reports, asking us to reconsider the previous decision on your study. I have now received the report from the arbitrator I contacted, you will find below. As you will see, the arbitrator supports the publication of your study in EMBO reports. The arbitrator has some comments and suggestions to improve the manuscript I ask you to address in a final revised manuscript.

As indicated by the arbitrator, it seems appropriate to tone down the immune tolerance conclusions (as per comment 48) and to use more objective language, and not to use anthropomorphized descriptions of neutrophil intent. Please revise text and title accordingly and, as suggested in your appeal letter, amend wording that may have been overstated.

Please also provide a final p-b-p-response addressing the remaining points.

- Please provide the abstract written in present tense throughout.
- Please update the 'Data availability section' (providing link and accession number) and make sure that the datasets are public latest on the publication date of the manuscript.
- We now use CRediT to specify the contributions of each author in the journal submission system. CRediT replaces the author contribution section. Please use the free text box to provide more detailed descriptions and do NOT provide your final manuscript text file with an author contributions section. See also our guide to authors:
<https://www.embopress.org/page/journal/14693178/authorguide#authorshippinguidelines>
- Per journal policy, we do not allow 'data not shown', which is stated three times in the manuscript (pages 8, 18 and 37). All data referred to in the paper should be displayed in the main or Expanded View figures, or an Appendix. Thus, please add these data (or change the text accordingly if these data are not central to the study). See:
<https://www.embopress.org/page/journal/14693178/authorguide#unpublisheddata>
- Please remove the list of abbreviations from the manuscript text file. Instead, please define each abbreviation the first time it is used in the text.
- Please make sure that the number "n" for how many independent experiments were performed, their nature (biological versus technical replicates), the bars and error bars (e.g. SEM, SD) and the test used to calculate p-values is indicated in the respective figure legends (for main, EV and Appendix figures) of the final revised manuscript. Please also check that all the p-values are explained in the legend, and that these fit to those shown in the figure. Please provide statistical testing where applicable. Please avoid the phrase 'independent experiment', but clearly state if these were biological or technical replicates. Please also indicate (e.g. with n.s.) if testing was performed, but the differences are not significant. In case n=2, please show the data as separate datapoints without error bars and statistics. See also:

<http://www.embopress.org/page/journal/14693178/authorguide#statisticalanalysis>

If $n < 5$, please show single datapoints for diagrams. Moreover:

- Please note that the legend for figure 2i is incorrectly labelled as 2h. This needs to be rectified.
- Please note that for the figure 5h, p-value and statistical test are indicated in the legend. However, comparison for the same, "" has not been represented in the figure. Please rectify this in the figure or legend as applicable.
- Although 'n' is provided, please describe the nature of entity for 'n' in the legends of figures 2a, e, g, i; EV 2a.
- Please note that the error bars are not defined in the legends of figures 1b; 2e; EV 1f.
- Please note that the white arrowheads are not defined in the legend of figure 2b, d; 5c, e-f, EV 1a-d; EV 2c-e; EV 3a-d. This needs to be rectified.
- Please add to each legend a 'Data Information' section explaining the statistics used or providing information regarding replicates and scales.

- Please make sure that all the funding information is also entered into the online submission system and that it is complete and similar to the one in the acknowledgement section of the manuscript text file. Presently, grant TR156; Pfizer and Boehringer Ingelheim; the University of Tübingen, the University Hospital Tübingen, is not entered in the submission system, and grants INST 35/1259-2 and INST 35/1259-3 seem missing from the acknowledgements. Please check. Finally, please provide the last sentence in the Acknowledgments (regarding EXC 2180 and EXC 2124) in English.

- Please make sure that all figure panels are called out separately and sequentially (main and EV figures). Presently, it seems a callout for Fig. 4H is missing. Please check.

- Please remove the reagents tables (supplementary tables) from the main manuscript text file. I have attached templates for the reagents and tools table in word or excel format. Please upload the filled in table to the manuscript tracking system as 'Reagent Table' file. Please also adjust any callouts. The example linked below shows how the table will display in the published article and includes examples of the type of information that should be provided for the different categories of reagents and tools. Please list your reagents/tools using the categories provided in the template and do not add additional subheadings to the table. Reagents/tools that do not fit in any of the specific categories can be listed under "Other":
https://www.embopress.org/pb%2Dassets/embo-site/msb_177951_sample_FINAL.pdf

- The correct nomenclature for the movies should be Movie EV1 and Movie EV2. Please correct the names and the callouts. Moreover, please remove the movie legends from the manuscript text file and provide these in a readme.txt file zipped up (and uploaded) together with each movie file.

- There seems to be a re-use of panels between Figure 2b,d and h and Figure EV3a, b and d, between Figure 5C , 5E and Figure EV5 D, and between Figure EV1B and Figure EV2C. See the attached report. Please clarify and clearly state this in the corresponding figure legends.

In addition, I would need from you:

Best,

Arbitrator:

My independent review:

RNA is present on NETs released by a variety of different agonists (mouse and human) - shown by antibody staining as well as 5-EU incorporation with click chemistry fluorescent labeling. The authors show that, in the presence of RNase inhibitor, LL37-associated RNA on NETs can promote a response by macrophages and keratinocytes that supports further inflammatory responses. This may provide important context for autoimmune disease such as psoriasis. Intriguingly, they show that RNA is present in resting neutrophil granules.

Statistical analysis is appropriate.

Some minor comments to address in revisions to manuscript text only (no new experiments):

1. The Barnes citation for reporting NETs in COVID-19 is not appropriate - it is a hypothesis piece and does not report any data supporting NET release in COVID-19 patients. This reference should be replaced with a primary literature source.
2. The 2019 Boeltz reference is rather outdated - there are certainly more than 1000 publications now 5 years later.
3. "NETs are intended for" - this is language I would avoid.
4. Host defense by NETs as a primary purpose is not likely a fair conclusion - most well studied or first studied, yes. Re-defining the primary purpose of NETs as DAMP release is a very strong statement...
5. Actual killing bacteria by NETs has been disputed so this should also be considered in the writing.

6. Chaotic assembly of the NET - is this really true? NETosis is a highly regulated process. I would remove this wording.

7. Line 317 - there is a typo in LL37

8. The quantification of NETs is a bit confusing - how many cells are in a given tile? Is it possible the method undercounts NETs if multiple NETs are in the same field of view? The figure supplied in Author Reply #35 could be included as a supplemental figure to help the reader to understand this not-often-reported approach to quantification.

9. The referenced Knight 2014 paper uses Cl-amidine which is not a specific PAD4 inhibitor but rather a pan-PAD inhibitor. Line 432 should be adjusted accordingly. The use of Cl-amidine in the manuscript should also not be classified as a PAD4 inhibitor but rather a nonselective PAD inhibitor.

10. To which temperature was FBS used in the in vitro NET experiments heat inactivated? This is an important experimental detail to add to the methods as the temperature may or may not have inactivated DNases/RNases in the serum (standard 56 or higher 70 degree inactivation?)

Arbitration review:

Bork and co-authors have provided an extensively revised manuscript in response to the thorough comments of the 3 peer reviewers. The strengths of this revision include the inclusion of additional supporting data for RNA identification on NETs. I think the authors have made substantial efforts to address the comments in both revision rounds. Some comments are out of scope of the current manuscript. I am, however, in general agreement with the title and substantial wording of the text being overreaching and going beyond what the data can conclude, particularly regarding purpose and intent.

The assessment of lack of novelty is rather superficial. Yes, it has been reported that RNA can induce immune responses. The authors should of course make sure to cite relevant sources for this in their introduction (some were suggested in the 2nd round of reviewer comments). Regarding the comment from reviewer, co-authors include experts in the field of RNA sensing and it is not unexpected that their prior publications would have some complementarity to this current manuscript. This manuscript reports the mechanism behind further immune response to self-packaged LL37-RNA complexes within neutrophil granules and released as part of NETs (termed NET-associated or na-RNA). The team has introduced this terminology in their 2020 Nature Communications paper (Herster et al., PMID 31913271). That paper showed that either bacterial-RNA-LL37 complex or na-RNA promotes additional NET release by neutrophils. This is done primarily in vitro and using human skin biopsies. A mouse model of imiquimod-driven psoriasis shows LL-37 and NETs in proximity but the mechanistic work performed in that model in that paper is generally limited showing the NET response itself (concluding TLR8/TLR13 dependence). One in vitro experiment shows TLR8-deleted monocytes unable to respond to LL37-RNA complexes.

Assessment that the data presented is too preliminary, I do not find to be a fair statement - all experiments have been performed in a manner that allows statistical analysis to make conclusions.

The use of 600 nM PMA has been addressed with new data which would be useful to incorporate into the EVs with a short line in the manuscript saying similar results were obtained with 60 nM PMA- 600 is a very high dose and indeed much higher than most of the literature using 25-100 nM for NET release. As the dose for RNA release does not need to be so high, this information will be helpful for readers for their future experiment that I'm sure will be motivated by reading this manuscript.

Comment 8 has already been addressed by the data provided by the reviewers showing differences in RNA composition between total RNA and naRNA.

Comment 9 is likely impossible to address - DNase will of course dismantle NETs and the relevance of experimentally producing RNA-only containing NETs has little biological relevance as this will probably never occur in vivo. The authors do show several methods of RNA detection so even if this one dye also binds DNA to an extent, this is something that can simply be mentioned as a limitation in the manuscript text.

Comment 10 is indeed also my experience - modifying the experimental setup to end up with the same number of unstimulated cells at the end of the experiment would mean starting with more cells in those conditions which is confounding.

Comment 12 - with the NF-kB experiments - overexpression of TLRs in the HEK293 cells generally makes them hypersensitive to responses - this type of subtle but still statistically significant reduction is not unexpected. The authors have other supporting data in primary cells.

Comment 13 - about detection of residual RNA - it's entirely possible that this would be possible to detect with adjusted microscope settings, which then would likely make the other signal saturated. I don't see the importance of demonstrating the residual RNA here, and the authors now explain that not all of their material stains the RNA added exogenously.

Comment 14 - For future reference to the authors, NETs could be digested using restriction enzymes into smaller fragments as has been demonstrated by Barrientos Front Immunol 2013. This could provide more quantitative results, but I do not find it necessary to add to this manuscript (microscopy is very commonly used for quantification even if it is much more labor intensive).

Comment about GSDMD and PAD4 dependence - the reviewer seems to have fallen into the trap that all NETs are produced by exactly the same pathway. It is very clear from the literature that different agonists induce NET release via different pathways, and that some pathways can override others. This does not mean there isn't an importance for these pathways in certain conditions. The same can be said about NADPH oxidase or NE dependence which were originally described at the "be-all-end-all" and have since also been demonstrated not to always be implicated in NETosis. I agree that this larger debate is out of scope for this manuscript as naRNA seems to be a common feature of NETs across a variety of stimuli and the discussion of it is not relevant to the current manuscript. An side note to the authors - with all due respect, the Kenny paper does NOT show PAD4-independent NET release (although others have, i.e. in cholesterol crystal induced NETosis). If you examine the supplemental data of the Kenny paper, you can see clearly that their PAD4-inhibition was not effective as histone citrullination is still high).

Citation of the Kato and Adase papers could be useful but not necessary.

New figure 4G and 4H are helpful additions to support the conclusions about NET differences.

Comment 21 seems to call into question use of IF in neutrophils, which is not a valid point as the authors have included relevant controls in their experiments. The authors have also added dye-based staining to support their findings with IF.

The PMA carryover concern is a legitimate one but has been addressed by the authors - this data should be added to the manuscript supplemental data.

The comment about demonstrating apoptosis with the inhibitors - this could be a line in the discussion but based on the data and based on the clear differences in cell biological changes I do not think apoptosis is responsible for the naRNA on NETs.

Comment 26 is an interesting one for a future manuscript; I agree it is out of scope of the current one.

The authors have made extensive efforts to further support pre-packaging of RNA into granules, which is indeed a major finding. I think this is quite convincingly demonstrated, but the wording of the text is very strong using terminology like "intended" or attempting to redefine the primary "purpose" of NETs. Generally the manuscript would strongly benefit from reducing the language regarding intent and purpose of this nicely demonstrated biological phenomenon. As noted in my independent review, the original claims of NETs as host defense are also called into question and the function is likely location/disease dependent and may not be the same in all individuals.

Comment 33 is also an interesting one for a future manuscript; it will likely take a large amount of extra studies to conclusively demonstrate this.

Comment 35 is a valid one and the quantification is a limitation in how it has been performed. I would suggest to add the image provided in the author response to the supplemental material so it is clear to all readers how this was performed and so they can make their own critical interpretation. I don't think this provides bias in a direction that it will change the overall conclusion, although it's unlikely to be a very accurate quantification based on the design (i.e. 1 NET in a tile is counted the same as 100 NETs in a tile).

The RNase comment is a valid one but can be discussed as future directions.

Comment 42 - the data is not available and repeating an experiment for this reason is indeed not in line with the 3R's principle. The authors have more detailed data than photographic images which would be demonstrative rather than quantitative.

Comment 43 on degranulation - also an interesting one but outside of the scope of the current manuscript.

Comment 44 - the new data provided is convincing and supports the conclusions of the manuscript.

Comment 45 - this is a comment I generally agree with. This language style may be more appropriate for presentations rather than written text. I find that it takes away from the story, which is strongly supported by the experimental data, rather than elevating it. The language regarding purpose and intent is actually quite offputting, and while it may elicit some positive feedback, I think the degree of harsh comments from all reviewers may highlight the potential negative side as well.

I agree to tone down the immune tolerance conclusions as per comment 48. This is a useful discussion point to keep in, but making clear the connection to previous literature (cited by the authors) vs the current work.

Universität Tübingen[®] Interfakultäres Institut für Zellbiologie
Abt. Immunologie[®] Auf der Morgenstelle 15[®] 72076Tübingen

Department of Immunology

Prof. Dr. Hans-Georg Rammensee
Direktor

Office Management

Lynne Yakes
Tel.: +49 7071 29-87628
Fax: +49 7071 29-5653
E-mail: lynne.yakes@uni-tuebingen.de
<http://www.immunology-tuebingen.de/>

Gesprächspartner/Your contact person:

Prof. Dr. Alexander Weber
Professor of Innate Immunity

Tel.: +49 7071 29-87623
Mobil: +49-173 215 7220
Fax: +49 7071 29-4759

Tübingen, 16 April 2024

We sincerely thank the Editor for the chance to appeal and the arbitrator for his/her unbiased review and appreciation of our work. Enclosed is a point-by-point response to the comments of the arbitrator. Authors' responses to arbitrator's comments are in blue.

We look forward to seeing our manuscript published in *EMBO reports*.

Point-by-point response

Arbitrator:

My independent review:

RNA is present on NETs released by a variety of different agonists (mouse and human) - shown by antibody staining as well as 5-EU incorporation with click chemistry fluorescent labeling. The authors show that, in the presence of RNase inhibitor, LL37-associated RNA on NETs can promote a response by macrophages and keratinocytes that supports further inflammatory responses. This may provide important context for autoimmune disease such as psoriasis. Intriguingly, they show that RNA is present in resting neutrophil granules.

Statistical analysis is appropriate.

We thank the reviewer for this assessment.

Some minor comments to address in revisions to manuscript text only (no new experiments):

1. The Barnes citation for reporting NETs in COVID-19 is not appropriate - it is a hypothesis piece and does not report any data supporting NET release in COVID-19 patients. This reference should be replaced with a primary literature source.

The Barnes citation has been replaced with (Middleton et al, 2020), Line 78.

2. The 2019 Boeltz reference is rather outdated - there are certainly more than 1000 publications now 5 years later.

We appreciate the reviewer's comment. We cited the Boeltz citation as it was authored by 50 or so authors from the NET field with a view to provide the broadest possible consensus review of the field. To our knowledge there is no more recent review that would have such a broad authorship/scope. Nevertheless, we agree with the reviewer that much has happened since 2019 and we therefore additionally cite (Melbouci et al, 2023) as a more recent review article, Line 80.

3. "NETs are intended for" - this is language I would avoid.

We appreciate the reviewer's concern and have re-phrased this, depending on the context with e.g. "inevitably entails", "naturally contains" or "inevitably". We hope this is an acceptable rendering of the idea that neutrophils are naturally able and "set up" to release naRNA-LL37 DAMPs during the NET process.

4. Host defense by NETs as a primary purpose is not likely a fair conclusion - most well studied or first studied, yes. Re-defining the primary purpose of NETs as DAMP release is a very strong statement...

We thank the reviewer for this feedback and apologize that we may have overstated our case. We have re-phrased this as follows:

"On the other hand, inflammation is essential for sterile insult removal, too, and the fact that NETs have been described for a multitude of sterile conditions indicates that antimicrobial activity is only one of several functions of NETs. Our findings support the notion that the NET response in the absence of infection is inevitably a DAMP response, an important new facet in the concepts in NET biology." (Line 372 and following).

"As most pathological conditions involving NETs are sterile, our data argue that a key function of sterile NET formation is DAMP release,..." (Line 118 and following).

5. Actual killing bacteria by NETs has been disputed so this should also be considered in the writing.

We agree with the reviewer that this has been disputed (see e.g. (Menegazzi et al, 2012)) and killing (i.e. the effect on CFU) was moderate in our hands. As antimicrobial effects of naRNA were, however, not the main point of our work, we chose to remain relatively silent about the existing

controversies. As suggested by the reviewer, we have now acknowledged that killing has also been questioned. Examples of rephrasing are “NETs counteract microbes” instead of “NETs trap and kill” (abstract); “NETs were proposed to be able to trap and kill” instead of just “trap and kill”; and “the first proposed function is ... killing ... (Brinkmann 2004) ” instead of the “the primary function is... killing” (Line 150). We hope this adequately reflects the situation. The reviews cited in our work ((Boeltz et al, 2019; Kruger et al, 2015)) mention (Menegazzi et al., 2012) so that the curious reader is put into a position to get a balanced view of the field should they choose so.

6. Chaotic assembly of the NET - is this really true? NETosis is a highly regulated process. I would remove this wording.

The adjective “chaotic” has been removed.

7. Line 317 - there is a typo in LL37

We thank the arbitrator for spotting the typo and have corrected it.

8. The quantification of NETs is a bit confusing - how many cells are in a given tile? Is it possible the method undercounts NETs if multiple NETs are in the same field of view? The figure supplied in Author Reply #35 could be included as a supplemental figure to help the reader to understand this not-often-reported approach to quantification.

We thank the arbitrator for this feedback. The quantification of NETs is indeed challenging and it is nearly impossible to identify how many NETs are in the same field of view as they may overlap from different neutrophils. Starting with identical numbers of cells aliquoted from the same prep, we felt that quantifying the number of tiles was a suitable way to quantify NETs in comparisons between different stimulatory conditions. In response to the reviewer’s comment, we have added a comment to the Methods section (Line 751), that the number of NETs may be underestimated when there is strong NETosis, due to potential overlap of NETs, which cannot be excluded by our method. Moreover, we included the figure in a new Appendix figure 2 called out in the Methods section.

9. The referenced Knight 2014 paper uses Cl-amidine which is not a specific PAD4 inhibitor but rather a pan-PAD inhibitor. Line 432 should be adjusted accordingly. The use of Cl-amidine in the manuscript should also not be classified as a PAD4 inhibitor but rather a nonselective PAD inhibitor.

The wording has been changed based on the suggestion of the arbitrator. In Line 433 “PAD4” was changed to “PAD” to correctly apply to the Knight 2014 paper. In Line 195, Cl-amidine was re-phrased “an inhibitor of multiple PADs”, the figure EV3A legend was also amended accordingly (Line 1392).

10. To which temperature was FBS used in the in vitro NET experiments heat inactivated? This is an important experimental detail to add to the methods as the temperature may or may not have inactivated DNases/RNases in the serum (standard 56 or higher 70 degree inactivation?)

Heat inactivation was done at 56 °C. This important detail has been added (Lines 449 and 530).

Arbitration review:

Bork and co-authors have provided an extensively revised manuscript in response to the thorough comments of the 3 peer reviewers. The strengths of this revision include the inclusion of additional supporting data for RNA identification on NETs. I think the authors have made substantial efforts to address the comments in both revision rounds. Some comments are out of scope of the current manuscript. I am, however, in general agreement with the title and substantial wording of the text being overreaching and going beyond what the data can conclude, particularly regarding purpose and intent.

We appreciate the clear feedback and have sought to amend the wording accordingly.

The assessment of lack of novelty is rather superficial. Yes, it has been reported that RNA can induce immune responses. The authors should of course make sure to cite relevant sources for this in their introduction (some were suggested in the 2nd round of reviewer comments).

We thank the arbitrator for this assessment and have added further references apart from the already cited papers by the Gilliet group. Line 94 now reads: "Synthetic or viral double-stranded RNA was also shown to act in concert with LL37 (Adase et al, 2016; Kato et al, 2023)." We hope this provides sufficient detail to the actions of RNA and LL37.

Regarding the comment from reviewer, co-authors include experts in the field of RNA sensing and it is not unexpected that their prior publications would have some complementarity to this current manuscript. This manuscript reports the mechanism behind further immune response to self-packaged LL37-RNA complexes within neutrophil granules and released as part of NETs (termed NET-associated or na-RNA). The team has introduced this terminology in their 2020 Nature Communications paper (Herster et al., PMID 31913271). That paper showed that either bacterial-RNA-LL37 complex or na-RNA promotes additional NET release by neutrophils. This is done primarily in vitro and using human skin biopsies. A mouse model of imiquimod-driven psoriasis shows LL-37 and NETs in proximity but the mechanistic work performed in that model in that paper is generally limited showing the NET response itself (concluding TLR8/TLR13 dependence). One in vitro experiment shows TLR8-deleted monocytes unable to respond to LL37-RNA complexes.

Assessment that the data presented is too preliminary, I do not find to be a fair statement - all experiments have been performed in a manner that allows statistical analysis to make conclusions.

We thank the arbitrator for appreciating the novelty and the statistical power of our analyses.

The use of 600 nM PMA has been addressed with new data which would be useful to incorporate into the EVs with a short line in the manuscript saying similar results were obtained with 60 nM PMA- 600 is a very high dose and indeed much higher than most of the literature using 25-100 nM for NET release. As the dose for RNA release does not need to be so high, this information will be helpful for readers for their future experiment that I'm sure will be motivated by reading this manuscript.

We appreciate the arbitrator's comment. In the first revision we had added in Line 570: "Similar results were obtained when NETs were generated using 60 nM PMA or the ionophore A23187, i.e. NETs generated with lower or alternative stimuli also functioned as DAMPs". This was previously referenced as "data not shown" but now amended in showing the data in a new Appendix figure 1.

Comment 8 has already been addressed by the data provided by the reviewers showing differences in RNA composition between total RNA and nRNA.

We appreciate the arbitrator's comment.

Comment 9 is likely impossible to address - DNase will of course dismantle NETs and the relevance of experimentally producing RNA-only containing NETs has little biological relevance as this will probably never occur in vivo. The authors do show several methods of RNA detection so even if this one dye also binds DNA to an extent, this is something that can simply be mentioned as a limitation in the manuscript text.

We thank the arbitrator for taking a similar view and have added a comment to Line 727: "Although high specificity to RNA has been confirmed in multiple studies (Li et al, 2006; Wu et al, 2020; Zhou et al, 2015) and the line plot analysis in Fig. EV5C-E supports this, a very low binding to DNA cannot be fully excluded".

Comment 10 is indeed also my experience - modifying the experimental setup to end up with the same number of unstimulated cells at the end of the experiment would mean starting with more cells in those conditions which is confounding.

We appreciate the arbitrator's comment.

Comment 12 - with the NF- κ B experiments - overexpression of TLRs in the HEK293 cells generally makes them hypersensitive to responses - this type of subtle but still statistically significant reduction is not unexpected. The authors have other supporting data in primary cells.

We thank the arbitrator for this comment.

Comment 13 - about detection of residual RNA - it's entirely possible that this would be possible to detect with adjusted microscope settings, which then would likely make the other signal saturated. I don't see the importance of demonstrating the residual RNA here, and the authors now explain that not all of their material stains the RNA added exogenously.

Thank you for this comment.

Comment 14 - For future reference to the authors, NETs could be digested using restriction enzymes into smaller fragments as has been demonstrated by Barrientos Front Immunol 2013. This could provide more quantitative results, but I do not find it necessary to add to this manuscript (microscopy is very commonly used for quantification even if it is much more labor intensive).

We appreciate the arbitrator's suggestions for amending our experimental protocol in future.

Comment about GSDMD and PAD4 dependence - the reviewer seems to have fallen into the trap that all NETs are produced by exactly the same pathway. It is very clear from the literature that different agonists induce NET release via different pathways, and that some pathways can override others. This does not mean there isn't an importance for these pathways in certain conditions. The same can be said about NADPH oxidase or NE dependence which were originally described at the "be-all-end-all" and have since also been demonstrated not to always be implicated in NETosis. I agree that this larger debate is out of scope for this manuscript as naRNA seems to be a common feature of NETs across a variety of stimuli and the discussion of it is not relevant to the current manuscript.

We are relieved to hear that the arbitrator agrees with our reasoning.

An side note to the authors - with all due respect, the Kenny paper does NOT show PAD4-independent NET release (although others have, i.e. in cholesterol crystal induced NETosis). If you examine the supplemental data of the Kenny paper, you can see clearly that their PAD4-inhibition was not effective as histone citrullination is still high).

We thank the arbitrator for this point and have added the reference (Knuckley et al, 2010) (Line 194) as this showed the characterization across multiple PADs. Kenny et al was completed by the suggested paper by (Warnatsch et al, 2015) (Line 194) to provide references for both Cl-amidine inhibited and unperturbed NET pathways.

Citation of the Kato and Adase papers could be useful but not necessary.

We appreciate the comment and chose not to discuss their work as it concerns dsRNA (which is not our focus), mainly uses the surrogate poly(I:C) and is also not relevant in this sterile setting (i.e. the absence of viral dsRNA). Nevertheless the citations were added to the introduction (Line 94).

New figure 4G and 4H are helpful additions to support the conclusions about NET differences.

We appreciate the positive comment.

Comment 21 seems to call into question use of IF in neutrophils, which is not a valid point as the authors have included relevant controls in their experiments. The authors have also added dye-based staining to support their findings with IF.

We thank the arbitrator for this point which reflects well the wide use of IF for examining neutrophil responses.

The PMA carryover concern is a legitimate one but has been addressed by the authors - this data should be added to the manuscript supplemental data.

We appreciate the comment and have added this point to the methods (Line 563) and Appendix figure 1.

The comment about demonstrating apoptosis with the inhibitors - this could be a line in the discussion but based on the data and based on the clear differences in cell biological changes I do not think apoptosis is responsible for the naRNA on NETs.

We thank the reviewer. We agree that this could be useful but felt the discussion was fragmented by introducing this point. We have instead included a short line along the argument in the previous point-by-point reply in the Methods that there were no visible signs of apoptosis, although this was not formally ruled out (Line 542-544).

Comment 26 is an interesting one for a future manuscript; I agree it is out of scope of the current one.

This is much appreciated.

The authors have made extensive efforts to further support pre-packaging of RNA into granules, which is indeed a major finding. I think this is quite convincingly demonstrated, but the wording of the text is very strong using terminology like "intended" or attempting to redefine the primary "purpose" of NETs. Generally the manuscript would strongly benefit from reducing the language

regarding intent and purpose of this nicely demonstrated biological phenomenon. As noted in my independent review, the original claims of NETs as host defense are also called into question and the function is likely location/disease dependent and may not be the same in all individuals.

Thank you for this feedback. As outlined above the wording along the lines of “intended” and “purpose” has been amended.

Comment 33 is also an interesting one for a future manuscript; it will likely take a large amount of extra studies to conclusively demonstrate this.

We appreciate the arbitrator’s recognition that could be the topic of a follow-up study.

Comment 35 is a valid one and the quantification is a limitation in how it has been performed. I would suggest to add the image provided in the author response to the supplemental material so it is clear to all readers how this was performed and so they can make their own critical interpretation. I don't think this provides bias in a direction that it will change the overall conclusion, although it's unlikely to be a very accurate quantification based on the design (i.e. 1 NET in a tile is counted the same as 100 NETs in a tile).

This is noted and is discussed above. The quantification figure was added as Appendix figure 2.

The RNase comment is a valid one but can be discussed as future directions.

This is discussed as a limitation in Lines 352-355.

Comment 42 - the data is not available and repeating an experiment for this reason is indeed not in line with the 3R's principle. The authors have more detailed data than photographic images which would be demonstrative rather than quantitative.

This is appreciated.

Comment 43 on degranulation - also an interesting one but outside of the scope of the current manuscript.

This will indeed be an interesting question for future study, thank you.

Comment 44 - the new data provided is convincing and supports the conclusions of the manuscript.

Gratefully acknowledged.

Comment 45 - this is a comment I generally agree with. This language style may be more appropriate for presentations rather than written text. I find that it takes away from the story, which is strongly supported by the experimental data, rather than elevating it. The language

regarding purpose and intent is actually quite offputting, and while it may elicit some positive feedback, I think the degree of harsh comments from all reviewers may highlight the potential negative side as well.

We concede this point and have sought to amend this as outlined above and by removing e.g. "road blocks" etc.

I agree to tone down the immune tolerance conclusions as per comment 48. This is a useful discussion point to keep in, but making clear the connection to previous literature (cited by the authors) vs the current work.

We appreciate the comment. We have sought to amend this by stating that our data only indirectly apply to our understanding of tolerance to self-RNA (Line 356) and have revised the paragraph following to delineate better previous work and concepts and how they are affected by our observations.

Literature cited

Adase CA, Borkowski AW, Zhang LJ, Williams MR, Sato E, Sanford JA, Gallo RL (2016) Non-coding Double-stranded RNA and Antimicrobial Peptide LL-37 Induce Growth Factor Expression from Keratinocytes and Endothelial Cells. *J Biol Chem* 291: 11635-11646

Boeltz S, Amini P, Anders HJ, Andrade F, Bilyy R, Chatfield S, Cichon I, Clancy DM, Desai J, Dumych T *et al* (2019) To NET or not to NET: current opinions and state of the science regarding the formation of neutrophil extracellular traps. *Cell Death Differ* 26: 395-408

Kato H, Ohta K, Akagi M, Fukada S, Sakuma M, Naruse T, Nishi H, Shigeishi H, Takechi M, Aikawa T (2023) LL-37-dsRNA Complexes Modulate Immune Response via RIG-I in Oral Keratinocytes. *Inflammation* 46: 808-823

Knuckley B, Causey CP, Jones JE, Bhatia M, Dreyton CJ, Osborne TC, Takahara H, Thompson PR (2010) Substrate specificity and kinetic studies of PADs 1, 3, and 4 identify potent and selective inhibitors of protein arginine deiminase 3. *Biochemistry* 49: 4852-4863

Kruger P, Saffarzadeh M, Weber AN, Rieber N, Radsak M, von Bernuth H, Benarafa C, Roos D, Skokowa J, Hartl D (2015) Neutrophils: Between host defence, immune modulation, and tissue injury. *PLoS Pathog* 11: e1004651

Li Q, Kim Y, Namm J, Kulkarni A, Rosania GR, Ahn YH, Chang YT (2006) RNA-selective, live cell imaging probes for studying nuclear structure and function. *Chem Biol* 13: 615-623

Melbouci D, Haidar Ahmad A, Decker P (2023) Neutrophil extracellular traps (NET): not only antimicrobial but also modulators of innate and adaptive immunities in inflammatory autoimmune diseases. *RMD Open* 9

Menegazzi R, Decleva E, Dri P (2012) Killing by neutrophil extracellular traps: fact or folklore? *Blood* 119: 1214-1216

Middleton EA, He XY, Denorme F, Campbell RA, Ng D, Salvatore SP, Mostyka M, Baxter-Stoltzfus A, Borczuk AC, Loda M *et al* (2020) Neutrophil extracellular traps contribute to immunothrombosis in COVID-19 acute respiratory distress syndrome. *Blood* 136: 1169-1179

Warnatsch A, Ioannou M, Wang Q, Papayannopoulos V (2015) Inflammation. Neutrophil extracellular traps license macrophages for cytokine production in atherosclerosis. *Science* 349: 316-320

Wu Y, Liu Y, Lu C, Lei S, Li J, Du G (2020) Quantitation of RNA by a fluorometric method using the SYTO RNASelect stain. *Anal Biochem* 606: 113857

Zhou B, Liu W, Zhang H, Wu J, Liu S, Xu H, Wang P (2015) Imaging of nucleolar RNA in living cells using a highly photostable deep-red fluorescent probe. *Biosens Bioelectron* 68: 189-196

Prof. Alexander Weber
University of Tübingen
Innate Immunity
Auf der Morgenstelle 15
Tübingen
Germany

Dear Prof. Weber,

I am very pleased to accept your manuscript for publication in the next available issue of EMBO reports. Thank you for your contribution to our journal.

Yours sincerely,
